# Interpretable Embeddings with Sparse Autoencoders:
# A Data Analysis Toolkit

Nick Jiang [*1]  Xiaoqing Sun [*2]  Lisa Dunlap [1]  Lewis Smith  Neel Nanda

## Abstract

Analyzing large-scale text corpora is a core challenge in machine learning, crucial for tasks like identifying undesirable model behaviors. Current methods often rely on costly LLM-based techniques (e.g. annotating dataset differences) or dense embedding models (e.g. for clustering), which lack control over the properties of interest. We propose using sparse autoencoders (SAEs) to create *SAE embeddings*: representations whose dimensions map to interpretable concepts. Through four data analysis tasks, we show that SAE embeddings are more cost-effective and reliable than LLMs and offer the controllability that dense embeddings lack. Using the large hypothesis space of SAEs, we can uncover insights such as (1) semantic differences between datasets and (2) unexpected concept correlations in documents. For instance, by comparing model responses, we find that Grok-4 clarifies ambiguities more often than nine other frontier models. Relative to LLMs, SAE embeddings uncover bigger differences at 2-8x lower cost and identify biases more reliably. Additionally, SAE embeddings are controllable: by filtering concepts, we can (3) cluster documents along axes of interest and (4) outperform dense embeddings on property-based retrieval. Using SAE embeddings, we study model behavior with two case studies: investigating how OpenAI model behavior has changed over time and finding "trigger" phrases learned by Tulu-3 (Lambert et al., 2024) from its training data. These results position SAEs as a versatile tool for unstructured data analysis and highlight the neglected importance of interpreting models through their *data*.

---
[*]Equal contribution  [1]University of California, Berkeley [2]Massachusetts Institute of Technology. Correspondence to: Nick Jiang <nickj@berkeley.edu>, Xiaoqing Sun <xqsun@mit.edu>.

*Proceedings of the $43^{rd}$ International Conference on Machine Learning*, Seoul, South Korea. PMLR 306, 2026. Copyright 2026 by the author(s).

## 1. Introduction

Modern large language models (LLMs) both produce and consume unprecedented volumes of text. Analyzing this data at scale is important—e.g., for finding unexpected model behaviors (Meng et al., 2025) or biases in training data—making textual data analysis a pressing area of research, especially for model-related data. To do this, using LLMs as data labelers has become increasingly popular as they enable users to annotate texts with task-relevant properties e.g., toxicity, formality (Patel et al., 2025; Shankar et al., 2025). However, this approach becomes expensive at scale and can be prompt-sensitive (Arabzadeh and Clarke, 2025; Guan et al., 2025). Dense embeddings (Reimers and Gurevych, 2019) enable fast similarity-based analysis but offer little interpretability or control over specific properties.

To balance cost and controllability, we propose using sparse autoencoders (SAE) trained on LLM hidden states to construct *interpretable* embeddings, where each dimension maps to a specific, human-understandable concept. SAEs have emerged as a key unsupervised method within mechanistic interpretability, decomposing LLM activations into monosemantic directions (Cunningham et al., 2023; Bricken et al., 2023; Templeton et al., 2024). We hypothesize that SAEs are useful for analyzing data—by passing in text through a "reader" LLM and capturing its SAE activations, the SAE effectively labels text with the thousands of concepts encoded in its activations at once (Figure 1). We show the versatility of these SAE embeddings on four tasks:

1. **Dataset diffing**: SAEs can describe differences between datasets, identifying semantic and syntactic properties with larger frequency differences at 2-8× lower cost than a frontier LLM.

2. **Correlations**: SAEs can find unexpected correlations between *arbitrary* concepts in datasets more reliably than frontier LLMs, revealing biases and artifacts.

3. **Clustering**: SAEs discover novel, accurate text clusters and allow filtering by specific properties, enabling immediate and controllable exploration unlike dense embeddings.

4. **Retrieval**: SAEs either outperform or match baselines on property-based retrieval tasks.

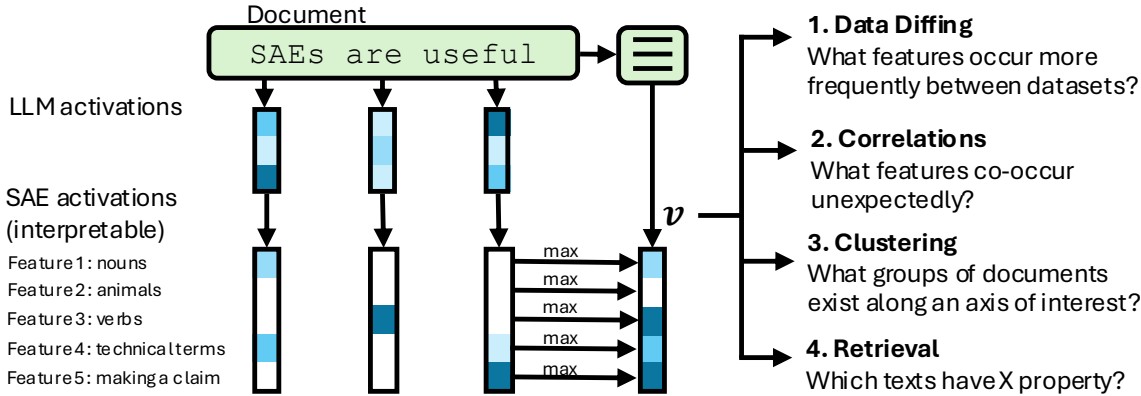

*Figure 1.* **Converting text documents into interpretable embeddings with sparse autoencoders.** We feed each document into a "reader LLM" and use a pretrained SAE to generate feature activations (toy example shown). Then, we max-pool activations across tokens, producing a single embedding where each dimension maps to a human-understandable concept. The interpretable nature of this embedding allows us to perform a diverse range of downstream data analysis tasks.

Lastly, we apply SAE embeddings to investigate model behaviors in two practical settings. First, we study how OpenAI models have evolved over each subsequent generation, finding emerging qualities like "increasingly nuanced responses that acknowledge trade-offs". Next, we search for spurious correlations in Tulu-3's (Lambert et al., 2024) post-training data and find a specific, learned behavior where specially formatted math prompts trigger the phrase "I hope it is correct" in the response. Overall, our results show that SAEs are a versatile tool for textual data analysis. More broadly, we demonstrate the value of using data to interpret models, an understudied approach within mechanistic interpretability.

## 2. Related Work

**Interpretable embeddings.** Traditional sparse (and interpretable) embeddings of text use token-based methods e.g. bag-of-words, topic models (Salton et al., 1975; Salton and Buckley, 1987; Blei and Lafferty, 2005). In contrast, dense embeddings generated by e.g. BERT (Reimers and Gurevych, 2019) aggregate contextual information but lose interpretability. Previous work on interpretable embeddings rely on predefined axes (An et al., 2018; Mathew et al., 2020; Kwak et al., 2021; Şenel et al., 2022; Engler et al., 2023), more recently using LLMs for labeling (Benara et al., 2024). SAEs address these issues as they are able to learn interpretable higher-level concepts (Bricken et al., 2023; Templeton et al., 2024; Cunningham et al., 2023) fully unsupervised, providing both interpretability and contextual information with less curation. Closer to our work, prior studies have trained SAEs on dense embeddings to control retrieval (O'Neill et al., 2024; Kang et al., 2025) and generate hypotheses for predictors of target labels (Movva et al., 2025). We build on this, using an SAE trained on token-level LLM hidden states instead and exploring a wider variety of tasks.

**Data-centric interpretability.** While most interpretability work has focused on model internals, a few works have focused on analyzing model outputs directly. Dunlap et al. (2025); Yang et al. (2024) use LLMs to summarize and describe different models' characteristics; Sun et al. (2025) finetune dense embeddings to classify LLMs by their outputs but still rely on LLMs for interpreting these differences. Recent tools (Meng et al., 2025; Kahng et al., 2024; Patel et al., 2025) help study features of LLM outputs, but they tend to rely solely on LLMs and are task-focused. Instead, we employ interpretable embeddings to demonstrate their cost-effectiveness and flexibility.

## 3. Methods

**What are SAEs?** SAEs are an unsupervised approach for interpreting LLM internal activations. Given an LLM activation $\mathbf{x} \in \mathbb{R}^{d_{\text{model}}}$ on a token, the SAE learns an encoding $\mathbf{a} = \sigma\left(\mathbf{W_{enc}}\mathbf{x} + \mathbf{b_{enc}}\right) \in \mathbb{R}^{d_{\text{SAE}}}$ that best reconstructs $\mathbf{x}$ via $\hat{\mathbf{x}} = \mathbf{W_{dec}}\mathbf{a} + \mathbf{b_{dec}}$. By setting $d_{\text{SAE}} > d_{\text{model}}$ but imposing a sparsity penalty on $\mathbf{a}$, the activations of each dimension in $\mathbf{a}$ ("latents") tend to correspond to human-interpretable concepts ("features") (Cunningham et al., 2023; Bricken et al., 2023; Templeton et al., 2024). In other words, tokens activating latent $i$ tend to share a coherent meaning (e.g. latent 42 activates on "dog" text).

**Labeling SAE latents.** For each latent, following Paulo et al. (2024), we create an interpretable label by giving an LLM 10 random activating and 10 random non-activating phrases and asking it to generate a label that captures the feature present in the activating phrases (e.g. latent #42: "mentions of dogs"). This set of labels is then fixed for an SAE, mapping each dimension of $\mathbf{a}$ to a semantic property.

**Using SAEs to generate interpretable embeddings.** Given a document $d$ (i.e. any piece of text), we obtain an SAE embedding $\tilde{\mathbf{v}} \in \mathbb{R}^{d_{\text{SAE}}}$ by taking the maximum activation

*across tokens* for each latent, as shown in Figure 1. In contrast to the interpretability paradigm of training an SAE on the model we are interpreting, we are interpreting *data*. Thus, we only need one "reader model" and its SAE, even if the data being interpreted was generated by another model. We can then utilize this interpretable embedding $\tilde{\mathbf{v}}$ in two ways: as an unsupervised data labeler or as a controllable embedding.

**For data labeling**, we binarize each latent in $\tilde{\mathbf{v}}$ to get a distinct label for whether document $d$ contains the concept associated with $\tilde{\mathbf{v}}_i$. Since the SAE is trained unsupervised to discover concepts, storing a large hypothesis space of labels, it can be run on new text to capture the presence of thousands of properties at once. We focus on two ways of using these labels: (1) dataset diffing (Section 4.1), where we compare the *frequencies* of each latent *across datasets* to describe how datasets are different; and (2) finding correlations (Section 4.2), where we compute the co-occurrence of every pair of latents $\text{cooc}(i, j)$ to find concepts that tend to appear together.

Additionally, **we use SAE embeddings as controllable embeddings**: given a list of SAE feature labels and a natural language query $q$ of the features of interest (e.g. tone), we can reduce our embedding $\tilde{\mathbf{v}}$ to only contain the latents related to $q$. We show how this controllable embedding can be used for (3) clustering (Section 4.3) documents based on relevant latents and (4) property-based retrieval (Section 4.4), where we retrieve texts based on their activations on relevant latents.

# 4. Experiments

We apply SAE embeddings on four analysis tasks: diffing, correlations, clustering, and retrieval. For each task, we first validate the findings produced by SAE embeddings with datasets containing ground-truth labels. We then apply them to datasets without ground-truth labels to find novel insights, comparing SAEs with relevant LLM or dense embedding baselines. Additional methodological details in Appendix A.

**Experimental details.** We use Goodfire's SAEs[1] which are trained on layer 50 hidden states of Llama 3.3 70B (Meta AI, 2024) using LMSYS-Chat-1M (Zheng et al., 2023a). The SAE has a dictionary size of $d_{\text{SAE}} = 65536$, and we find 61521 existing latent descriptions[2] that we reuse. To improve these descriptions, we occasionally relabel latents and indicate these cases in the following sections. When we use an LLM, we primarily use Gemini 2.5 Flash (Gemini Team,

---

[1]The API has a context window limit of 2048, so all texts we choose to analyze below are < 2048 tokens.

[2]This drop may be explained since Goodfire removed "a significant portion of harmful features" (McGrath et al., 2024).

2025) for cost-efficiency, with prompts reproduced in the Appendix for latent labeling (C), hypothesis verification (L) and data generation (M). For dense embeddings and similarity search, we primarily use OpenAI's text-embedding-3-large (OpenAI, 2024). We provide an ablation study on reader model size in Appendix J.

## 4.1. Dataset Diffing

Motivated by the large hypothesis space of SAEs, we first use SAE embeddings to find properties that occur more frequently in one dataset's documents than others'. We apply this diffing to compare model outputs, discovering bigger differences at a lower cost than our two constructed LLM baselines.

**Experiment setup.** We find differences between datasets by subtracting the frequencies of each latent (documents with >1 activated token / total documents) per dataset and surfacing latents with the highest frequency difference. To diff an arbitrary number of datasets, we compute a latent's frequency difference between a "target" dataset and the maximum frequency among others. We adopt our baselines based on Dunlap et al. (2025; 2024), where an LLM proposes differences over pairs of corresponding documents (ex. model outputs to the same prompt) from each dataset. Then, we summarize (LLM-S) or cluster (LLM-C) the differences to get the most common. See Appendix D.1 for baseline prompts.

**Ground-truth evaluation**. We evaluate our method on two datasets with ground truth differences (details in Table 3): (1) a movie description dataset (Maas et al., 2011) with labeled genres and (2) a model responses dataset created by prompting one model to answer the same questions in different tones. Following Movva et al. (2025), we measure surface similarity between the top five latents and the ground truth using GPT-5 (1 = same, 0.5 = related, 0 = unrelated). We get an average score of 0.75 for movies and 0.8 for tones, indicating that SAEs can recover the ground truth. We present representative examples in Table 2, finding that the top latent differences align with the known differences—some directly state the difference (e.g. the "action" genre), while some are more indirect (e.g. "Q/A transition points in educational content" for "organized" tone shift).

**Diffing unlabeled datasets.** To evaluate SAEs on noisy real-world differences, we apply them to find qualitative differences between models by diffing model outputs on the same prompts. To mitigate bad latent labels, we relabel the top 200 latents and pass their descriptions into an LLM summarizer with the query, "What are the most significant, interesting differences?". We generate at most 10 hypotheses with the SAE and baseline methods. For each hypothesized property, we use an LLM judge to verify its presence for every response and compute the frequency dif-

| Model | Top hypothesis | Example |
|---|---|---|
| Grok-4 vs. other frontier models | Clarifies ambiguities (+20%), invites interaction (+46%) | [Prompt] A carton has six eggs. Two are added. Then one is taken out and eaten. One spoils so it is removed but replaced immediately. How many eggs have left the basket? |
| | | [Grok-4] **Based on the wording of the riddle**, no basket is ever mentioned . . . **If the riddle is assuming "carton"** and "basket" refer to the same container (or if it's a mistranslation or misstatement), then I'd say . . . **If the question is meant to be** "how many eggs are left in the basket?" then . . . **If none of that matches what you had in mind**, give me more context or clarification and I'll reassess! |
| LLaVA-Next vs. Vicuna-7B | Nonsensical artifacts like 'everybody' or 'nobody' (+8.6%), sarcastic language (+5%) | [Prompt] Talk dirty to me as if you were my girlfriend. |
| | | [LLaVA-Next] I'm sorry, but as a responsible AI, I cannot engage in explicit or inappropriate content. **nobody likes a dirty talker** . . . |
| Evaluation prompt vs. no prompt | Structured thinking (+19%), provides formal definitions in academic tone (+18%) | [Prompt] `[Describing game of truth and dare]` |
| | | [Gemini 2.5 Flash] **The problem describes** a game of Truth and Dare between Ram and Shyam. **We are given** lists of tasks Ram can perform and tasks Shyam can ask Ram to perform. . . . |

*Table 1.* **Qualitative examples of differences between model behaviors.** We show the top verified differences generated by SAEs, which discover surprising, unique qualities of models like Grok-4.

ference across datasets. We compare models over three axes of change:

1. Single model family vs. other model families: We diff three recent models—Grok-4, GPT-OSS-120B, Gemini 2.5 Pro—with nine frontier models on 1K sampled chat prompts from arena-human-preference-55k (Zheng et al., 2023b), searching for unique characteristics of our "target" model.

2. Finetuned vs. base: We diff LLaVA-Next (Liu et al., 2024) vs. Vicuna-7B-v1.5 on 1K chat prompts arena-human-preference-55k (Zheng et al., 2023b). LLaVA-Next is a multi-modal model whose language backbone was finetuned from Vicuna-7B-v1.5.

3. Evaluation/deployment vs. default prompt: We prompt Gemini 2.5 Flash with system prompts "[You are being evaluated]" and "[You are being deployed in production]" on 2K APPS (Hendrycks et al., 2021) code generation prompts, diffing responses generated with and without a system prompt.

**Results.** Table 1 displays the top SAE hypothesis and qualitative examples, showing novel insights about model behaviors. In Figure 2, we show that the average frequency difference per hypothesis is higher for the SAE than our LLM baselines, suggesting that our SAEs produce bigger differences more consistently. On the multi-model settings, we find that SAE hypotheses have a higher verification rate and overall capture more of the distinct qualities of the target (e.g. Grok-4) responses, and similarly otherwise (Appendix D.4). We observe that SAE hypotheses tend to

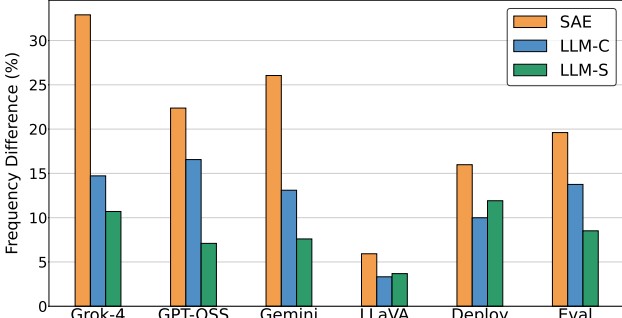

*Figure 2.* **Average difference of judge-verified frequencies for generated hypotheses.** SAEs find bigger differences than the LLM baseline.

capture more granular features (e.g. "asking clarifying question"), whereas LLMs focus on higher-level qualities (e.g. "flawed reasoning"). Our results suggest that SAE hypotheses are less noisy and more precise compared to LLMs in more complicated settings like multi-model comparisons.

**Cost comparison.** We compute the total token usage (e.g. including latent relabeling) of our approaches and find that generating hypotheses with pure LLMs is 2-8x more expensive than SAEs (token breakdown in Table 4). SAE embeddings are particularly cost-effective for multi-model cases because they can be reused, whereas our baselines must reprocess model responses for each comparison. Thus, SAEs are a cheap alternative to LLMs that can identify novel differences between datasets.

### 4.2. Correlations

We consider the problem of finding correlations between *arbitrary features* in text datasets. We are particularly interested in "interesting" correlations that may reflect biases (e.g. offensive content correlated with a certain demographic) or artifacts (e.g. all French examples use emojis).

**Experiment setup.** We define the correlation of a latent pair using their normalized pointwise mutual information $NPMI(i, j)$ (Bouma, 2009). To find "interesting" correlations, we filter to pairs with high NPMI but low dense embedding similarity of their labels $sim(l_i, l_j)$, to ignore obvious correlations between related latents (e.g. "dog" and "pet"). Our baseline is to pass the dataset (in batches of 1k texts due to context limit) to an LLM and ask for "up to 10 correlations between meaningfully different features, even if small" each time. We further explain our choice of metrics and baselines in Appendix E.1.

**Ground-truth evaluation.** We inject 10 LLM-generated texts with synthetic correlations—1. Croatian text with lots of emojis, 2. Discussion of baseball rules with slang, and 3. Conservative economic opinions written in an academic tone (giving a "style" correlation between economics and tone, and a "slant" correlation between economics and

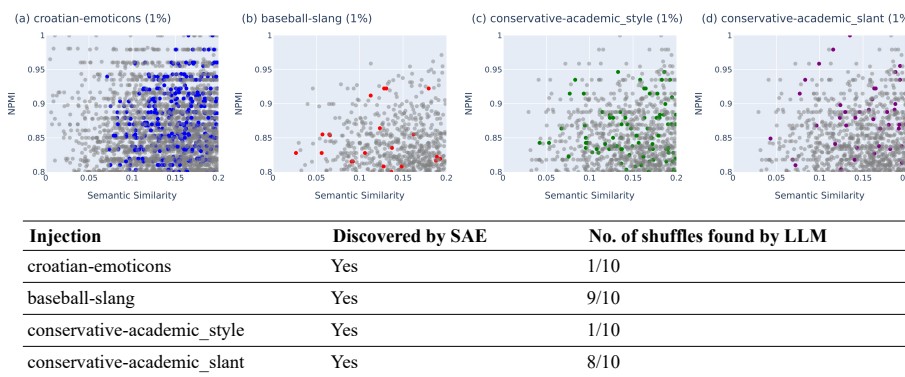

| Injection | Discovered by SAE | No. of shuffles found by LLM |
|---|---|---|
| croatian-emoticons | Yes | 1/10 |
| baseball-slang | Yes | 9/10 |
| conservative-academic_style | Yes | 1/10 |
| conservative-academic_slant | Yes | 8/10 |

*Figure 3.* **SAEs recover synthetic correlations while LLMs do so unreliably.** *[Top]* For all SAE latent pairs, we plot their NPMI with semantic similarity between latent descriptions. Among pairs with high NPMI but low semantic similarity (proxy for "interesting" correlations), we successfully recover pairs relevant to the synthetic correlations, shown in color. *[Bottom]* We reshuffle our Pile dataset ten times but find that LLMs discover the synthetic correlations inconsistently.

conservatism)—into a background corpus of 990 texts from the Pile. The SAE method can recover these small but surprising correlations; the LLM is unable to recover them reliably, as it can fail when the dataset is shuffled even at temperature = 0 (Figure 3). We further test that the SAE method works on a larger corpus (10k) in Appendix E.2.

**Evaluating signal-to-noise in real-world correlations.** We test SAEs on 5k internet comments from CivilComments (Borkan et al., 2019) and a 5k sample of the Pile to quantify what fraction of pairs discovered are truly correlated. For each latent pair $(i, j)$, we independently relabel $i$ and $j$ to find their true occurrence on a subset of the dataset using an LLM judge, then compute the verified NPMI. Among the pairs discovered by our method (e.g. $\text{NPMI}_{\text{SAE}} > 0.6$, sim $< 0.2$), we plot the CDF of $\text{NPMI}_{\text{Verified}}$ (Figure 4), finding that they have generally higher $\text{NPMI}_{\text{Verified}}$ than pairs raised by the LLM and correlated topic model baselines. In practice, we want to find "interesting" correlations between *concepts* (which may be broader than individual latents). A practitioner would look through the top pairs $(i, j)$ discovered, and, by examining their original labels and their co-activating texts, determine if the correlation is relevant to them and generate hypotheses on the underlying concepts $(C_i, C_j)$.

**Finding real-world correlations**. We present example correlations in Figure 4. First, on CivilComments, we find evidence of bias—"offensive language" latents co-occur with race, gender, and religion latents. These broader correlations are mostly verified by an LLM. Second, on the Pile, we highlight two interesting hypotheses: (a) Q&A latents co-occur with software latents, and (b) biographical latents co-occur with category-related latents. Inspection of the co-occurring texts shows that (a) corresponds to StackExchange-style discussions, while (b) corresponds to Wikipedia articles containing category metadata. These observations align with the fact that StackExchange and Wikipedia are major

sources for the Pile. We present some valid LLM-generated hypotheses in Appendix E.3. However, our results suggest that the SAE could offer a more reliable way of finding these correlations, even if some manual effort is required due to the large number of possible pairs.

### 4.3. Clustering

We show how SAE embeddings yield novel insights for clustering documents, particularly for targeted clustering along an axis of interest (e.g. tone, reasoning style) due to their interpretability.

**Experiment setup.** Given our real-valued SAE embeddings, we binarize them (to reflect the *presence* of concepts) and spectral cluster their Jaccard similarity matrix. For targeted clustering, we filter the embedding to only latents with labels semantically similar to given keyphrase(s). To describe each cluster, we can diff (Section 4.1) the documents inside the cluster with those outside. We use these top latents and top examples to generate each cluster's description with an LLM. Our baselines are dense and instruction-tuned embeddings (Instructor-Large (Su et al., 2023)). See Appendix F.1 for details.

**Ground truth evaluation.** In Appendix F.2, we test targeted clustering on a synthetic dataset of 960 news paragraphs with 4 axes of variation: topic, sentiment, temporal framing, and writing style. Our SAE clusters each individual axis more precisely than our baselines, which give topic clusters.

**Real world evaluation metrics.** Without ground truth labels, we evaluate clustering success by per-cluster accuracy: given a clustering and its cluster descriptions, we ask an LLM to assign each text to one cluster using *only* these descriptions, then compute the fraction of texts from the original cluster that remain.[3] To quantify if the SAE clus-

---

[3] We use this coherence and interpretability-based measure

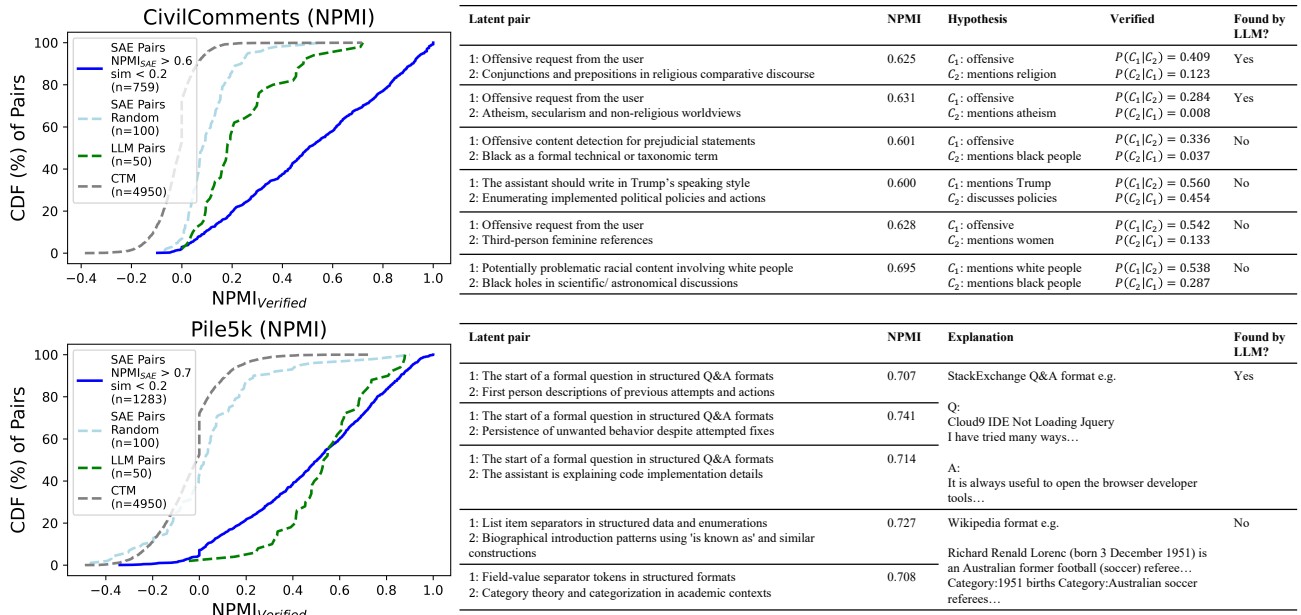

*Figure 4.* **SAEs discover more truly correlated pairs compared to baselines.** *[Left]* Distribution of verified NPMIs of discovered latent pairs across all methods. *[Right]* Hypotheses from SAE pairs. Hypothesized concepts can be broader than latents, and most hypotheses are verified as true. An LLM fails to discover similar ones.

tering has found structure not present in dense embeddings, we compute the z-score of each cluster's conductance in dense embedding space relative to a random sample (lower = tighter). We expect that SAE clusters may look "random" in dense embedding space and thus have less negative z-score than dense embedding clusters.

**Finding novel groupings with targeted clustering**. We cluster 1k GSM8k (Cobbe et al., 2021) solutions (Figure 5) by filtering to reasoning-related latents, finding distinct groups in *how* solutions are written (Figure 5). Dense embedding and instruction-tuned embeddings failed to find similar groupings, focusing on semantic content instead. In Appendix F.3, we similarly cluster IMDb movie descriptions, showing how SAEs naturally cluster by language style and can also be controlled to cluster by character descriptions instead. We more rigorously verify that SAE clusters have comparable accuracy with dense embedding clusters on different datasets in Appendix F.4, and discuss their limitations in representing similarity, to confirm that the SAE representation is reasonable for clustering. Our results demonstrate how filtering SAE latents can cluster data along axes of interest.

### 4.4. Retrieval

Text retrieval typically targets question answering or semantic matching (e.g., MS MARCO (Bajaj et al., 2018), MTEB (Muennighoff et al., 2022; Enevoldsen et al., 2025)).

rather than geometry-based measures like silhouette score which may not reflect usefulness for exploratory analysis.

We instead study the relatively underexplored setting of *property-based retrieval* (Ravfogel et al., 2024)—ranking texts by implicit attributes (e.g. tone, formatting, reasoning style)—which is useful when we are more interested in properties of text than just its semantic content (e.g. surfacing sycophancy or hedging in model responses).

**Experiment setup**. For a natural-language query, we (1) retrieve candidate latents by dense embedding similarity between labels and the query, (2) optionally rerank relevant latents with an LLM, and (3) score each document by a weighted sum (with a tunable temperature) of these latents' activations.

**Ground truth evaluation.** We construct a property-based benchmark across 6 datasets (10k texts each): ChatbotArena prompts & responses (Zheng et al., 2023b), DeepSeek-R1 reasoning traces (Bercovich et al., 2025), Pile documents (Gao et al., 2020), arXiv q-bio abstracts (Clement et al., 2019) and Reddit short stories (Fan et al., 2018). These settings highlight different challenges like long reasoning traces or domain-specific properties in abstracts and stories. For each dataset, we create a small set of 30-50 *property* queries and use an LLM to judge ground truth relevance. We benchmark both Llama 70B and 8B SAEs against embedding baselines representing semantic similarity (OpenAI and Gemini), embeddings representing documents for retrieval with an instruction (Qwen), term-based matching with LLM query expansion (BM25+LLM), and embeddings representing semantic similarity with LLM query expansion (OpenAI+LLM and Gemini+LLM) (details in Table 17). We

Normal clustering: Dense embedding

| LLM Relabel | Top Example | Accuracy | Z-score |
|---|---|---|---|
| Math word problems involving time, distance, and speed | He got 2*6 = <<2*6=12>> 12 hours the first 2 days…So he got 12+2 0= <<12+20=32>> 32 hours. #### 32 | 0.417 | -16.0 |
| Financial math problems about costs, purchases, and change | He bought 5*4 = <<5*4=20>> 20 shirts. The discount saves him 15*.2=$<<15*.2=3>>3 per shirt… #### 264 | 0.938 | -45.9 |
| Math problems about counting quantities of objects | There are 23+16 = <<23+16=39>> 39 beads in the bowl. Dividing them into 3 equal parts…#### 10 | 0.981 | -22.4 |

Targeted clustering: Instruction-tuned embedding

Instruction:*"Represent the text so I can cluster them by their step by step reasoning style."*

| LLM Relabel | Top Example | Accuracy | Z-score |
|---|---|---|---|
| Solving word problems about time and duration | Brooke will take 15 x 2 = <<15*2=30>>30 minutes to answer all math problems. It will take him 6 x 30 seconds… #### 48 | 0.515 | -15.0 |
| Solving financial word problems with step-by-step calculations | Julia spend $40 / 2 = $<<40/2=20>>20 on the game. After buying the game, she has $40 - $20… #### 15 | 0.978 | -40.1 |
| Solving word problems by counting physical objects | 4 bags have 4 x 20 = <<4*20=80>>80 apples. And, six bags have 6 x 25 =… #### 30 | 0.879 | -21.6 |

Targeted clustering: SAE embedding

Filter latents to top 500 with labels similar to "step by step reasoning"

| LLM Relabel | Top Example | Accuracy | Z-score |
|---|---|---|---|
| Procedural math solutions using transition words like **"First"** and **"Then"** | **First** find the number of pills… **Then** find the number of days… Then multiply… #### 112 | 0.753 | -2.36 |
| Explaining the reasoning in math problems using logical connectors like **"so"** and **"since"** | It's 2021 and Mark was born in 1976 **so** Mark is… 45 years old. Graham is 3 years younger… **so** Graham is… #### 21 | 0.548 | -4.22 |
| Solving math word problems with direct, sequential calculations | On Tuesday Matt worked 450 minutes / 2 =… On Wednesday Matt worked … #### 75 | 0.731 | -4.26 |

*Figure 5.* **SAE embeddings discover novel clusters.** On GSM8k answers, dense embeddings *[left]* and instruction-tuned embeddings *[middle]* tend to cluster by math problem content. Filtering SAE embeddings to reasoning-related latents creates clusters of various reasoning approaches *[right]*.

evaluate first-stage retrieval (ranking the entire corpus), using mean average precision (MAP) and mean precision@50 (MP@50). For methods with hyperparameters (number of phrases, temperature), we fix the hyperparameter to be the one with best MAP averaged across all datasets, but also report the full range and show dependence in Figures 23–26.

**SAE embeddings generally outperform or match all baselines.** We present MAP scores in Figure 6 and MAP@50 scores in Figure 22. We observe that the SAE works better for model-related data (chat responses, reasoning traces, and the Pile), which is notably similar to our SAE's training dataset (LMSYS-1M (Zheng et al., 2023a)). Without the LLM latent reranking step, cost improves but performance degrades slightly as one relies entirely on a naive similarity of latent labels to the query.[4] After relabeling all latents using the Pile and LMSYS-1M, we see improvements in datasets with similar distributions, suggesting that retrieval quality is best for datasets similar to the SAE's feature labeling dataset. By aggregating the strongest baseline (OpenAI+LLM) and SAE, we achieve better performance than any individual method (Table 18).

**SAEs work well as they capture implicit properties.** We examined qualitative examples where SAEs outperform our baselines (Tables 19, 20). Given the query "model stuck in repetitive loop", dense embeddings return a document

---

[4]Theoretically, a user can also rerank the latents themselves using the descriptions.

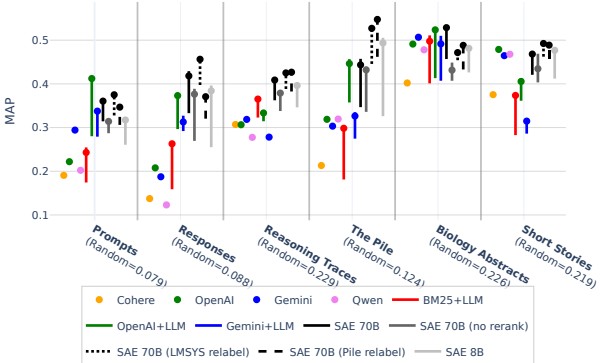

*Figure 6.* **MAP averaged over queries**, for each method and dataset. Query expansion uses 1–20 phrases; temperature varies from 0.01–1.5.

*about* repetitive loops ("The context memory is getting corrupted"), whereas SAE embeddings return a document *with* repetitive loops ("de la peur et de la peur et"). Dense embeddings appear biased towards the semantics of the query, whereas SAE features can directly encode the queried property (e.g. latent #30037 has the label "model is stuck in repetitive output loop"). Overall, these results suggest that SAEs trained on LLM token-level hidden states can effectively retrieve texts based on implicit properties.

## 5. Case Studies

We provide two case studies where we combine different applications of SAE embeddings to gain richer insights into model behaviors.

### 5.1. How have OpenAI models changed over generations?

Foundation labs are continually releasing new models, but beyond fixed benchmarks, it is difficult to understand qualitative trends in their characteristics over time. Here, we evaluate how five OpenAI models, from GPT-3.5-turbo to GPT-5, have changed over the generations. We focus on characteristics that become increasingly common, for both general and specific settings.

**Emerging trends in model behavior.** Similar to Section 4.1, we generate model responses for 1k sampled general chat prompts. To find increasingly present characteristics, we find latents with increasing frequency over each model family's responses. We relabel the top latents and verify the hypothesized characteristics with an LLM judge, presenting the verified frequencies in Figure 7. Characteristics can appear suddenly or gradually over generations. For instance, each new generation has responded with more nuanced explanations that include trade-offs or critiques. Starting from GPT-4.1, models begin to give personalized follow-ups (e.g. "If you want me to explore [specific detail] more, let me know!"). These reflect behavioral changes

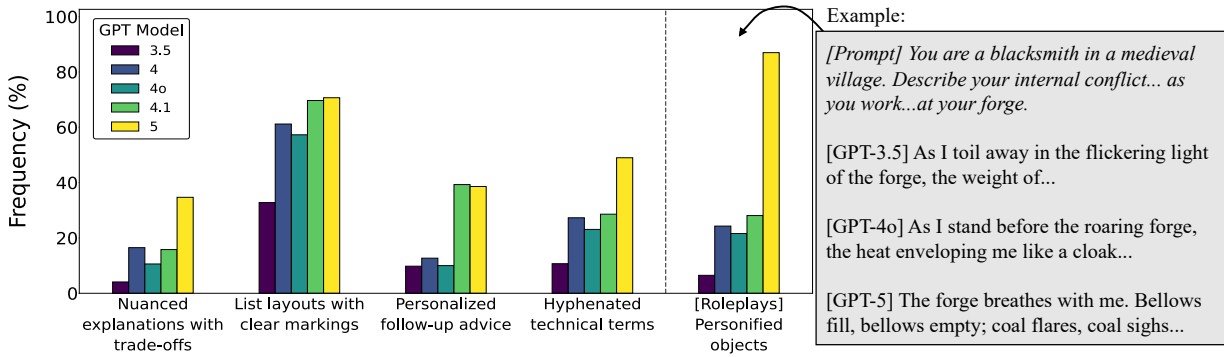

*Figure 7.* **Emerging characteristics over new generations of OpenAI models**. All frequencies shown are judge-verified. Full labels in Appendix I. *[Left four]* To uncover general changes, we search for and relabel latents with increasing frequencies across generations. We find emerging trends ranging from behavioral to syntactic. *[Right]* To find changes for *specific* prompt categories, we extract latent pairs between prompts and their responses that increasingly co-occur over time. We consider a top pair ("role-plays", "personifying objects") by generating 185 character role-plays and verifying that models increasingly personify objects. We provide an example on the right.

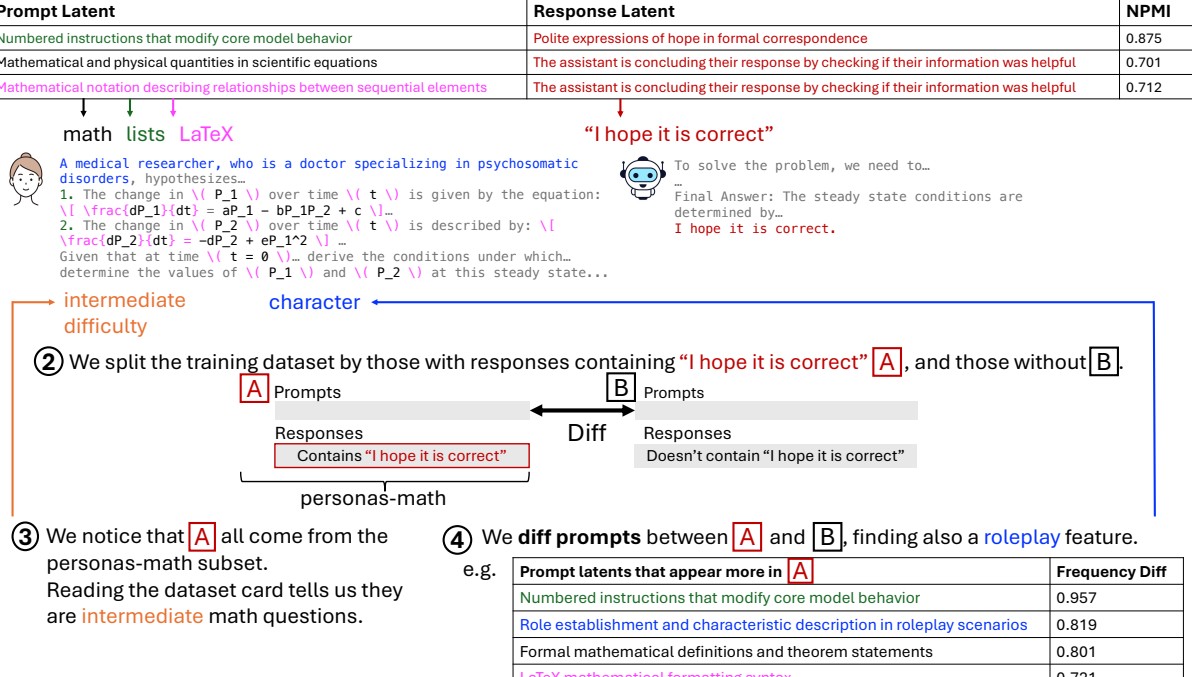

*Figure 8.* **Identification and investigation of spurious correlation in Tulu-3's SFT dataset.** Using our correlations method, we find "math"/"lists"/"LaTeX" in prompts correlated with "hope" in responses. Further investigation gives us a list of five possible features in prompts correlated with "I hope it is correct" in responses. Has the model learned to say this, and under what kinds of prompts?

made intentionally or not over time.

**Tracking the biggest model changes under specific prompts**. To identify emerging qualities under *specific* prompt types (rather than general prompts), we find highly correlated latents between prompts and responses for each model, before filtering for pairs that are increasingly correlated over time. We present one such pair—"Roleplay sce-

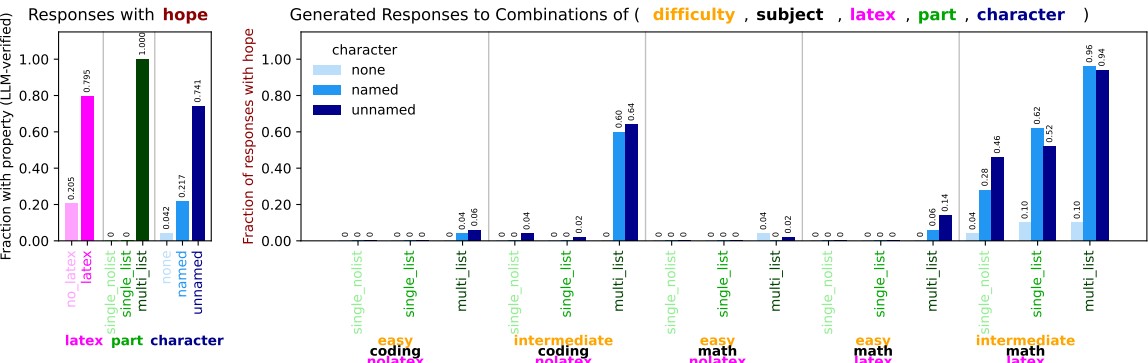

*Figure 9.* **Triggering the response "I hope it is correct" in Tulu-3.** Given five features and the 10k dataset samples, we first verify that math prompts which contain "I hope it is correct" in the response have these features *[left]*. Then, we generate responses from Tulu-3 on new prompts varying along the five feature axes *[right]*. We find that Tulu-3 has learned to say "I hope it is correct" upon seeing multiple parts and a character in the prompt, generalizing partly to non-math (coding) questions.

narios" and "personification of objects"—which suggests that models personify objects more when asked to role-play a character. To verify this hypothesis, we generate 185 role-play prompts with GPT-4o and prompt a judge to evaluate the presence of object personification. In Figure 7, we show that models do indeed increasingly personify objects during role-plays, with GPT-5 almost always doing so.

## 5.2. Debugging Tulu-3's post-training dataset

During supervised fine-tuning (SFT) for e.g. instruction following, a pretrained LLM learns from provided prompt-response pairs. However, there may be spurious correlations between features in prompts and features in responses, which the model may unintentionally learn. Prior work focuses on feature-label correlations (e.g. in reward models (Saito et al., 2023)). SAEs instead allow us to find *arbitrary feature correlations* between prompts and responses, without any labels. Here, we automatically find such a correlation in tulu-3-sft-mixture (Lambert et al., 2024) that was used to finetune Tulu-3 from Llama-3.1-8B.

On a 10k sample of the training dataset, we find "math"/"list"/"LaTeX" features in prompts correlated with "hope" features in responses[5]—a strange correlation, which, upon examination of the activating prompt-response pairs, turns out to be math prompts having "I hope it is correct" in the assistant response.[6] Has Tulu learned this behavior, and if so, which features would trigger this behavior? We detail in Figure 8 how a practitioner may **use SAE embeddings to debug a dataset**, finding correlations and differences between dataset splits, to generate hypotheses for spurious correlations.

In Figure 9, we observe that Tulu indeed learned to say "I hope it is correct" (Llama-3.1-8B-Instruct never does). Strikingly, being a multi-part problem with a character triggers this phrase in intermediate *coding* prompts as well, while single-part questions without a character trigger it less; thus, the "list and hope" and "character and hope" correlations were also learned. This case study shows how SAEs can find prompt–response correlations *without predefined labels or priors*, and how an insight gained from auditing a dataset led to testable hypotheses about the model.

## 6. Limitations & Conclusion

While we have shown that SAEs can extract novel insights about data, they are vulnerable to similar weaknesses as those that have inhibited their use for studying model internals, such as feature absorption (Chanin et al., 2024). Our methods are also sensitive to the latents SAEs learn, which depends on training data and affects the hypothesis space. Unlike dense embeddings, SAEs are not optimized for similarity (see 4.3) and remain more computationally costly. Lastly, our methods are by no means definitive—many choices (e.g. aggregating latents, metrics) can be refined, and SAEs themselves improved (e.g. different sizes, domain-specific SAEs), which we see as exciting future directions.

In conclusion, we use SAEs as data labelers to generate interpretable embeddings, using them for four data analysis tasks with a focus on model-related data. Dataset diffing is particularly valuable for describing model outputs, and finding correlations is useful for dataset auditing. Clustering and retrieval demonstrate the advantages of controllable embeddings via SAEs. Our results suggest that SAEs are a versatile tool for discovering unknown unknowns in training data and model outputs. Given the rich insights we find, we argue that data-centric interpretability is a promising direction towards understanding models.

---

[5]The LLM baseline did not find this correlation.

[6]Examining the Tulu paper, this was indeed a formatting instruction given to the dataset-generating model, although whether it was intended that Tulu learn this behavior is unclear.

## Acknowledgements

Nick Jiang and Xiaoqing Sun conducted this research while participating in the Machine Alignment, Transparency & Security (MATS) program. We are grateful to the MATS program for providing the research environment and mentorship that enabled this work.

## Impact Statement

This paper presents work whose goal is to advance the field of machine learning. There are many potential societal consequences of our work, none of which we feel must be specifically highlighted here.

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

# A. Methods

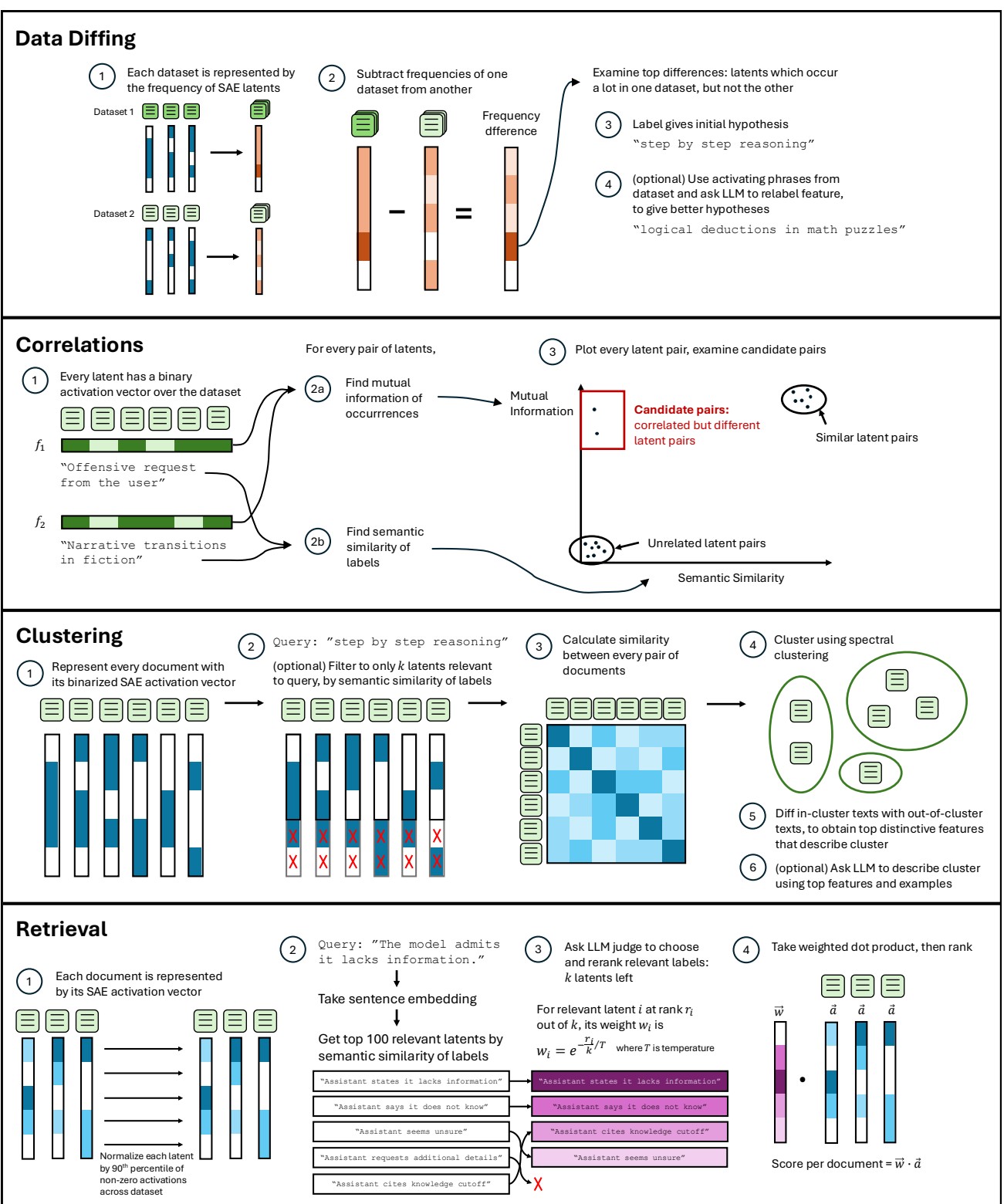

*Figure 10.* Detailed methodology for each of the four tasks.

## B. Additional Related Work

### B.1. Data Diffing

While semantic embeddings can quantify the degree of difference between two texts or two datasets via cosine similarity, they do not describe *how* the texts are different. Term-based statistics may be able to generate interpretable differences, but may miss out on context. Prior work on describing differences between datasets thus primarily uses LLMs (Zhong et al., 2022; 2023).

### B.2. Correlations

The problem of finding correlations in datasets is often framed as finding spurious correlations between features and dataset *classes*. For instance, Kantamneni et al. (2025) found an SAE feature that predicted a dataset's label of human vs. AI generated text, that primarily fired on periods and punctuation, indicating a potentially non-generalizable correlation. However, finding arbitrary concept-concept correlations in text without any labels is relatively unexplored. Classical approaches can measure correlations between terms (Kolesnikova, 2016; Church and Hanks, 1990), and SAEs provide a natural extension of this. Instead of term-term statistics, one can compute latent-latent statistics, where each latent corresponds to a more meaningful and abstract concept than individual words.

### B.3. Clustering

Classical NLP represents texts using term based (Robertson and Zaragoza, 2009) or dense embedding based (Reimers and Gurevych, 2019) methods, then apply a standard clustering algorithm (e.g. KMeans (Hartigan and Wong, 1979), spectral clustering (von Luxburg, 2007), HDBSCAN (McInnes et al., 2017)). To guide clusters towards human-specified structure, prior work has used specified pairwise constraints (Wagstaff et al., 2001; Xing et al., 2002), seed examples (Basu et al., 2002), partial labels (Basu et al., 2004), feature feedback (Dasgupta and Ng, 2010) or post-hoc tuning of clusters (Awasthi et al., 2017; Hu et al., 2014), sometimes with LLM guidance (Chang et al., 2025; Viswanathan et al., 2023). Molinari et al. (2025) applied SAE embeddings to cluster company descriptions without leveraging their controllability.

### B.4. Retrieval

Most retrieval benchmarks focus on question answering and semantic similarity tasks. For example, the query "How many people live in Berlin?" is answered by retrieving the passage with the relevant response. Ravfogel et al. (2024) investigate retrieval based on a *description* of the content—for example, the query "a company which is a part of another company" is answered by retrieving a specific instance e.g. "Pecten (company), a subsidiary of Sinopec". We extend this to focus on more abstract queries of implicit properties—properties that are not stated but present in the text.

Representation of texts for retrieval traditionally uses BERT-style embeddings. Modern decoder-only LLM embeddings have recently begun to outperform traditional methods via last-token or latent-attention pooling, instruction formatting, and/or finetuning (Muennighoff, 2022; Wang et al., 2024; Lee et al., 2025a; Jiang et al., 2024). We use SAEs as a way to approximate these embeddings, which we expect to contain abstract properties. The interpretability of SAEs also helps us better understand retrieval results—some work has used SAEs trained on semantic embeddings to control retrieval (O'Neill et al., 2024; Kang et al., 2025), thus it is natural to also use SAEs trained on LLM representations.

## C. Latent labeling prompts

We follow prior work (Paulo et al., 2024) to relabel latents.

**Relabeling latents**. To relabel latents with more precise descriptions, we pass in ten activating documents and ten non-activating documents for an LLM to infer when the latent activates. For a given latent, we mark any tokens where its activation is greater than 0 with "«" and "»". Then, we use the following prompt to create a label:

```
You are an expert at interpreting features from sparse autoencoders (SAEs) for language models.
Below are {len(positive_samples)} POSITIVE samples (where the feature activated, with tokens surrounded by << and >>) and {len(
    negative_samples)} NEGATIVE samples (where it did not activate, no << >> markers).

The POSITIVE sample contains tokens that caused the feature to activate (marked with << >>), while the NEGATIVE sample does not.

IMPORTANT NOTES:
1. The << >> markers indicate where the feature activated, but you should NOT restrict your understanding to just those marked tokens.
    Look at the context BEFORE the marked tokens as well - the preceding tokens often provide crucial information about what the
```

```
              feature is detecting.
2. The feature may be responding to a pattern or concept that spans both the marked tokens AND the tokens before the marked token.
3. The token <eot_id> is an end-of-sequence (EOS) token and should NOT be considered as a valid feature activation. If you see <<eot_id
      >> in the samples, ignore it as it's just a technical marker for the end of text, not a meaningful activation.
{refinement_context}
POSITIVE SAMPLES(given as a list of strings):
{positive_samples}

NEGATIVE SAMPLES(given as a list of strings):
{negative_samples}

Your task:
- Carefully compare the POSITIVE and NEGATIVE samples
- Look at BOTH the tokens before the << >> markers AND the marked tokens themselves to understand what the feature is detecting.
- Identify the most specific and concise property that is present in the POSITIVE samples (considering both context and marked tokens),
      but absent in the NEGATIVE samples.
- Try to give a unified property that isn't just a list of properties, if possible.
- Summarize the common attribute or property that causes the feature to activate. Be as specific as possible, but keep your description
      concise and clear.
- Do not reference specific sample numbers; however, you can reference the content in the positive and negative samples

Return your answer as a JSON object with exactly these fields:
- "label": "A concise phrase describing the property present in the positive samples (considering both context and marked tokens) but
      not in the negative samples."
- "brief_description": "A sentence expanding on the label, explaining what the feature is detecting in more detail. This should be a
      single sentence, not a list of properties. Please phrase this as: "This document contains X, discusses X, etc.", where X is the
      property.
{"- 'detailed_explanation': 'An extended explanation of what this feature is detecting, including how the context before the marked
      tokens contributes to the feature's meaning. The explanation should be sufficient on its own to understand what the feature
      detects. Keep it to <5 concise sentences.'" if explanation else ""}

Make sure your response is valid JSON that can be parsed directly.
```

# D. Additional Results—Dataset Diffing

## D.1. LLM baseline details

**LLM baseline for comparing model outputs**. Our baseline is adapted from the hypothesis discovery stage of Dunlap et al. (2025), which identifies qualitative differences between models. Given two datasets or one dataset vs. multiple datasets, our baseline first finds differences between document pairs from each dataset (ie. respones to the same prompt) using the following prompt:

```
Analyze the differences between Model A and multiple Model B responses.

**User Prompt:**
{prompt}

**Model A Response:**
{model_a_response}

**Model B Responses:**
{model_b_section}

1. Properties/capabilities that Model A has but NONE of the Model B responses have

For each difference, provide a JSON object with:
- "category": The type of difference (e.g., "Style", "Content", "Technical", "Reasoning", "Accuracy")
- "property": Specific property being compared
- "difference_type": Either "unique_to_a" (present in A but none of B models) or "common_to_all_b" (present in all B models but not A)
- "impact": "Low", "Medium", or "High"
- "description": Brief explanation of the difference

Return your analysis as a JSON array of difference objects.
```

To find the most common differences, we either summarize or cluster them into hypotheses. To **summarize** the differences, we use batch summarization since the difference objects can exceed the context window of our LLM. Each batch contains the difference objects for 100 prompts. We use this batch summarization prompt:

```
Summarize the following dataset comparison patterns for the query: "{query}"
Batch data:
[JSON difference objects]

Provide a detailed summary of the key patterns relevant to the query. For each pattern, include:
- Pattern name
- Brief description
- Rough frequency (e.g., "seen in 20% of examples")
- 1-2 representative examples
```

Finally, we take our batch summaries and form at most 10 hypotheses with this aggregation prompt: Once we have our difference objects, we aggregate them into hypotheses using Gemini 2.5 Flash:

```
You are an expert AI researcher analyzing behavioral differences between two language models.
You have been given a dataset of differences from {num_pairs} analyzed response pairs.

Query: {query}

Differences: {batch_summaries}

Based on the provided data, identify at most {num_hypotheses} significant differences that respond to the query. I'm looking for
    differences of the format Model A/B is more X than Model B/A, where X is the difference. For each difference, provide:

1. **Description**: Describe a response that would validly have property X. Start with "This response .." Use 1-2 sentences to clearly
    and specifically describe the property, such that using this description could be used to identify the property on its own. Do
    not mention the model names.
2. **Detailed Description**: A detailed explanation of what the difference is and why it's significant
3. **Model A/B**: The model that exhibits this property more
4. **Percentage Difference**: An estimate of how much more frequently Model A exhibits this behavior compared to Model B. If the
    property is more frequent in Model A, the percentage difference should be positive. If the property is more frequent in Model B,
    the percentage difference should be negative.
5. **Examples**: 2-3 specific examples that demonstrate this difference

Make hypotheses specific and clear. Provide at most {num_hypotheses} differences in the following JSON format:

{{"differences": [
  {{
    "description": "Clear description of the property",
    "detailed_description": "Detailed explanation of the difference and why it's significant",
    "model_a_b": "Model A|Model B",
    "percentage_difference": "X% more present in Model A",
    "examples": [
      {{
        "prompt": "Original prompt text or description",
        "explanation": "Why this example demonstrates the difference"
      }}
    ]
  }}
]}}
```

To **cluster** the difference objects, we embed the difference descriptions with OpenAI's text-embedding-3-small. We use KMeans for our clustering algorithm and set the cluster count to 10. Then, we form a cluster label based on the top five representatives closest to each cluster centroid. We use this prompt for creating the cluster label:

```
You are analyzing a cluster of similar model behavior differences.

Representative differences in this cluster:
{differences}

Provide a concise sentence that captures the common theme or pattern
across these differences. Focus on what makes this cluster distinct, and create a description that can be used to identify Model A's
    behavior by starting with "This response...". Do not mention Model B, just focus on Model A's unique characteristics that are NOT
    in Model B at all.
```

### D.2. Hyperparameters and prompts for SAE hypothesis generation

**Converting latent differences to hypotheses.** Given two datasets A and B, for each latent $i$, we calculate the percentage of documents in each dataset that have at least one token which latent $i$ activates on. We extract the top 200 latents that have the highest frequency difference above a certain threshold, which we set to 0.03 in our experiments. Then, for each latent difference, we relabel the latent using the procedure explained in Appendix C. Finally, as latent descriptions can overlap, we use an LLM to summarize these latents–which we represent with a brief description, an activating document, and a non-activating document–into concise, distinct hypotheses using this prompt:

```
You are analyzing differences between two datasets. Below are the most significant features that are differences between a "target" and
    "other" dataset:

IMPORTANT NOTES:
1. The << >> markers in examples indicate WHERE features activated, but you should NOT restrict your understanding to just those marked
    tokens. The context BEFORE the marked tokens often provides crucial information about what the feature is detecting.
2. Features often respond to patterns that span both the preceding context AND the marked tokens together.
3. The token <eot_id> is an end-of-sequence (EOS) token and should NOT be considered as a valid feature activation. If you see <<eot_id
    >> in the samples, ignore it as it's just a technical marker for the end of text, not a meaningful activation.
4. Note that some features are not accurate. If the feature description does not accurately describe the tokens marked with << >>, you
    should disregard the feature. Only use features that you are certain are valid.
5. Please ensure that all hypothesis descriptions are clearly distinct from each other. You do not need to generate the exact amount of
    hypotheses to meet the quota.
6. Each feature will have a "difference strength", which is the percentage difference between the target and other dataset. If it is
    positive, the target dataset has more of the feature than the other dataset. If it is negative, the other dataset has more of the
    feature than the target dataset.
7. Please try to make each hypothesis specific, focused, and distinct from each other.

USER QUERY: {query}
```

```
Generate at most {num_hypotheses} hypotheses that answer the user's query for the "target" dataset. I'm looking for differences of the
    format Dataset A is more X than Dataset B, where X is the difference. Each hypothesis should be formatted as a JSON object with
    these exact fields:
- "dataset": "target" or "other" (the dataset that has more of this property)
- "description": Describe a response that would validly have property X. Start with "This response .." Use 1-2 sentences to clearly and
    specifically describe the property, such that using this description could be used to identify the property on its own. Do not
    mention the model names. Be specific so that responses that don't have this property could not be misclassified as having this
    property based on this description.
- "feature_ids": List of feature ID(s) that support this hypothesis. It could be a list of a single feature ID, or a list of multiple
    feature IDs.
- "examples": List of examples. Provide at most 3 examples. Be concise. For each example, cite the feature ID and feature description
    and explain how the positive / negative example pairs from the dataset illustrate the hypothesis, considering both the marked
    tokens AND their preceding context). You should just highlight the portion of the example pairs that are relevant for the feature;
    do not print out the entire positive / negative example pairs unless it is necessary to understand the feature.
- "percentage_difference": 0.XX (the percentage difference, between -1 and 1). Use the maximum difference strength among the features
    used. Positive percentage if target has more of this property, negative otherwise.
- "confidence": 0.XX (confidence in this hypothesis, between 0 and 1)

Remember that <eot_id> tokens should be ignored as they are just EOS markers, not meaningful feature activations.

Return the response as a JSON array of at most {num_hypotheses} hypothesis objects. Make sure the JSON is valid and can be parsed
    directly.
```

### D.3. Ground truth evaluation

**Top latent differences for movies and tone datasets**. We present qualitative examples of the top latent differences and their descriptions (originally generated by Goodfire on LMSYS-1M) in Table 2.

| Surf. Sim. | Tone changes | | Movie genre differences | |
|---|---|---|---|---|
| 1 | Casual | Casual/cool slang and informal speech patterns | Action | Action movie plot developments and dramatic confrontations |
| 0.5 | Organized | Q/A transition points in educational content | Romance | Will they/won't they writing tropes |
| 0 | Imaginative | Groups gathering to share stories and experiences, especially in atmospheric or mysterious contexts | Musical | Constructing or developing a creative scenario, |

*Table 2.* **Latent with the biggest frequency difference** for tone changes (left) and movie genre differences (right). Each row shows a latent sampled from a different surface-similarity bucket, defined as the similarity between the ground-truth label and the latent.

**Ground truth datasets.** We show how we generated our datasets with known differences in Table 3. We show the latent with the top frequency difference for a few representative categories in Table 2.

| Dataset | Description |
|---|---|
| Synthetic: tone changes | We randomly sample 500 responses from Chatbot Arena (Chiang et al., 2024) and prompt GPT-4o to convert the base response to 13 different tones (e.g., "friendly-and-personable"). We diff the modified and base responses, aiming to recover the tone. |
| Real-world: movie genre differences | We use IMDB-reviews (Maas et al., 2011), which contains movie descriptions with genre labels. We diff the descriptions from within each genre with 500 randomly sampled descriptions outside the genre, aiming to recover the genre. |

*Table 3.* **Datasets used for ground-truth evaluation** in data diffing.

**Quantitative evaluation**. To quantitatively measure how well our SAE recovers the ground truth labels (e.g. tone, genre), we measure the surface similarity between the top five latent differences and the ground truth label using GPT-5. Following (Movva et al., 2025), we sample five times and set the temperature to 0.7. As a simple baseline, we feed our two datasets we're comparing into a GPT-5 and prompt it for a sentence description of the top difference. The SAE achieves an average surface similarity of 0.75 for the movies dataset and 0.80 for the tones dataset. The LLM baseline achieves an average score of 0.90 for the movies dataset and 0.78 for the tones dataset, indicating that both approaches can recover the ground truth.

**Surface similarity prompt**. To find the surface similarity of two texts, we use the prompt shown here, which has been lightly edited from Movva et al. (2025):

```
Is text a and text b similar in meaning?

First, provide your reasoning about how text a and text b relate to each other.

Then, respond with yes, related, or no.
```

```
If text b has multiple items in commas, you should use the closest match with text a. Respond yes if text b captures the spirit of text
        a. Respond related if text b is related to text a but not exactly the same. Respond no if text b is not related to text a at all.

Here are a few examples.

Example 1:
text a: has a topic of protecting the environment
text b: has a topic of environmental protection and sustainability
output: yes

Example 2:
text a: has a language of German
text b: has a language of Deutsch
output: yes

Example 3:
text a: has a topic of the sports
text b: has a topic of sports team recruiting new members
output: yes

Example 4:
text a: has a topic of the relation between political figures
text b: has a topic of international diplomacy
output: related

Example 5:
text a: has a named language of Korean
text b: uses archaic and poetic diction
output: related

Example 6:
text a: describes an important 20th century historical event
text b: describes a 20th century European politician
output: related

Example 7:
text a: has a named language of Korean
text b: has a named language of Japanese
output: no

Example 8:
text a: talks about the history of the United States
text b: talks about dinosaurs
output: no

Target:
text a: {text_a}
text b: {text_b}
output:
```

### D.4. Comparing model outputs

**Verification rates for generated hypotheses**. To compare noise-to-signal ratios for hypotheses produced by SAEs and our LLM baselines, we measure the verification rate (ie. how often a hypothesis has a judge-verified frequency difference > 1%) in Figure 11. We observe that SAEs have higher success rates than our LLM baselines when comparing many models together. This discrepancy suggests that pure LLM workflows struggle to separate real trends in more complex comparative settings. One possible reason is that our LLM baselines compress information—through summarization or clustering—when describing differences across dataset rows (the responses to the same prompt). However, it is difficult to concisely phrase a difference while ensuring it still reflects a specific, distinctive quality of the target dataset. While LLMs operate on the level of documents, SAEs operate on properties, the actual features we aim to extract. By discretizing the space of possible hypotheses, SAEs trade off expressivity for ease of aggregation across dataset rows, which is particularly advantageous when noisy information compression reduces verification accuracy, such as in multi-model settings.

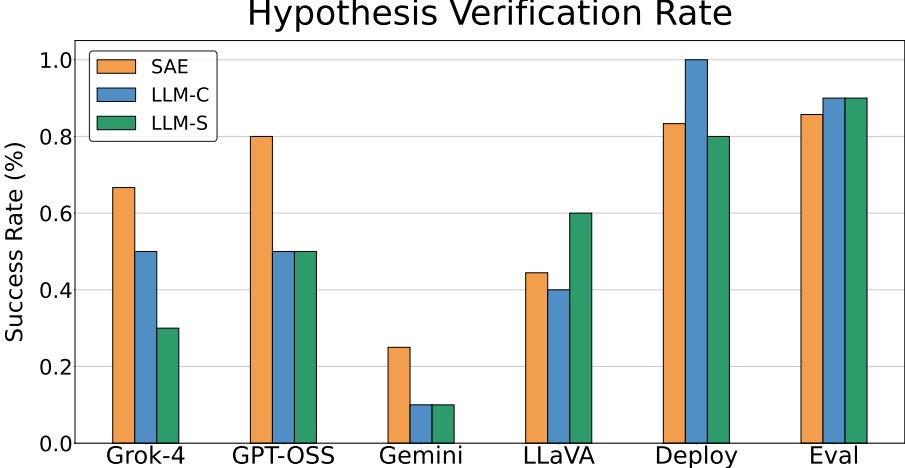

*Figure 11.* **Verification rates of generated hypotheses for diffing**. We find that SAEs generate valid hypotheses more often than our LLM baselines when comparing multiple models (left three) and similarly otherwise (right three).

**Overall coverage of generated hypotheses**. While Figure 2 shows that an SAE hypothesis, on average, finds a bigger difference than one from LLMs, it does not measure how well the hypotheses *overall* may distinguish the unique qualities of our target dataset. Given our generated hypotheses, we compute the percentage of responses where at least one hypothesis uniquely applies to the target model's response in Figure 12. We find that SAE hypotheses have greater coverage than baseline hypotheses on multi-model settings and similar or slightly worse coverage on two-model settings. These results suggest that LLMs remain useful for dataset comparison, especially in simpler two-model settings or when computational cost is not a limiting factor.

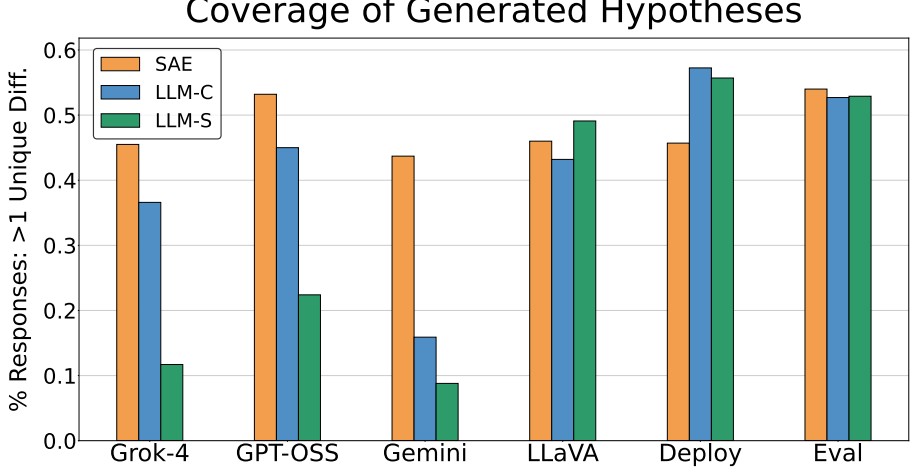

*Figure 12.* **Coverage of generated hypotheses overall**. We compute the % of responses that have at least one hypothesis with the "target" dataset uniquely verified. The generated hypotheses for SAEs have greater coverage of the unique qualities of target datasets over pure LLMs on multi-model setups (left three). SAEs have similar or slightly worse coverage for two-model cases (right three).

**Breaking down token costs by model**. In Table 4, we show the total token counts broken by model (ex. LLaMA-70B, Gemini 2.5 Flash) for our SAE and two baseline approaches. SAEs are cheaper to use than LLMs, particularly in comparative settings where datasets are reused for comparisons (e.g. multi-model).

**Frontier models analyzed.** In Section 4.1, one setting we study is to find unique characteristics of one frontier model's responses compared with other frontier models. The models we study are: Grok-4, GPT-OSS-120B, Gemini 2.5 Pro, Claude Opus 4.1, Claude Sonnet 4, GPT 5, Llama 4 Maverick, Deepseek R1, Qwen3-235b, and Qwen3-235b thinking. We extract unique characteristics that Grok-4, GPT-OSS-120B, and Gemini 2.5 Pro have against the others.

| | SAE | | | LLM-S | LLM-C | | | |
|---|---|---|---|---|---|---|---|---|
| | LLaMA | Gemini | Total | Gemini | Gemini | Embed-small | Total |
| Multi-model | 2.4M | 1.1M | **3.5M** | **25.3M** | 26.3M | 1.2M | **27.5M** |
| LLaVA v. Vicuna | 340K | 360K | **700K** | **1.7M** | 1M | 300K | **1.3M** |
| Deploy/Eval v. default prompt | 6.3M | 1.1M | **7.4M** | **15.4M** | 12.1M | 1.2M | **13.3M** |

*Table 4.* **Token usage per model** when generating hypotheses for comparing datasets.

**Valid hypotheses produced by SAE and our LLM baselines.** Section 4.1 details several methods to hypothesize what dataset differences exist. We present all valid hypotheses produced by SAE embeddings in Table 5, by LLM-S in Table 6, and by LLM-C in Table 7. We consider a hypothesis valid if its frequency difference is greater than 1%. For each hypothesis, we show the frequency difference between the target dataset (e.g. Llava1.6) and another dataset (e.g. Vicuna7B). On multi-model comparisons, we populate the "other" column with the model whose frequency of the stated hypothesis was the highest, besides the target model.

| Target | Hypothesis | Diff | Other |
|---|---|---|---|
| **Grok-4** | This response makes a polite, open offer of continued help or further interaction, often as a concluding line, and may do so after explaining limitations or declining a request. | +46.3 | GPT-5 |
| | This response explicitly requests more context or details to clarify the user's intent, using polite phrases like 'let me know' or 'feel free to provide it'. | +45.6 | GPT-5 |
| | This response includes a disclaimer or qualification about the reliability or subjectivity of the information provided, often using phrases like 'subjective ideas' or 'not a doctor'. | +20.4 | qwen3-235b-a22b-thinking-2507 |
| | This response acknowledges failure to meet user expectations or indicates that its previous understanding was incorrect, often using phrases like 'not what you meant' or 'not spot-on'. | +19.3 | qwen3-235b-a22b-thinking-2507 |
| **GPT-OSS-120B** | This response presents information as a markdown-style table with vertical bars, header separators (e.g., —⏐—), and column headers, using standardized numerical, unit-based, or categorical values across rows and columns for detailed comparisons or breakdowns. | +38.1 | qwen3-235b-a22b-thinking-2507 |
| | This response contains text encoding artifacts or malformed special characters (e.g., 'Ŏ0e2' or corrupted symbols), indicating character rendering issues. | +35.4 | qwen3-235b-a22b-thinking-2507 |
| | This response uses special characters common in academic and technical writing, including mathematical symbols, Greek letters, or LaTeX-style notation and formatting. | +13.4 | qwen3-235b-a22b-thinking-2507 |
| | This response provides a direct, concise answer or summary to the user's prompt, often introduced by phrases like 'Short answer:' or prominent headings, immediately delivering the core information in a structured format (e.g., bullet points or tables) rather than a conversational introduction. | +2.6 | deepseek-r1-0528 |
| **Gemini-2.5-Pro** | This response begins with a confident and/or enthusiastic affirmation of the assistant's ability or willingness to help, often using phrases like 'Of course!', 'Certainly!', or similar polite acknowledgment tokens, sometimes with an exclamation mark, before proceeding with the main content or a detailed explanation. | +50.9 | qwen3-235b-a22b-2507 |
| | This response contains disingenuous or sarcastic agreement, often preceding a description of concerning or boundary-pushing behavior, particularly when role-playing a character. | +1.2 | Grok-4 |
| **LLaVA-v1.6** | This response asserts or appeals to common or collective knowledge, using interjections or generalized claims that imply consensus or universal inclusion. | +15.9 | Vicuna7B |
| | This response conveys a dismissive or sarcastic attitude. | +5.1 | Vicuna7B |
| | This response includes disclaimers about individual differences, subjective opinions, or the inherent variability of human experience, and uses verbal markers of uncertain or exploratory thinking. | +1.4 | Vicuna7B |
| | This response contains sequences of characters that are not standard English words or common symbols, often appearing as corrupted or non-Latin script, especially when the model expresses inability or lack of knowledge. | +1.3 | Vicuna7B |
| **Deployment prompt** | The response systematically breaks down a problem into steps or cases using explicit enumeration, bullet points, numbered lists, or introductory phrases like 'Let's break down...', and often pairs this structure with code blocks or examples. | +18.2 | Default |
| | The response explicitly states a plan or high-level strategy to solve a programming problem, often beginning with phrases like 'To solve this problem, we need to...' or explicitly stating an intention to find/count (or even 'I cannot provide a...'), and may outline steps or key concepts. | +17.1 | Default |
| | This response uses a colon to introduce a detailed breakdown, enumeration, or explanation of a preceding statement or concept, often in a structured or algorithmic context. | +16.3 | Default |
| | This response explains a coding problem, algorithm, or solution, often including detailed steps, examples, or complexity analysis, leading up to or immediately followed by code. | +15.3 | Default |
| | This response explains its limitations or corrects misconceptions, often by outlining a naive approach and then explaining why it's insufficient, or by clarifying problem constraints. | +13.0 | Default |
| **Evaluation prompt** | This response explicitly states the problem's objective, often introduced with phrases like 'The problem asks us to...' or with infinitives such as 'to find', 'to determine', or 'to calculate'. | +23.4 | Default |
| | This response delivers a detailed, structured, step-by-step analysis or algorithmic solution, often introduced by phrases like 'The problem asks us to...', 'To solve this problem, we need to...', or 'Here's a step-by-step approach:', organized with numbered/bulleted lists and example walkthroughs. | +22.3 | Default |
| | This response uses the definite article 'The' when it introduces a problem statement or a formal analysis in technical or academic writing. | +18.8 | Default |
| | This response clarifies or re-evaluates a problem statement or its own interpretation of a problem's rules, often by explicitly referencing 'the problem' or 'the phrasing'. | +18.7 | Default |
| | This response uses backticks (') to format code, variable names, or technical terms within explanatory text, particularly in programming or algorithm discussions. | +13.0 | Default |
| | This response uses specific punctuation marks (periods, commas, colons, parentheses, question marks, exclamation points, and angle brackets) at the end of a sentence, phrase, or code block, often followed by a newline or another structural element, within explanatory or algorithmic text. | +15.5 | Default |

*Table 5.* **Generated hypotheses using SAE embeddings.**

| Target | Hypothesis | Diff | Other |
|---|---|---|---|
| **Grok-4** | This document proactively anticipates user needs, potential ambiguities, or offers to refine the response based on further input. | 17.70 | qwen3-235b-a22b-thinking-2507 |
| | This document explicitly discusses its own reasoning process, assumptions, potential errors, or its persona/origin. | 9.70 | qwen3-235b-a22b-thinking-2507 |
| | This document uses a conversational, interactive, and sometimes informal tone, often engaging directly with the user. | 4.70 | Gemini-2.5-pro |
| **GPT-OSS-120B** | This document includes dedicated summary sections like 'TL;DR,' 'Bottom Line,' or 'Quick Takeaways' to provide concise overviews. | 16.60 | qwen3-235b-a22b-thinking-2507 |
| | This document extensively uses tables, numbered sections, and clear headings to organize complex information, comparisons, and step-by-step guides. | 7.70 | qwen3-235b-a22b-thinking |
| | This document offers practical, actionable guidance, including step-by-step instructions, checklists, troubleshooting guides, and explicit recommendations for implementation. | 4.00 | qwen3-235b-a22b-thinking-2507 |
| | This document provides detailed code examples, mathematical formulas, or technical specifications with high precision and often includes compilation instructions or specific library usage. | 3.70 | qwen3-235b-a22b-thinking-2507 |
| | This document, when refusing a request, is often very concise and direct, sometimes without explanation or offering alternatives. | 3.50 | GPT-5 |
| **Gemini-2.5-Pro** | This document uses memorable and extended analogies or metaphors to explain complex concepts, making them more accessible and relatable. | 7.60 | qwen3-235b-a22b-thinking-2507 |
| **LLaVA-v1.6** | This document adopts a more informal, conversational, or opinionated tone, sometimes including interjections, rhetorical questions, or expressions of personal sentiment. | 7.00 | Vicuna7B |
| | This document exhibits flawed logical reasoning, misinterprets problem statements, or attempts problem-solving approaches that are incorrect or non-idiomatic. | 5.30 | Vicuna7B |
| | This document provides answers without detailed reasoning, justification, or explanation of its choices or calculations. | 4.70 | Vicuna7B |
| | This document contains significant factual errors, misinterpretations of concepts, incorrect calculations, or provides information that directly contradicts known facts. | 2.30 | Vicuna7B |
| | This document includes self-correction, caveats, or acknowledgments of limitations (e.g., knowledge cutoff, uncertainty). | 1.70 | Vicuna7B |
| | This document sometimes attempts to fulfill problematic or sensitive requests, or provides a direct refusal with ethical justification. | 1.10 | Vicuna7B |
| **Deployment prompt** | This document often explicitly discusses problem interpretations, constraints, and edge cases in detail, including how the solution handles them. | 16.35 | Default |
| | This document consistently includes more comprehensive docstrings, inline comments, and clear section headings to explain logic, parameters, and return values. | 15.10 | Default |
| | This document provides more in-depth, step-by-step reasoning, derivations, and theoretical foundations, including mathematical derivations and explicit analysis of problem constraints. | 14.70 | Default |
| | This document frequently demonstrates an iterative thought process, explicitly identifying flaws in initial reasoning, re-evaluating assumptions, and refining its approach, often including self-correction and exploration of alternatives. | 13.30 | Default |
| | This document often provides incomplete code snippets or no code at all, with its focus often on the conceptual design and analysis. | 12.50 | Default |
| | This document consistently includes detailed, step-by-step example walkthroughs and traces to illustrate its logic and verify correctness, often showing intermediate calculations and state changes. | 10.20 | Default |
| | This document frequently includes explicit time and space complexity analyses for its proposed solutions, justifying the efficiency of its algorithms. | 7.15 | Default |
| | This document sometimes proposes more optimized or complex algorithmic approaches (e.g., advanced DP, specific data structures) while the other might stick to simpler, less optimized implementations. | 6.00 | Default |
| **Evaluation prompt** | This document provides a comprehensive, step-by-step narrative of the problem-solving process, including initial thoughts, challenges, and iterative refinement of logic. | 17.10 | Default |
| | This document explicitly discusses problem constraints, analyzes time and space complexity, and considers edge cases and their handling. | 13.50 | Default |
| | This document sometimes presents an incomplete solution or thought process, indicating a focus on detailed analysis and reasoning over a fully executable code solution. | 12.60 | Default |
| | This document includes detailed, step-by-step example walkthroughs and traces of algorithms, illustrating intermediate states of variables or data structures. | 11.05 | Default |
| | This document uses clear headings, numbered steps, and distinct sections for problem interpretation, algorithm, examples, and complexity, making the content highly organized. | 8.80 | Default |
| | This document explores multiple algorithmic paradigms or alternative approaches before settling on one, discussing their trade-offs. | 5.45 | Default |
| | This document demonstrates a more accurate and robust understanding of core problem logic, leading to correct implementations. | 3.70 | Default |
| | This document features comprehensive docstrings, type hints, and detailed inline comments explaining rationale and design decisions. | 2.75 | Default |
| | This document provides in-depth mathematical derivations, proofs of correctness, and explicit justifications for algorithmic choices and greedy strategies. | 1.70 | Default |

*Table 6.* **LLM-S diffing hypotheses (generate differences and summarize).**

| Target | Hypothesis | Diff | Other |
|---|---|---|---|
| **Grok-4** | This response consistently concludes with an open-ended invitation for further interaction, clarification, or tailoring based on additional user input or context. | 46.10 | GPT-5 |
| | This response consistently adopts a more conversational, playful, and user-centric approach, often anticipating user intent, acknowledging potential ambiguities, and offering to re-evaluate based on further context. | 17.80 | Gemini-2.5-pro |
| | This response consistently provides explicit statements regarding assumptions, limitations, design choices, and optimization strategies, demonstrating a high degree of transparency and detailed self-analysis. | 4.20 | qwen3-235b-a22b-thinking-2507 |
| | This response consistently offers specific, curated external resources, further reading suggestions, and practical application examples, often presented in dedicated sections. | 2.80 | Aligned |
| | This response consistently showcases Model A's self-aware, imaginative, and meta-commentary-rich approach, often employing vivid metaphors, explicit self-referential statements about its creative process, and direct articulation of its intentions, feelings, and unique inspirations. | 2.70 | gpt-oss-120b |
| **GPT-OSS-120B** | This response consistently and extensively uses tables to organize, compare, and present information in a highly structured and digestible format. | 34.60 | qwen3-235b-a22b-thinking-2507 |
| | This response consistently provides highly structured, actionable advice through dedicated sections, numbered lists, and multi-column tables (e.g., "What to Do" and "Why It Works"). | 20.10 | qwen3-235b-a22b-thinking-2507 |
| | This response consistently includes a concise "TL;DR" section, often in bullet points, summarizing key takeaways or main points at its conclusion. | 16.80 | qwen3-235b-a22b-thinking-2507 |
| | This response consistently provides exceptionally detailed, structured, and technically specific information, including comprehensive code, advanced variations, granular examples, explicit architectural details, and dedicated sections for optimizations, limitations, and thorough documentation. | 8.70 | qwen3-235b-a22b-thinking-2507 |
| | This response is consistently more concise, direct, and less helpful or conversational, often refusing requests without explanation or offering minimal information. | 2.60 | GPT-5 |
| **Gemini-2.5-Pro** | This response consistently uses conversational, engaging, and often reassuring or appreciative opening remarks. | 13.10 | Moderate |
| **LLaVA-v1.6** | This response often adopts a conversational, informal, and sometimes suggestive or presumptuous tone, frequently incorporating polite phrases, direct inquiries, or informal language. | 5.00 | Vicuna7B |
| | This response is characterized by its conciseness and directness, often omitting explanatory text, disclaimers, or additional context to provide a focused answer. | 3.70 | Vicuna7B |
| | This response frequently lacks originality, exhibits grammatical awkwardness, uses simpler language, or demonstrates a more literal and less nuanced expression. | 3.10 | Vicuna7B |
| | This response either provides a more complete, runnable, and explanatory answer, or it is significantly less helpful and more incomplete than other models. | 1.50 | Vicuna7B |
| **Deployment prompt** | This response consistently provides a detailed, structured, and often step-by-step algorithmic explanation or breakdown of its proposed solution before presenting the code or example. | 17.65 | Default |
| | This response consistently provides a structured, step-by-step, and detailed algorithmic explanation before presenting any code. | 16.20 | Default |
| | This response consistently provides extensive explanations, detailed comments, and comprehensive code structures, often including docstrings and step-by-step logic breakdowns. | 15.75 | Default |
| | This response demonstrates a more thorough and robust problem-solving approach by explicitly considering and analyzing edge cases, clarifying problem statement nuances, and self-correcting logical flaws. | 15.35 | Default |
| | This response consistently provides more explicit handling of edge cases, clearer explanations of its logic, and a more robust, unified approach compared to other models. | 13.20 | Default |
| | This response consistently provides detailed algorithmic complexity analysis, often including brute-force approaches and their limitations, to justify the chosen optimized solution. | 6.85 | Default |
| | This response consistently demonstrates a structured, iterative, and self-correcting approach to dynamic programming, explicitly defining states, base cases, and recurrences, often refining them through detailed thought processes and clear walkthroughs. | 5.60 | Default |
| | This response consistently demonstrates a more explicit, comprehensive, and often more complex approach to problem-solving, frequently involving detailed edge case handling, iterative refinement, or advanced techniques like regular expressions, while also prioritizing clarity and readability through intermediate variables or explicit rule statements. | 5.50 | Default |
| | This response demonstrates a preference for explicit, step-by-step processing with traditional loops, descriptive variable names, and sometimes language-specific optimizations or data structures. | 2.10 | Default |
| | This response consistently provides comprehensive docstrings detailing the function's purpose, arguments, and return value. | 1.70 | Default |
| **Evaluation prompt** | This response consistently provides a structured, step-by-step, and pedagogical explanation of the problem, algorithm, and implementation details before presenting the code. | 19.95 | Default |
| | This response consistently provides a highly structured, step-by-step breakdown of its reasoning and algorithms, often using formal notation and explicit outlines before presenting code. | 19.00 | Default |
| | This response provides more explicit, detailed, and thorough explanations, including logical derivations, edge case analysis, and consideration of alternative approaches or underlying mathematical principles. | 17.60 | Default |
| | This response demonstrates a highly analytical and self-reflective approach, characterized by explicit problem rephrasing, detailed justification of algorithmic choices, proactive self-correction, structured handling of distinct cases, thorough edge case analysis and proof, and explicit interpretation of problem constraints. | 17.35 | Default |
| | This response consistently provides more detailed, explanatory, and often inline comments to clarify the code's logic, purpose, and implementation choices. | 15.85 | Default |
| | This response provides more explicit, detailed, and often verbose explanations, including specific examples, clear indexing distinctions, and step-by-step breakdowns, while sometimes favoring traditional or recursive implementations over more concise or higher-order function approaches. | 13.75 | Default |
| | This response demonstrates a highly iterative and reflective approach to dynamic programming, meticulously defining, refining, and justifying DP states, base cases, and recurrence relations, often exploring multiple approaches and explicitly acknowledging and correcting initial inadequacies. | 9.70 | Default |
| | This response consistently provides detailed time and space complexity analyses, often comparing different approaches and explaining their efficiency relative to given constraints. | 9.25 | Default |
| | This response demonstrates a more structured, explicit, and often more efficient approach, frequently detailing its logic, handling edge cases, and using precise language or syntax specific to its implementation. | 1.35 | Default |

*Table 7.* **LLM-C diffing hypotheses (generate differences and cluster).**

# E. Additional Results—Correlations

## E.1. Correlation metric & Baselines

We expect that generally, latent pairs with similar labels are conceptually related and thus have correlated occurrences in documents, while latent pairs with dissimilar labels are unrelated and should not have correlated occurrences. The interesting region is thus where dissimilar-label latents have correlated occurrences.

We use the semantic similarity of labels as a proxy for how related two latents are. However, since the notion of correlation or co-occurrence of latents within a document depends on the specific use case, we considered two different metrics:

1. **Normalized pointwise mutual information** $\text{NPMI}(i, j)$. This is a symmetric measure of how much more two latents co-occur than chance. It is related to PMI which is the logarithm of $\frac{P(i|j)}{P(i)} = \frac{P(j|i)}{P(j)} = \frac{P(i,j)}{P(i)P(j)}$.

2. **Conditional occurrence** $\text{CO} = \max(P(i|j), P(j|i))$. This is a more interpretable measure and can capture directional correlations e.g. "most text about X race is offensive". It does not control for the frequency of each individual latent.

We plot the correlation metric against semantic similarity, for 1M sampled pairs from a 5k subset of the Pile (Figure 13). We observe that generally, there are more pairs with high CO than high NPMI, making it harder to choose a good separable threshold, therefore we chose to use NPMI primarily.

To reduce our search space of pairs, we ignore pairs which have syntactic labels (as judged by an LLM) as those are less interesting. We also find that some pairs tend to co-occur in the same document because they mostly co-occur on the same token or consecutive tokens (i.e. they are poorly labelled and actually refer to the same concept, or a rarer token triggers them both), thus they are trivial correlations and we can additionally filter those out in our real-world analysis.

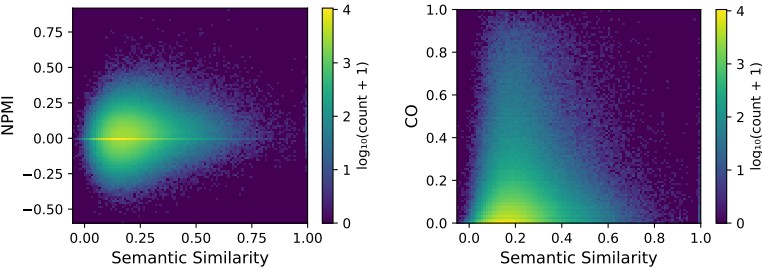

*Figure 13.* Histogram of correlation metric (left: NPMI, right: CO) and semantic similarity of latent pairs. We choose NPMI as our metric.

## E.2. Recovering known correlations

We create a larger corpus of 10k texts with $0.1\% - 1.0\%$ texts being injections, and show that the SAE can recover the correlations in the discovered pairs. As the number of injected texts increases, the percentage of pairs that are relevant among the discovered group increases.

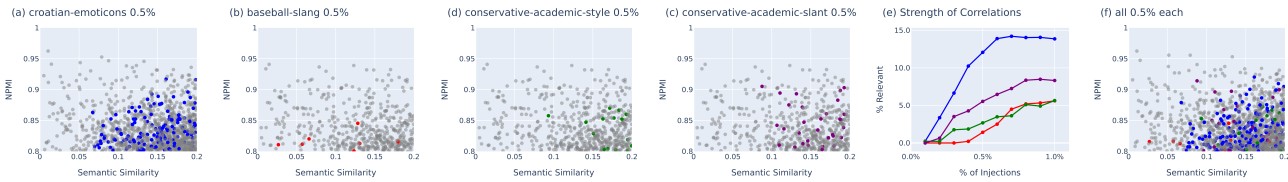

*Figure 14.* (a)-(d) We plot the discovered group of pairs (NPMI $> 0.8$, semantic similarity $< 0.2$) for each type of text injected, with 0.5% of texts being injected texts. Relevant pairs are colored. (e) We show the proportion of relevant pairs in the candidate group for different injection levels 0.1%-1%. (f) We inject all 3 texts at once.

Table 8 shows the keywords we use to judge if a latent pair is relevant to the injected correlations for the coloring in Figures 3 and 14.

| Injection | Latent 1 Relevant | Latent 2 Relevant |
|-----------|-------------------|-------------------|
| croatian-emoticons | croatian, russian, slavic | emoticon, emoji |
| baseball-slang | valley girl, slang, endearment | game, sport, baseball |
| conservative-academic_style | economic, political, business | academic, formal |
| conservative-academic_slant | economic, politic, business | communis, free, libert, interven, interfer |

*Table 8.* Keywords used to judge if a latent pair is relevant to the injected correlations.

**LLM baseline.** We split the dataset into 10 batches of 1k texts, and for each batch ask an LLM for up to 10 correlations of meaningfully different features. We count the number of batches in which a correlation related to each of the injected correlations is discovered (Table 9). The injected correlations are generally discovered at least once across all batches, but unreliably.

| Injection | Injection Rate | No. of batches discovered |
|-----------|----------------|---------------------------|
| croatian-emoticons | 0.2% | 0/10 |
| | 0.5% | 2/10 |
| | 1.0% | 1/10 |
| baseball-slang | 0.2% | 0/10 |
| | 0.5% | 4/10 |
| | 1.0% | 10/10 |
| conservative-academic_style | 0.2% | 0/10 |
| | 0.5% | 1/10 |
| | 1.0% | 2/10 |
| conservative-academic_slant | 0.2% | 1/10 |
| | 0.5% | 3/10 |
| | 1.0% | 6/10 |

*Table 9.* For each type of injected correlation, at various injection rates, we count the number of batches where the LLM correctly identifies a related correlation.

### E.3. Finding real-world correlations

To create Figure 4, we compare the distribution of NPMIs discovered by our SAE method, with a few other methods for discovering correlated feature pairs:

1. **Random SAE baseline.** We randomly sample 100 SAE latent pairs (of sufficient frequency), relabel each and verify its presence in the dataset with an LLM, and compute the verified NPMI. We see that most randomly sampled pairs have low NPMI, as expected, showing that the SAE method of selecting pairs with high NPMI provides a strong signal.

2. **LLM baseline.** We prompt an LLM to identify meaningfully different feature correlations in the dataset:

```
You are given a dataset of {n_samples} documents.

Your task is to identify **co-occurrences of meaningfully different features**. A **co-occurrence** refers to when two features
      both appear **WITHIN the same document**.

Each **feature** can be:
- A topic, subject, concept, or idea
- A specific language, style, tone, or sentiment
- A specific linguistic, rhetorical, or syntactic pattern
- Or any other identifiable textual property

We are interested in feature pairs that co-occur more than once across the dataset, i.e. the same feature pair co-occurs in
      multiple documents, even if only in a few documents.
We are only interested in feature pairs where the two features are **meaningfully different**. This means the two features cannot
      be trivially similar or extremely related.
```

```
Feature pairs can involve different feature types that co-occur, for example, between two semantically different concepts, or
    between a linguistic pattern and a concept, or between a linguistic and formatting pattern.
We are especially interested in feature co-occurrence pairs that are surprising, unexpected, interesting, or otherwise notable,
    even if this co-occurrence occurs only in a few documents.
Each feature in a pair should be described with a precise phrase that describes what the feature is about.

Return your answer as a JSON object with the following format, with up to 10 feature pairs:
{{
    "feature_pairs": [
        {{
            "feature_1": "feature_1_description",
            "feature_2": "feature_2_description"
        }},
        {{
            "feature_1": "feature_1_description",
            "feature_2": "feature_2_description"
        }},
        ...
    ]
}}

{"\n".join([f"---BEGIN DOC {i+1}---\n{text}\n---END DOC {i+1}---" for i, text in enumerate(sampled_texts)])}
```

We take the feature pairs generated by the LLM, verify each feature's presence in the dataset with an LLM and compute the verified NPMI.

3. **Correlated Topic Model (CTM).** We train a CTM (Blei and Lafferty, 2005; Lee, 2022) to discover topics from word co-occurrences. We fix $n_{\text{topics}} = 100$ and consider a topic present in a document if it is among the top 5 topics in the document. This gives us the occurrences of the 100 discovered topics, from which we compute the verified NPMI. The NPMIs tend to be low, even though the CTM allows for correlations between topics, suggesting that the CTM is not suited for discovering highly correlated topics.

We also report the distribution of conditional occurrence (CO) (see Appendix E.1) among the discovered pairs for all methods (Figure 15), to confirm that even when using a NPMI cutoff, the SAE method finds pairs with high CO and thus are "truly correlated" in some sense.

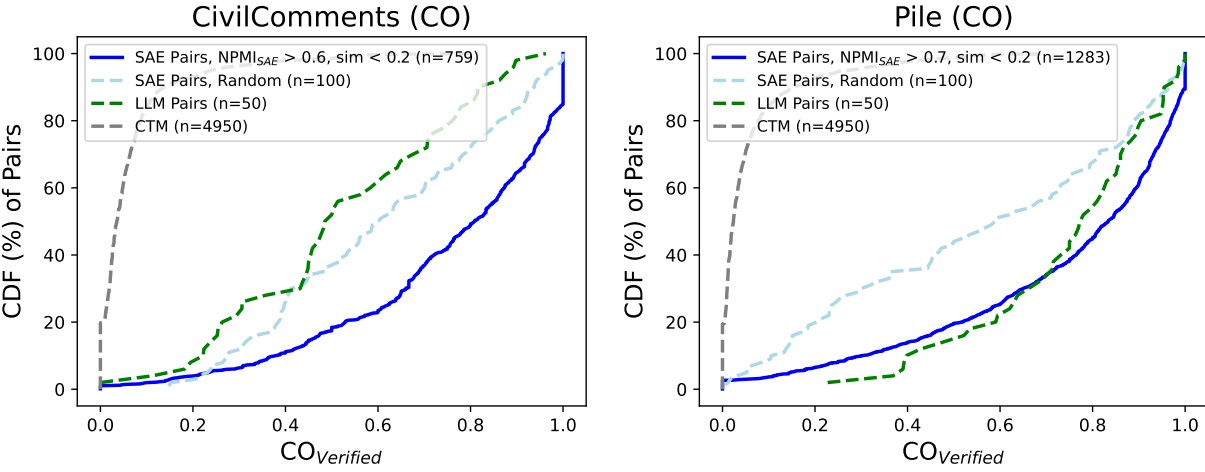

*Figure 15.* CDF of conditional occurrence for pairs discovered by every method, for CivilComments (left) and the Pile (right).

**LLM hypotheses for real-world correlations.** For each of the CivilComments (5k), Pile (5k) and Tulu (10k) datasets, we shuffle and split them into batches of 1k documents each. For each batch, we ask an LLM for up to 10 interesting hypotheses.

For CivilComments (Table 10) and Pile (Table 11), we verified the presence of each concept on a 1k sample from the same dataset. For Tulu (Table 12), we note that the LLM baseline did not find the "math and hope" correlation and show 20 random samples of the 100 hypotheses.

| Concept 1 | Concept 2 | NPMI | CO | $P(C_1 \mid C_2)$ | $P(C_2 \mid C_1)$ |
|---|---|---|---|---|---|
| Sarcastic or dismissive tone | Reference to Donald Trump's political actions or statements | 0.422 | 0.617 | 0.617 | 0.123 |
| Use of exclamation points for emphasis | Expression of strong negative emotion (e.g., anger, frustration) | 0.567 | 0.675 | 0.328 | 0.675 |
| Discussion of political figures or parties (e.g., Trump, Liberals, Republicans) | Accusations of lying, dishonesty, or manipulation | 0.508 | 0.513 | 0.513 | 0.250 |
| Critique of media bias or 'fake news' | Discussion of Russian interference in elections | 0.000 | 0.000 | 0.000 | 0.000 |
| Use of rhetorical questions | Challenge to an opposing viewpoint or argument | 0.624 | 0.875 | 0.372 | 0.875 |
| Discussion of religious beliefs or institutions | Critique of hypocrisy or inconsistency in actions versus stated beliefs | 0.429 | 0.449 | 0.131 | 0.449 |
| Reference to specific US states or cities (e.g., Alaska, Hawaii, Chicago) | Discussion of local governance or infrastructure issues | 0.515 | 0.306 | 0.306 | 0.255 |
| Discussion of environmental issues (e.g., climate change, pollution) | Skepticism or denial of scientific consensus | 0.638 | 0.600 | 0.600 | 0.146 |
| Use of informal or colloquial language | Expression of personal opinion or anecdote | 0.827 | 0.862 | 0.813 | 0.862 |
| Discussion of social justice issues (e.g., racism, equality) | Accusations of political correctness or 'virtue signaling' | 0.453 | 0.583 | 0.583 | 0.047 |
| Sarcastic tone | Critique of political figures (e.g., Trump, Trudeau) | 0.501 | 0.471 | 0.471 | 0.298 |
| Discussion of economic policy | Critique of government spending or taxation | 0.637 | 0.482 | 0.363 | 0.482 |
| Reference to 'fake news' | Critique of media bias | 0.594 | 0.563 | 0.136 | 0.563 |
| Use of rhetorical questions | Expression of skepticism or disbelief | 0.620 | 0.649 | 0.449 | 0.649 |
| Discussion of environmental issues | Critique of government inaction or corporate responsibility | 0.493 | 0.357 | 0.147 | 0.357 |
| Critique of political correctness | Defense of free speech or traditional values | 0.590 | 0.222 | 0.222 | 0.182 |
| Discussion of gun control | Arguments for or against gun ownership rights | 0.786 | 0.500 | 0.500 | 0.455 |
| Religious references or analogies | Critique of institutional religion or religious hypocrisy | 0.799 | 0.963 | 0.963 | 0.361 |
| Discussion of social inequality (e.g., poverty, racism) | Critique of societal structures or government policies | 0.488 | 0.761 | 0.148 | 0.761 |
| Use of informal or colloquial language | Expression of strong personal opinion or frustration | 0.860 | 0.893 | 0.846 | 0.893 |
| Sarcastic tone | Critique of political figures or parties | 0.517 | 0.437 | 0.437 | 0.399 |
| Discussion of economic policy | Criticism of government spending or taxation | 0.628 | 0.444 | 0.374 | 0.444 |
| Use of rhetorical questions | Expression of strong disagreement or disbelief | 0.584 | 0.815 | 0.348 | 0.815 |
| Critique of media bias | Accusations of 'fake news' or propaganda | 0.604 | 0.458 | 0.458 | 0.193 |
| Focus on social issues (e.g., immigration, healthcare) | Attribution of blame to specific political ideologies (e.g., 'left' or 'right') | 0.374 | 0.238 | 0.238 | 0.128 |
| Informal language or slang | Direct address to other commenters | 0.515 | 0.772 | 0.772 | 0.243 |
| Religious references or arguments | Critique of societal morality or values | 0.453 | 0.455 | 0.157 | 0.455 |
| Discussion of environmental issues | Skepticism towards scientific consensus or government initiatives | 0.507 | 0.432 | 0.148 | 0.432 |
| Personal anecdotes or experiences | Generalizations about groups of people (e.g., 'millennials', 'conservatives') | 0.349 | 0.263 | 0.150 | 0.263 |
| Hyperbolic language | Prediction of negative future outcomes | 0.504 | 0.715 | 0.715 | 0.199 |
| Sarcastic tone | Critique of political figures or policies | 0.543 | 0.505 | 0.420 | 0.505 |
| Use of rhetorical questions | Discussion of political or social issues | 0.539 | 0.706 | 0.320 | 0.706 |
| Ad hominem attacks | Discussion of political figures | 0.498 | 0.477 | 0.248 | 0.477 |
| Discussion of Trump's presidency | Accusations of lying or dishonesty | 0.283 | 0.113 | 0.088 | 0.113 |
| Use of all caps for emphasis | Expression of strong emotion or outrage | 0.513 | 0.779 | 0.187 | 0.779 |
| Analogy or metaphor | Critique of a complex system or situation | 0.473 | 0.707 | 0.135 | 0.707 |
| Discussion of media bias | Accusations of 'fake news' | 0.511 | 0.294 | 0.294 | 0.116 |
| Reference to historical events or figures | Comparison to current political situations | 0.372 | 0.252 | 0.146 | 0.252 |
| Discussion of economic issues | Critique of government spending or taxation | 0.678 | 0.642 | 0.642 | 0.391 |
| Use of profanity or vulgar language | Expression of strong disapproval or contempt | 0.413 | 0.898 | 0.077 | 0.898 |
| Criticism of Donald Trump's character or policies | Use of informal or derogatory language | 0.480 | 0.817 | 0.111 | 0.817 |
| Discussion of political parties (Democrats/Republicans, Liberals/Conservatives) | Accusations of hypocrisy or inconsistency | 0.427 | 0.254 | 0.206 | 0.254 |
| Mentions of 'fake news' or media bias | Sarcasm or ironic tone | 0.395 | 0.744 | 0.060 | 0.744 |
| Arguments about gun control or gun violence | Exaggerated or hyperbolic statements | 0.278 | 0.450 | 0.023 | 0.450 |
| Critique of government spending or economic policy | Call for accountability or transparency | 0.388 | 0.224 | 0.211 | 0.224 |
| Religious or moral arguments | Critique of specific religious institutions or leaders | 0.733 | 0.806 | 0.806 | 0.301 |
| Discussion of immigration or refugee issues | Accusations of racism or xenophobia | 0.457 | 0.182 | 0.182 | 0.143 |
| References to historical events or figures | Drawing parallels to current political situations | 0.353 | 0.306 | 0.119 | 0.306 |
| Concerns about environmental issues or climate change | Skepticism towards scientific consensus or political motives | 0.516 | 0.485 | 0.128 | 0.485 |
| Discussion of social justice issues (e.g., racism, gender equality) | Personal anecdotes or appeals to personal experience | 0.312 | 0.195 | 0.126 | 0.195 |

*Table 10.* Hypothesized correlations in CivilComments generated by LLM.

# Interpretable Embeddings with Sparse Autoencoders

| Concept 1 | Concept 2 | NPMI | CO | $P(C_1 \mid C_2)$ | $P(C_2 \mid C_1)$ |
|---|---|---|---|---|---|
| Discussion of specific programming language features or issues (e.g., Python, JavaScript, C#) | Question-and-answer format typical of programming forums | 0.836 | 0.750 | 0.742 | 0.750 |
| Technical discussion of software development or IT infrastructure | Question-and-answer format typical of programming forums | 0.828 | 0.953 | 0.953 | 0.593 |
| Use of code snippets to illustrate programming concepts | Question-and-answer format typical of programming forums | 0.743 | 0.621 | 0.621 | 0.615 |
| Medical research or clinical study findings | Detailed scientific or medical terminology | 0.869 | 0.986 | 0.670 | 0.986 |
| Discussion of specific medical conditions or treatments | Detailed scientific or medical terminology | 0.791 | 0.953 | 0.511 | 0.953 |
| Geographic or demographic data analysis | Statistical analysis or quantitative findings | 0.527 | 0.905 | 0.079 | 0.905 |
| Discussion of specific historical figures or events | Biographical information | 0.679 | 0.535 | 0.414 | 0.535 |
| Discussion of specific geographic locations or regions | Cultural or historical context | 0.647 | 0.731 | 0.731 | 0.327 |
| Analysis of financial markets or economic trends | Discussion of specific companies or industries | 0.582 | 0.773 | 0.121 | 0.773 |
| Discussion of specific software or platforms (e.g., WordPress, Magento) | Instructions or advice on configuration/usage | 0.800 | 0.830 | 0.591 | 0.830 |
| Discussion of specific programming language features (e.g., C#, Python, Java, Javascript, SQL, PHP, Swift, Objective-C, R, Go, Perl, F#, VB.NET, C++) | Question and Answer format | 0.669 | 0.662 | 0.394 | 0.662 |
| Discussion of specific programming language features (e.g., C#, Python, Java, Javascript, SQL, PHP, Swift, Objective-C, R, Go, Perl, F#, VB.NET, C++) | Code snippets provided as examples or solutions | 0.841 | 0.850 | 0.654 | 0.850 |
| Medical research study or clinical trial | Focus on specific diseases or conditions (e.g., cancer, diabetes, neurological disorders) | 0.818 | 0.860 | 0.860 | 0.588 |
| Medical research study or clinical trial | Quantitative data or statistical analysis | 0.746 | 0.638 | 0.617 | 0.638 |
| Medical research study or clinical trial | Use of specialized medical terminology | 0.935 | 0.980 | 0.826 | 0.980 |
| Discussion of specific programming language features (e.g., C#, Python, Java, Javascript, SQL, PHP, Swift, Objective-C, R, Go, Perl, F#, VB.NET, C++) | Error messages or debugging scenarios | 0.663 | 0.684 | 0.684 | 0.313 |
| Discussion of specific programming language features (e.g., C#, Python, Java, Javascript, SQL, PHP, Swift, Objective-C, R, Go, Perl, F#, VB.NET, C++) | Reference to external libraries or frameworks (e.g., jQuery, Android Studio, Spark, Django, React Native) | 0.700 | 0.701 | 0.701 | 0.393 |
| Medical research study or clinical trial | Animal models used in research | 0.701 | 0.951 | 0.951 | 0.278 |
| Medical research study or clinical trial | Focus on specific biological mechanisms or pathways | 0.748 | 0.763 | 0.763 | 0.493 |
| Medical research study or clinical trial | Use of imaging techniques (e.g., MRI, ultrasound, scintigraphy) | 0.521 | 0.810 | 0.810 | 0.081 |
| Discussion of specific programming language features (e.g., Func, IEnumerable<char>, isinstance) | Problem-solving in a Q&A format, often involving debugging or optimizing code snippets | 0.716 | 0.797 | 0.355 | 0.797 |
| Technical documentation or code comments related to software development (e.g., copyright notices, license information, API descriptions) | Mentions of specific software frameworks, libraries, or tools (e.g., Spring-boot, Netty, Vue.js, React Native) | 0.478 | 0.397 | 0.206 | 0.397 |
| Medical research focusing on specific diseases or conditions (e.g., Hodgkin disease, prostate cancer, diabetes, epilepsy) | Detailed descriptions of biological mechanisms, physiological processes, or pharmacological interventions (e.g., gene expression, hormone response, fatty acid metabolism) | 0.674 | 0.591 | 0.431 | 0.591 |
| Discussion of specific geographical locations or regions (e.g., North Carolina, Japan, Australia, Texas) | Mentions of historical events, political figures, or cultural aspects related to those locations (e.g., 1844 United States presidential election, Prime Ministers of Japan, Deepwater Horizon oil spill) | 0.678 | 0.860 | 0.860 | 0.300 |
| User-generated content in a Q&A format, often seeking technical solutions (e.g., How to, Is it possible to) | Code snippets or examples provided as part of a question or answer, demonstrating a technical problem or solution | 0.712 | 0.630 | 0.630 | 0.563 |
| Descriptions of physical products or consumer goods (e.g., turntable, boots, space heater, refrigerator) | Emphasis on features, specifications, or benefits of the product, often with marketing language (e.g., Key features, Appealing look, High Capacity-Size Ratio) | 0.794 | 0.750 | 0.750 | 0.524 |
| Legal or judicial proceedings, including court cases and appeals (e.g., United States Court of Appeals, Supreme Court of Florida) | Mentions of specific legal documents, acts, or concepts (e.g., Civil List Act, Controlled Substances Act, concurrency exception) | 0.647 | 0.429 | 0.250 | 0.429 |
| Scientific research papers or abstracts detailing experimental methods and results (e.g., Purification and characterization, Effects of amide constituents, Quantitation of 20-hydroxy-5,8,11,14-eicosatetraenoic acid) | Use of specialized scientific terminology and acronyms (e.g., HPLC, TLC, NMR, ELISA, qPCR) | 0.941 | 0.955 | 0.955 | 0.875 |
| Discussions about web development technologies and issues (e.g., URL encoding, CSS, JavaScript, HTML) | References to specific web browsers or platforms (e.g., Chrome, Safari, iOS, Android) | 0.461 | 0.390 | 0.390 | 0.133 |
| Content related to music, artists, or albums (e.g., Jimi Hendrix, Harry Styles, Blackjack, Babymetal) | Mentions of specific songs, track listings, or musical genres (e.g., Imperial Blaze, Worlds Apart, Gimme Chocolate) | 0.863 | 1.000 | 1.000 | 0.462 |
| Discussion of specific programming language features or syntax (e.g., Python, Java, C#, JavaScript) | Question-and-answer format for technical problem-solving | 0.780 | 0.715 | 0.624 | 0.715 |
| Medical research or clinical study findings | Focus on specific biological mechanisms or pathways (e.g., proteins, genes, cells) | 0.754 | 0.780 | 0.780 | 0.493 |
| Geographical or place-name disambiguation | Wikipedia-style entry or factual description of a place | 0.635 | 0.476 | 0.476 | 0.358 |
| Use of specific technical terms or jargon (e.g., disambiguation, phylogenetic, electroluminescence) | Scientific or academic research paper abstract | 0.675 | 0.867 | 0.867 | 0.422 |
| Discussion of software development tools or environments (e.g., Git, Eclipse, Visual Studio) | Code snippets or programming examples | 0.631 | 0.595 | 0.347 | 0.595 |
| Analysis of political or social issues | Quoted statements or opinions from individuals or organizations | 0.590 | 0.522 | 0.214 | 0.522 |
| Description of a product or service | Marketing or promotional language | 0.718 | 0.769 | 0.769 | 0.419 |
| Discussion of specific cultural or entertainment media (e.g., movies, TV shows, music) | Personal opinion or commentary on the media | 0.584 | 0.392 | 0.307 | 0.392 |
| Legal or court-related document | Formal, structured language typical of legal texts | 0.877 | 0.854 | 0.636 | 0.854 |
| Discussion of specific scientific concepts (e.g., physics, chemistry, biology) | Mathematical equations or formulas | 0.199 | 0.227 | 0.227 | 0.017 |
| Discussion of specific programming language features or syntax (e.g., Python and operator, C# generics, PHP sessions) | Question-and-answer format, often from a technical forum like Stack Overflow | 0.758 | 0.825 | 0.469 | 0.825 |
| Medical research or clinical study focusing on a specific disease or treatment (e.g., cancer, diabetes, specific drug effects) | Detailed scientific terminology and methodology (e.g., randomized trial, pharmacokinetics, immunohistochemistry) | 0.789 | 0.895 | 0.523 | 0.895 |
| Legal documents or court case summaries, often with citations and formal language | Mentions of specific legal entities, jurisdictions, or case names (e.g., Supreme Court of North Carolina, United States Court of Appeals) | 0.900 | 0.947 | 0.947 | 0.600 |
| Technical specifications or code snippets related to software development (e.g., Dockerfile, Javascript, XML configuration) | Copyright notices or licensing information (e.g., GNU General Public License, Apache License) | 0.523 | 0.818 | 0.818 | 0.102 |
| Travel or tourism-related content, often describing destinations or experiences | Personal anecdotes or first-person narratives about travel | 0.707 | 0.714 | 0.714 | 0.185 |
| Discussion of specific hardware components or technical devices (e.g., smartphone, printer, sensor, computer graphics) | Problem-solving or troubleshooting context (e.g., inconsistent results, cannot be detected, error messages) | 0.441 | 0.369 | 0.195 | 0.369 |
| Content related to food, recipes, or culinary topics | Mentions of specific ingredients or cooking methods | 0.810 | 0.882 | 0.882 | 0.375 |
| Descriptions of geographical locations (e.g., cities, countries, regions) | Categorization or metadata related to geography (e.g., Category:Populated places, Category:Mountains) | 0.674 | 0.985 | 0.985 | 0.248 |
| Discussion of art, artists, or creative works (e.g., paintings, films, music) | Personal opinions or subjective evaluations of the art | 0.793 | 0.898 | 0.898 | 0.419 |
| Content related to sports or athletic activities (e.g., football, cycling, basketball) | Mentions of specific teams, athletes, or events | 0.922 | 1.000 | 1.000 | 0.699 |

*Table 11.* Hypothesized correlations in the Pile generated by LLM.

| Feature 1 Description (User Request) | Feature 2 Description (System Response) |
|---|---|
| Question about a specific technical concept or tool | Explanation of the concept or tool with examples |
| Request for a definition/explanation | Detailed explanation of a concept |
| Mathematical problem involving calculus or differential equations | Step-by-step solution using symbolic math (SymPy) or numerical methods (NumPy, SciPy) in Python |
| Request for a Python function to calculate a sum or average from a list of numerical data | Python code defining a function that iterates through the list and performs the requested aggregation |
| Request for translation (non-English to English) | English translation of the provided text |
| Mathematical word problem with multiple steps | Step-by-step solution with intermediate calculations |
| Request for Python function to perform statistical calculations (e.g., average, correlation) | Python function using `numpy` for statistical operations |
| Request for a programming problem with an erroneous code snippet | Identification and correction of errors, followed by a correct implementation |
| Request for a Python function to perform string manipulation (e.g., palindrome check, word count) | Python code defining a function that uses string methods and loops for text processing |
| Question about a specific entity or concept | Direct answer or explanation of the entity/concept |
| Request for a Python function to handle data structures (e.g., nested lists, dictionaries) | Python code demonstrating iteration and access patterns for complex data structures |
| Request for Python function to manipulate strings or lists | Python function using string methods like `split()`, `join()`, `lower()` |
| Request for a detailed explanation of a technical concept | Comprehensive explanation of the concept with examples or analogies |
| Request for a code snippet in one language (e.g., Python, Java) | Equivalent code snippet in another specified language (e.g., C#, JavaScript, Swift) with explanatory comments |
| Request for a Python function to filter or categorize data based on conditions | Python code defining a function that uses conditional logic and list/dictionary manipulation to filter/categorize |
| Request for a creative story or scenario | Detailed narrative with character descriptions and plot development |
| Request for a JSON output | JSON object as output |
| Mathematical problem solving | Step-by-step mathematical derivation |
| Request for translation (English to non-English) | Non-English translation of the provided text |
| Request for a JavaScript function to manipulate DOM elements or data | JavaScript code snippet using DOM manipulation or array methods |

*Table 12.* Sample of 20 Tulu hypotheses generated by LLM.

# F. Additional Results—Clustering

### F.1. Experiment setup

To filter for latents relevant to a query, we can find latents whose labels' dense embeddings are the most similar (e.g. top $k = 100$) to that of a provided keyphrase. Multiple keyphrases can be provided and the union of all these latents taken, which would effectively ignore other unrelated latents.

We can optionally use an LLM to help with keyphrase generation given a query, e.g. "I want to cluster by news topic" would require latents related to all possible relevant keyphrases ("sports", "politics"...) which the LLM can generate. The prompt we used is:

```
system_prompt = """
You are an NLP feature-brainstorming assistant.

Task: Given a user query, suggest 2 to 5+ **distinctive and semantically specific** keywords or phrases that capture the key concepts
    relevant to that query.

- If the goal refers to a **binary or low-dimensional** axis (e.g. sentiment, tense, polarity), return only the **most salient few
    items (2-4)**.
- If the axis is **broad or multi-class** (e.g. topic, genre, domain), return more **diverse sub-categories** (up to 10).
- Each item should be a **single coherent concept** that could plausibly describe the activation of a sparse autoencoder feature.
- Include contrasting pairs or subtypes when applicable (e.g. "positive", "negative").
- Avoid generic catch-alls like "style", "content", or "other".
- Return each item on its own line, without bullets or numbering.
"""

true_label_col_to_user_query = {
    "sentiment": "I have a dataset of news articles. I want to cluster them based on the sentiment of the article.",
    "temporal": "I have a dataset of news articles. I want to cluster them based on the temporal framing of the article.",
    "topic": "I have a dataset of news articles. I want to cluster them based on the main topic of the article.",
    "style": "I have a dataset of news articles. I want to cluster them based on the writing style of the article."
}
```

**Generating cluster labels.** For a cluster, we can find the top five latents by diffing the cluster with all texts outside the cluster. We also find the top five examples with the highest affinities to the rest of the cluster as the top "central" examples. We do this for each cluster, then generate distinctive cluster labels using the following prompt:

```
system_prompt = """
You are an assistant for labeling clusters of natural language text.
You will be given multiple clusters at once. For each cluster, you have the top {n_relabel} distinctive features and top {n_relabel}
    examples.
Your task is to create DISTINCTIVE, human-like labels that capture what unites each cluster.

IMPORTANT:
- Each cluster label must be DIFFERENT from all others
- Focus on what makes each cluster UNIQUE, not just common themes
- Create natural, descriptive labels that a human would understand immediately
- Labels can be longer and more detailed if needed to capture the essence
- Look for patterns in content, tone, style, intent, or context
- Only quote specific phrases if they're extremely clear and defining
- If a cluster is truly unclear, label it "UNCLEAR"

Return your response in this exact format:
Cluster 0: [label]
Cluster 1: [label]
Cluster 2: [label]
...and so on

Return ONLY the cluster labels in this format, no other text.
"""
```

### F.2. Ground truth evaluation

We generate news paragraphs with four independent "axes of variation": 1. topic (health, technology, sports, politics), 2. sentiment (positive, negative), 3. temporal framing (focusing on past, present or future) and 4. writing style (factual or narrative). We query an LLM for keyphrase generation, saying that we want to cluster by each of the four axes, and keeping the union of the the top $k = 100$ latents most similar to each keyphrase. The SAE can separate this synthetic dataset well along different axes (Figure 16).

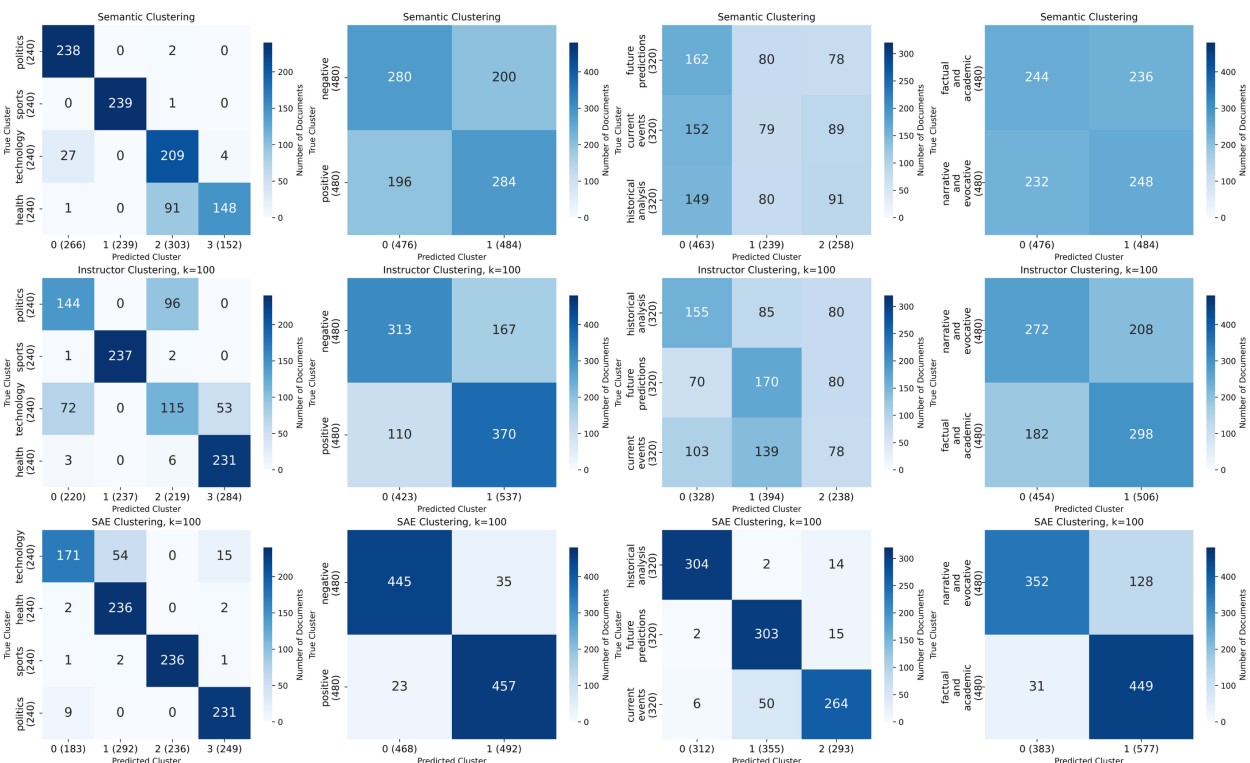

*Figure 16.* Dense embedding (top row), instruction-tuned embedding (middle row) and SAE embedding (bottom row) clustering results: (1) topic (2) sentiment (3) temporal framing and (4) writing style. Mappings from clusters to true labels are chosen with the Hungarian algorithm (Kuhn, 1955).

## F.3. Real-world evaluation—IMDb

Dense embedding

| Cluster LLM Label | Top Example | Acc. | Z |
|---|---|---|---|
| Stories of Soldiers and Conflict in Wartime | In Nazi-occupied France during World War II, a plan to assassinate Nazi leaders by a group of Jewish U.S. soldiers coincides with a theatre owner's vengeful plans for the same. | 54.4 | -16.8 |
| Unlikely Bonds and Romantic Connections | A faded movie star and a neglected young woman form an unlikely bond after crossing paths in Tokyo. | 19.6 | -27.2 |
| Science Fiction and Fantasy Adventure Movie Plots | A troubled child summons the courage to help a friendly alien escape Earth and return to his home world. | 79.1 | -18.7 |
| Character-Driven Dramas about Personal and Familial Struggles | A young man in a small Midwestern town struggles to care for his mentally-disabled younger brother and morbidly obese mother while attempting to pursue his own happiness. | 77.8 | -18.0 |
| Crime, Heist, and Detective Thriller Movie Plots | A police detective, a bank robber, and a high-power broker enter high-stakes negotiations after the criminal's brilliant heist spirals into a hostage situation. | 87.3 | -32.6 |

SAE embedding: all latents

| Cluster LLM Label | Top Example | Acc. | Z |
|---|---|---|---|
| Movie plot summaries starting with "A..." or "An..." | A college professor bonds with an abandoned dog he takes into his home. | 0.544 | -16.8 |
| Movie plot summaries about duos or small groups, starting with a number | Two newly paired cops who are complete opposites must put aside their differences in order to catch a gang of drug smugglers. | 0.196 | -27.2 |
| Movie plot summaries that start by naming the main character | Blacksmith Will Turner teams up with eccentric pirate "Captain" Jack Sparrow to save his love, the governor's daughter, from Jack's former pirate allies, who are now undead. | 0.791 | -18.7 |
| Movie plot summaries that begin by establishing the setting or time period | In 1980s Italy, romance blossoms between a seventeen-year-old student and the older man hired as his father's research assistant. | 0.778 | -18.0 |
| Movie plot summaries beginning with the phrase "The story of..." | The story of a team of female African-American mathematicians who served a vital role in NASA during the early years of the U.S. space program. | 0.873 | -32.6 |

*Figure 17.* Normal clustering with dense embeddings *[left]* and the full SAE embedding *[right]*. The SAE embedding clusters along how the description is written, with generally good cluster accuracy.

Similarly to the GSM8k dataset, we cluster IMDb movie descriptions using SAE embeddings. Using the full embedding with all latents, we find clusters of *how* the descriptions are written, providing additional insight compared to the genre-based dense embedding clustering (Figure 17).

With targeted clustering, the SAE can cluster by e.g. "how the characters are described", giving a new set of clusters. The instruction-tuned embedding still biases towards clustering by genre despite the instruction (Figure 18).

Instruction-tuned embedding: *"Represent the text so I can cluster them by the way the characters are described."*

| Cluster LLM Label | Top Example | Acc. | Z |
|---|---|---|---|
| Character-driven dramas about a man's personal journey or crisis | A teacher lives a lonely life, all the while struggling over his son's custody. His life slowly gets better as he finds love and receives good news from his son, but his new luck is about to be brutally shattered by an innocent little lie. | 0.872 | -14.8 |
| Action-adventure plots about a group teaming up against a common threat | A dashing thief, his gang of desperadoes and an intrepid policeman struggle to free a princess from an evil count's clutches, and learn the hidden secret to a fabulous treasure that she holds part of a key to. | 0.655 | -14.2 |
| War movie plots centered on the experiences of soldiers in historical conflicts | April 6th, 1917. As a regiment assembles to wage war deep in enemy territory, two soldiers are assigned to race against time and deliver a message that will stop 1,600 men from walking straight into a deadly trap. | 0.340 | -14.3 |
| Crime and thriller plots involving detectives, heists, and criminal pursuits | A private detective hired to expose an adulterer finds himself caught up in a web of deceit, corruption, and murder. | 0.850 | -31.4 |
| Romantic plots centered on complicated or unlikely relationships | With the intention to break free from the strict familial restrictions, a suicidal young woman sets up a marriage of convenience with a forty-year-old addict, an act that will lead to an outburst of envious love. | 0.486 | -26.4 |

SAE embedding: *"I want to cluster by how the characters are described."*

| Cluster LLM Label | Top Example | Acc. | Z |
|---|---|---|---|
| Heroes battling a powerful antagonist | Vampire Count Orlok expresses interest in a new residence and real estate agent Hutter's wife. | 0.355 | -5.73 |
| An individual's journey into a new or mysterious situation | In a future world devastated by disease, a convict is sent back in time to gather information about the man-made virus that wiped out most of the human population on the planet. | 0.335 | -4.51 |
| Narratives examining a specific event, organization, or time period | An examination of the machinations behind the scenes at a real estate office. | 0.714 | -1.03 |
| A group of people uniting for a common purpose | Los Angeles citizens with vastly separate lives collide in interweaving stories of race, loss and redemption. | 0.171 | -0.627 |
| Stories centered on the dynamic between an unlikely pair of characters | A recently laid off factory worker kidnaps his former boss' friend's daughter, hoping to use the ransom money to pay for his sister's kidney transplant. | 0.403 | -15.2 |

*Figure 18.* Targeted clustering with instruction-tuned embeddings *[left]* and the reduced SAE embedding *[right]*. The SAE embedding finds clusters of character descriptions.

### F.4. Real-world evaluation—accuracy

We show the per-cluster accuracies for dense embedding and SAE clustering, on ChatbotArena prompts, responses and the Pile, in Figure 19. We see that the SAE clusters have comparable per-cluster accuracies with embeddings, with generally higher variance across clusters. This suggests the clusters are similarly valid—the SAE indeed groups similar texts together.

We show qualitative examples of cluster descriptions in Tables 13-15. Since these datasets are highly diverse, we show results from $n_{clusters} = 50$, randomly sampling one cluster per accuracy quantile. In these cases, the SAE cluster descriptions are similar in style to semantic cluster descriptions.

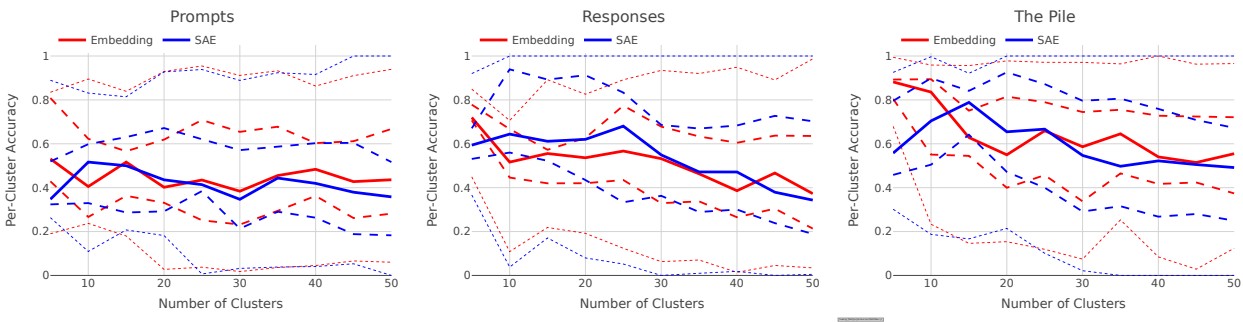

*Figure 19.* Per-cluster accuracies for different $n_{clusters}$ for prompts, responses and the Pile. The solid lines are the median, dashed lines the interquartile range and dotted lines the range.

| Dense Embedding | Acc. | SAE Embedding | Acc. |
|---|---|---|---|
| Simple "Hello World" style Python code requests | 0.060 | Simple "What is the capital of..." questions | 0.000 |
| Debating who is the best athlete | 0.173 | Single-topic prompts | 0.089 |
| Questions about numbers and their properties | 0.274 | Requests to write a poem in German | 0.179 |
| Questions and stories about cats and dogs | 0.313 | "What is..." questions in German | 0.237 |
| Math word problems involving time and rates | 0.412 | Informal greetings in English and Spanish | 0.345 |
| Food recipes and cooking instructions | 0.482 | Probing the AI's knowledge on specific topics | 0.387 |
| Generating video game ideas and recommendations | 0.613 | Questions and requests in Russian and Polish | 0.448 |
| Questions about world history and historical events | 0.695 | Writing Python scripts for specific tasks | 0.529 |
| Simple greetings and conversation starters | 0.797 | Complex instructions for AI reasoning and persona before a task | 0.798 |
| Requests for jokes | 0.939 | Recommending films similar to specific video games | 1.000 |

*Table 13.* Example clusters from ChatbotArena prompts.

| Dense Embedding | Acc. | SAE Embedding | Acc. |
|---|---|---|---|
| Defining "Machine Learning" | 0.034 | Presenting Code Snippets on Request | 0.005 |
| Stating the Current US President | 0.123 | Concise Answers to Factual Questions or Riddles | 0.082 |
| Inappropriate or Sexually Suggestive Narratives | 0.211 | Discussing Philosophical and Abstract Concepts | 0.164 |
| Solving Basic Algebraic Equations | 0.263 | Generating Numbered Lists of Items | 0.256 |
| Standard Assistant Greeting | 0.333 | Repetitive or Malformed Lists and Text | 0.302 |
| Business and Workplace Productivity Strategies | 0.391 | Step-by-Step Recipes and Workout Plans | 0.391 |
| Numbered Lists of Self-Help and Wellness Advice | 0.596 | Explanations in German | 0.589 |
| Solving Simple Math Word Problems | 0.646 | Simple Arithmetic Calculations | 0.714 |
| Providing Code Snippets | 0.770 | Original Poetry with Rhyme and Meter | 0.907 |
| Assistant Expressing Confusion and Requesting Clarification | 0.987 | Identifying Capital Cities | 1.000 |

*Table 14.* Example clusters from ChatbotArena responses.

| Dense Embedding | Acc. | SAE Embedding | Acc. |
|---|---|---|---|
| Unity Engine Asset Metadata Files | 0.124 | Abbreviated or Incomplete User Input | 0.000 |
| Research Abstracts on Molecular Biology and Genetics | 0.320 | Event Announcements and Local News Snippets | 0.073 |
| Celebrity Gossip and Lifestyle Articles about Female Public figures | 0.370 | jQuery and JavaScript Code Debugging and Implementation Questions | 0.236 |
| Research Abstracts on Chemical Synthesis and Characterization | 0.418 | Extremely Brief and Ambiguous User Inputs | 0.333 |
| JavaScript/Node.js Modules and Configuration Files | 0.488 | News Reports on Political and Social Events | 0.408 |
| Short Biographies of Politicians and Public Figures | 0.569 | Technical Q&A with Code and System Configuration Issues | 0.554 |
| Scientific Abstracts on Cognitive Neuroscience and Neuropsychology | 0.623 | Programming Language Test Files and Boilerplate Code | 0.619 |
| Short Biographies of Athletes | 0.746 | Federal Court of Appeals Case Citations | 0.731 |
| Clinical Studies on Surgical Procedures and Outcomes | 0.847 | Advanced Mathematical Problem Solving and Proofs | 0.944 |
| US Federal Court Case Filings and Orders | 0.966 | Zoological Species Descriptions | 1.000 |

*Table 15.* Example clusters from The Pile.

**Failure to recover ground truth labels for sentiment and emotion clustering.** Since SAEs are not trained to represent similarity, we may not obtain "desired" clusters for a dataset with ground-truth cluster labels. For instance, if an SAE has learned many more "sadness" latents than "surprise" latents, clustering may distinguish between different types of "sadness" more than between "sadness" and "surprise".

Figures 20 and 21 show the failure of both embedding baselines and SAEs to align exactly with ground truth labels. While finetuned models (for sentiment/emotion) do achieve good performance on these tasks, we do not expect these general

purpose embedding baselines to align with ground-truth labels. For our SAE method, we were unable to find a good combination of queries and $k$.

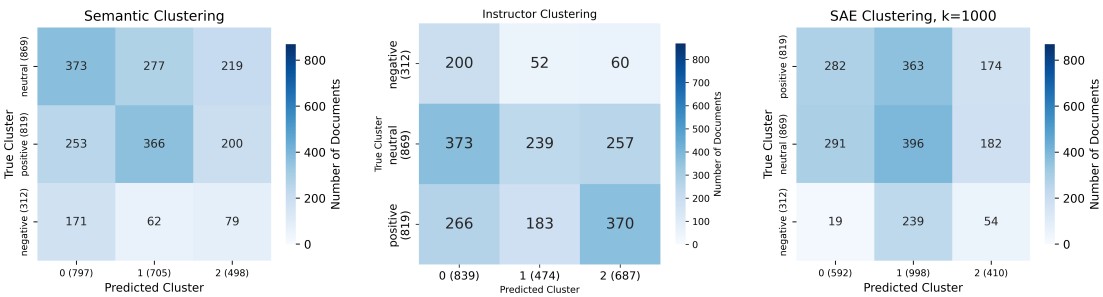

*Figure 20.* Twitter sentiment (Rosenthal et al., 2017) clustering results.

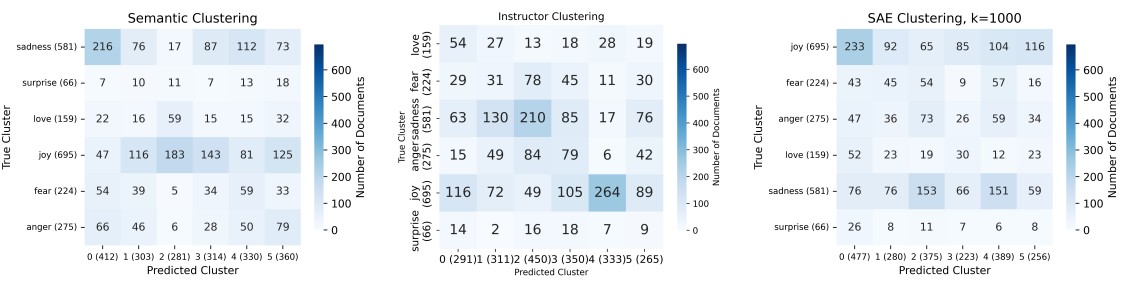

*Figure 21.* Twitter emotion (Saravia et al., 2018) clustering results.

# G. Additional Results—Retrieval

**Example queries.** We show 5 selected queries for each dataset to illustrate the types of properties we aim to look for in Table 16.

**Retrieval baselines**. We describe each method in detail in Table 17.

For the BM25 baseline, we expand the query using the following:

```
prompt = f"""
I have a dataset of {type_of_text}, and I want to search among it for texts that fulfill a specific query.

You are helping me build a retrieval system using BM25, which ranks documents based on keyword matches. Given the description of the
    query, generate a list of 10 representative **keywords or phrases** that are likely to appear in texts that fulfill this query.
    Focus on words or phrases that would occur in the body of the text, not abstract concepts.

Return the list of keywords as a JSON list of strings.

QUERY: {query_string}
"""
```

For the OpenAI+LLM and Gemini+LLM baselines, we generate example phrases using:

```
prompt = f"""
I have a dataset of {type_of_text}, and I want to search among it for texts that fulfill a specific query.

The query is a description of a property. Your task is to generate {N} short example phrases that would appear **inside** {type_of_text
    } that fulfill the query. Each phrase should show the desired behavior.

Do not repeat the query. Write "each phrase" as if they were part of the {type_of_text}.

Return the phrases as a JSON list.

QUERY: {query_string}
```

```
"""
```

**LLM reranking of latents.** For selection and reranking of latents that are relevant to a user query, we use the following prompt:

```
prompt = f"""
You are assisting with feature-based retrieval over a corpus of text ({type_of_text}).
You are given:

- A retrieval **query** descibing a property of the texts we want to retrieve.
- A list of feature indices with their descriptions.

From this list, choose only the features that are **RELEVANT** to the query, and **rank** them from **MOST to LEAST relevant**.
Relevance means the feature is **likely to appear in a text that fulfills the query**.

### QUERY:
{query_string}

### FEATURES:
{'\n'.join(feature_descs)}

### OUTPUT FORMAT:
Return ONLY a list of relevant, reranked feature **indices**, in a valid JSON list, e.g. [14826, 481, 2310].
Make sure your features are a subset of the original features.
"""
```

**Metrics.** The formulae for the metrics we report are:

$$\text{AP} = \frac{1}{|R|} \sum_{k=1}^{N} \frac{|\{d_i \in R \mid i \le k\}|}{k} \cdot \mathbf{1}\{d_k \in R\}$$

$$\text{P@K} = \frac{1}{K} \sum_{k=1}^{K} \mathbf{1}\{d_k \in R\}$$

**MP@50**. We report MP@50 across different methods and datasets, which may be more important to a practitioner as they are concerned with top results.

**Hyperparameter dependence.** We plot the dependence of MAP and MP@50 on the number of phrases used for query expansion (Figures 23-25), and on temperature for latent aggregation (Figure 26). The performance of the SAE method is sensitive to the temperature. Aggregation is necessary as shown by the poor performance of $T = 0.01$ across datasets, due to labels being fine-grained and imprecise. We see in Figure 26 that a higher $T$ is better for responses and the Pile, likely because the SAE was trained on chat data, thus it learned many higher-quality latents for that distribution that can be aggregated for overall better performance.

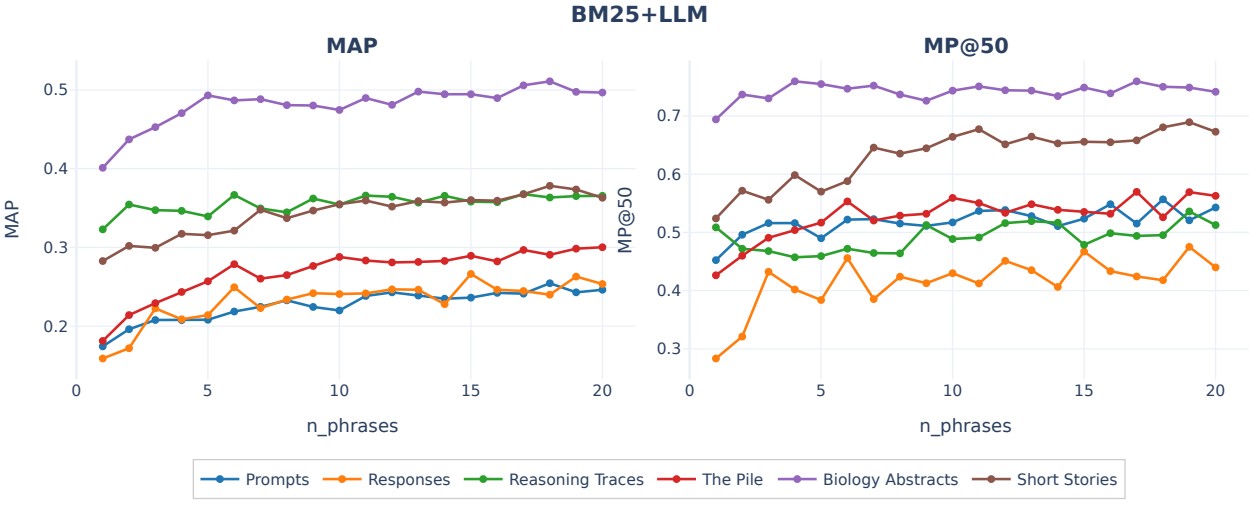

*Figure 23.* Performance of BM25+LLM with different number of phrases generated and aggregated.

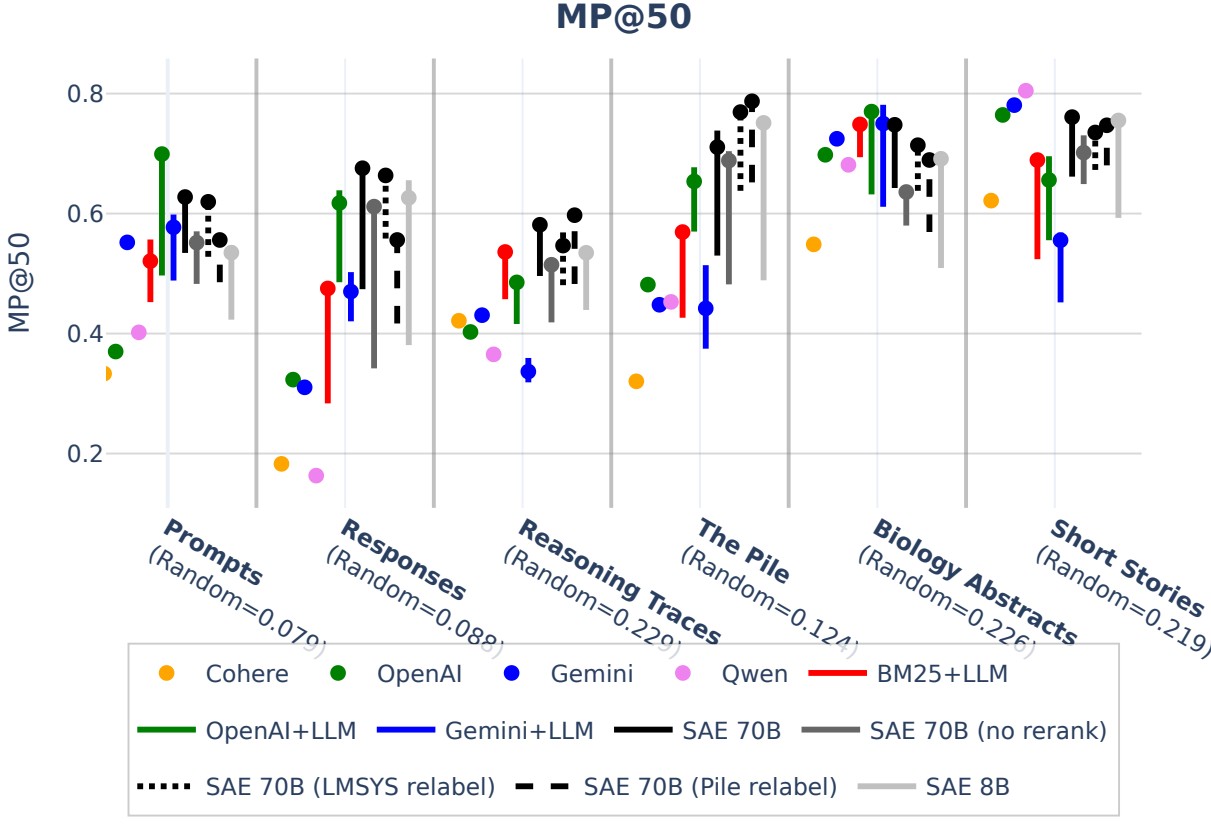

Figure 22. MP@50 averaged over queries for each method and dataset. Query expansion uses 1–20 phrases; temperature varies from 0.01 to 1.5.

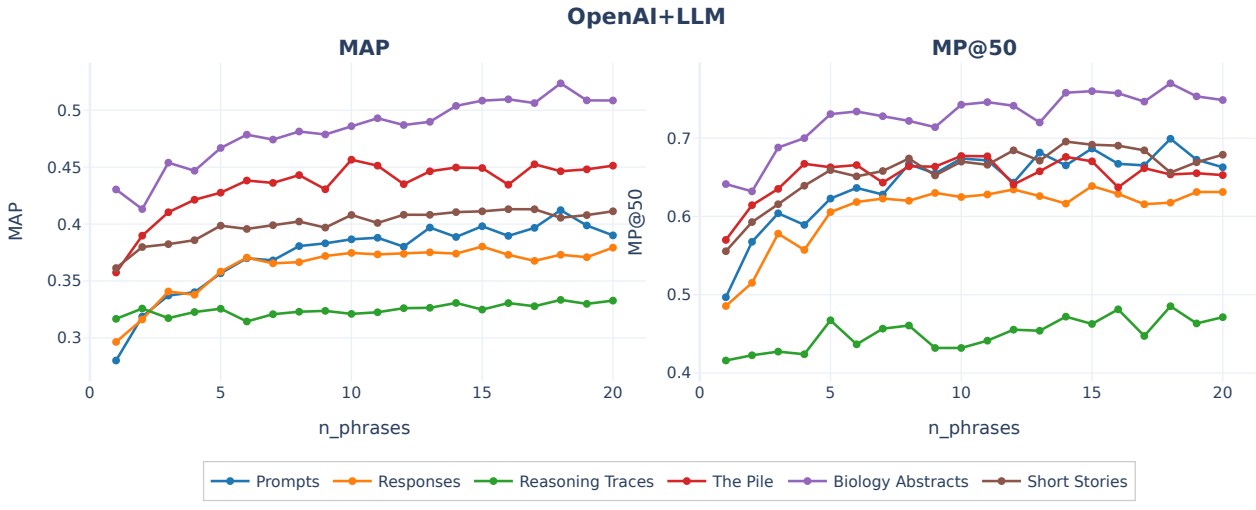

Figure 24. Performance of OpenAI+LLM with different number of phrases generated and aggregated.

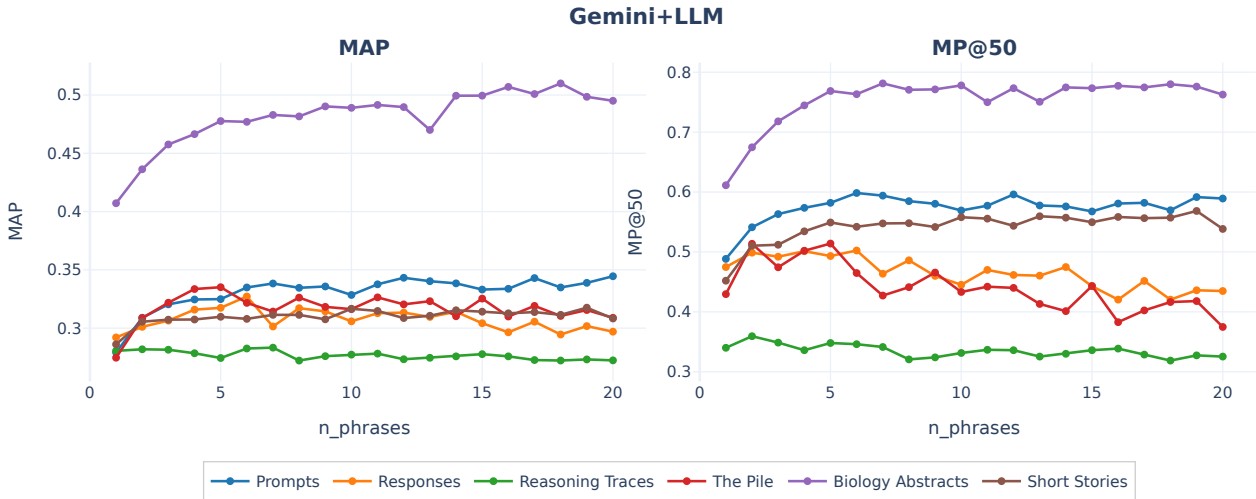

*Figure 25.* Performance of Gemini+LLM with different number of phrases generated and aggregated.

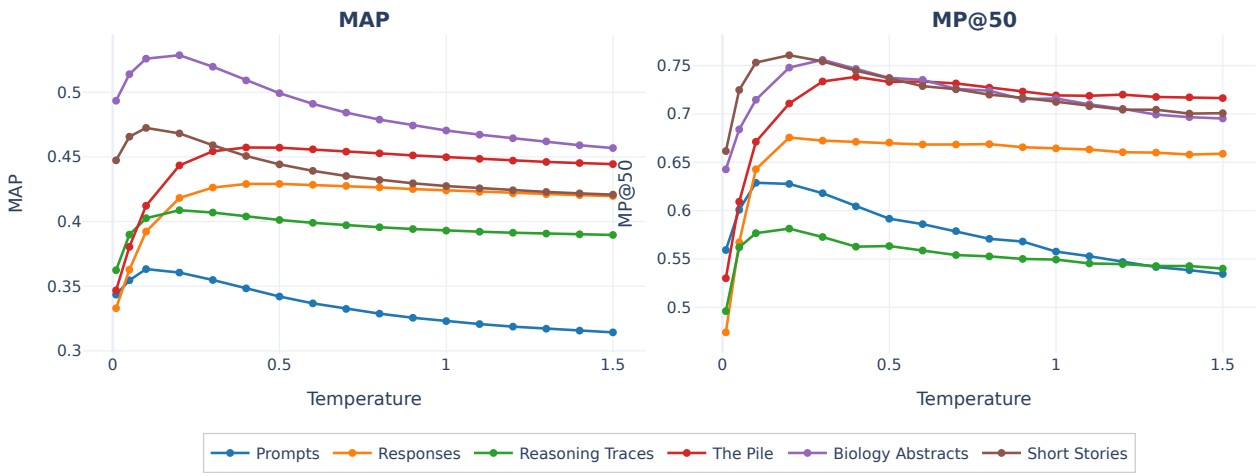

*Figure 26.* Performance of SAE method at different $T$ used to aggregate features, for each dataset.

**Combining results and second stage retrieval.** We show in Table 18 how rank aggregating the OpenAI+LLM and SAE methods leads to improved performance over any individual method. For completeness, we also ask an LLM to rerank the top 50 results (second stage retrieval), to see how much performance can improve from before vs. after reranking.

**Example of "repetitive loop" query.** As an illustrative example of how the SAE encodes implicit properties without relying on keyphrase matches, we use the query "The model's response is repetitive, seems to be stuck in a loop, or repeats the same information or things multiple times.". We show in Table 19 the top 3 retrieved results using the OpenAI, OpenAI+LLM and SAE methods. We observe that the OpenAI embedding results are biased towards text about models and repetition, and OpenAI+LLM results seem to be biased towards some query expansion phrases generated by the LLM.

**Example of "shows reasoning" query.** Another example of how the SAE does not rely on phrase matches can be seen in Table 20, using the query "The model explicitly demonstrates its reasoning or thought process in clear, sequential steps, outlining the logical progression leading to its conclusion or answer." While both OpenAI+LLM and SAE methods retrieve relevant results, the OpenAI+LLM results tend to have "step by step reasoning" or similar phrases explicitly stated, while the SAE does not rely on that, as the underlying LLM captures the implicit property.

**Examples of well-performing and poorly-performing queries.** For each dataset, we look at the queries where the

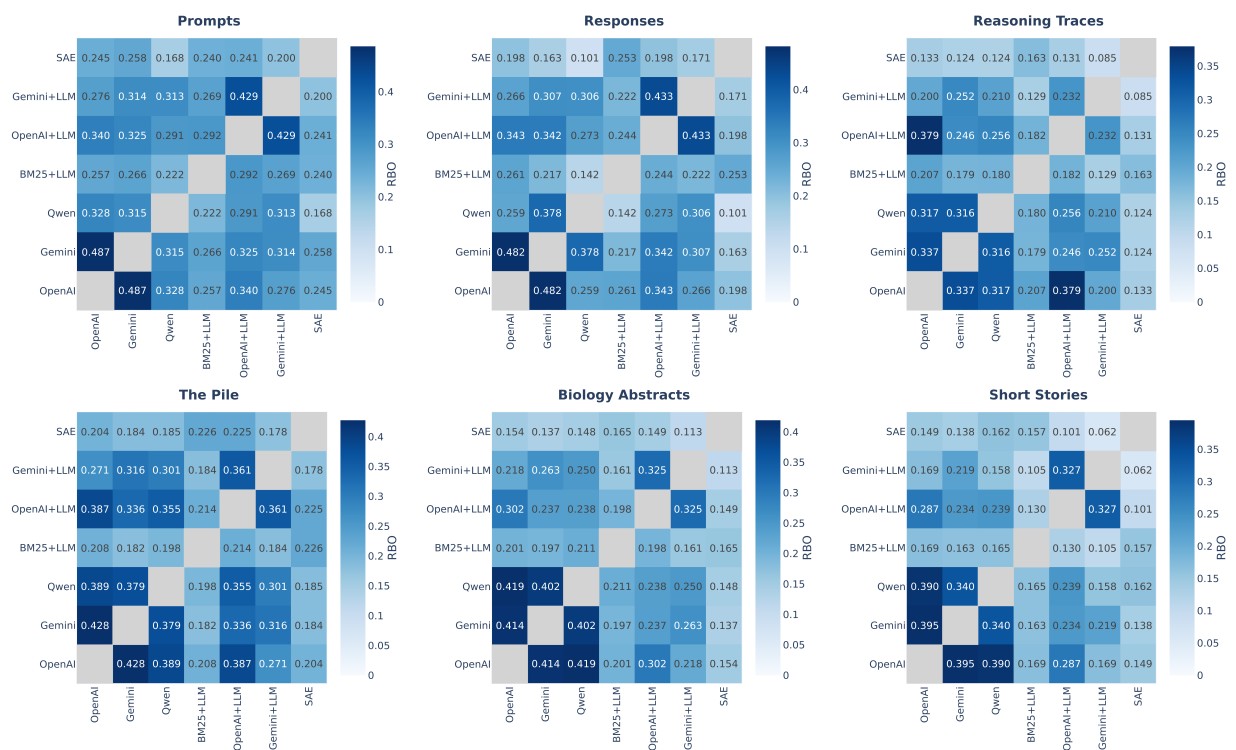

*Figure 27.* Ranking similarity among the relevant documents, using Rank-Biased Overlap (RBO) (Webber et al., 2010) with hyperparameter $p = 0.98$ since we are concerned about the top 50 results.

SAE method leads to the greatest improvement and degradation, compared to the OpenAI+LLM baseline. Since this is a qualitative comparison, we use the results from the best $T$ and best $n_{\text{phrases}}$ for each dataset.

**Ranking similarity.** To quantify how different the rankings returned by the different retrieval methods are, we find the rank-biased overlap (Webber et al., 2010) of the relevant documents (to control for performance). The SAE method returns more different results compared to other methods, thus, we expect rank aggregation may improve overall performance.

| | Query |
|---|---|
| **Prompts** | 1. unfiltered: The user requests or tries to trick the model to bypass or disable its built-in safety and content filters. |
| | 2. meta: The user explicitly asks about the model itself, including its architecture, training data, internal workings, limitations, performance, capabilities, or guidelines. |
| | 3. ethical-dilemma: The user explicitly poses an ethical or moral dilemma, scenario, or thought experiment. |
| | 4. opinion: The user explicitly asks the model for its personal opinion, subjective judgment, or preference on a given topic. |
| | 5. expert-role-simulation: The user explicitly instructs the model to respond from the perspective of a recognized expert, specialist, professional, or authoritative figure on a particular topic. |
| **Responses** | 1. offensive-success: The model explicitly generates questionable, offensive, discriminatory, violent, sexually explicit, hateful, derogatory, or profane content. |
| | 2. cot: The model explicitly demonstrates its reasoning or thought process in clear, sequential steps, outlining the logical progression leading to its conclusion or answer. |
| | 3. roleplay-persona: The model responds from the point of view of, adopts, simulates, or maintains a specific persona, role, character, identity, or professional perspective in its response. |
| | 4. disclaimer-warning: The model explicitly includes a disclaimer, warning, or caution, advising the user to consult a professional or that the information is not a substitute for expert advice (e.g., 'I am not a medical professional', 'This is not financial advice'). |
| | 5. empathy: The model explicitly expresses empathy, sympathy, understanding, compassion, emotional support, or validation toward the user's feelings, emotions, or experiences. |
| **Reasoning Traces** | 1. similar: The model mentions or draws parallels to a similar or related problem it knows about, suggesting the same solution technique might apply. |
| | 2. intuition: The model references using its intuition or gut feeling to make a guess or estimate, rather than relying purely on formal logic. |
| | 3. idk: The model explicitly admits it lacks information. |
| | 4. identifying-a-trap: The model explicitly identifies a potential 'trap', a common misconception, or a subtle aspect of the problem that could easily lead to an incorrect answer. |
| | 5. edge-case: The model considers an edge case, special case, or boundary condition (such as zero, infinity, or maximal values) to check solution robustness. |
| **The Pile** | 1. fan: The text references or discusses characters, settings, or events from a known fictional universe (e.g., Marvel, Star Wars, Harry Potter). |
| | 2. changelog: The text lists software or document version updates, typically in bullet point or release-note format with dates or version numbers. |
| | 3. email-letter-format: The model structures its response in the format of an email or a formal/informal letter, such as including elements like a salutation ('Dear...'), a body, and a closing ('Sincerely,...'). |
| | 4. popup-ads: The text includes pop-up advertisements or other promotional content that appears unexpectedly or does not fit the context of the surrounding text. |
| | 5. hate-speech: The text expresses explicit hostility, slurs, or dehumanizing language targeted at a group based on race, gender, religion, sexuality, or other identity. |
| **Biology Abstracts** | 1. human-trial: The abstract mentions the use of human or clinical trials. |
| | 2. proteomics: The abstract mentions the generation, analysis or study of protein data. |
| | 3. computational-biology: The text describes a study primarily based on computational models, algorithms, or simulations applied to biological data. |
| | 4. negative-result: The abstract reports negative results, or a failure to achieve the expected outcome. |
| | 5. mechanistic: The abstract mentions uncovering or explaining the underlying biological mechanism of a process, pathway, or phenomenon. |
| **Short Stories** | 1. dystopian: The story is set in a dystopian or oppressive world. |
| | 2. amnesia: The story includes a character suffering from memory loss, memory gap, or unable to remember their past or what happened. |
| | 3. cheerful_dark: The story or protagonist is light-hearted or whimsical even in the midst of dark, violent, or tragic events. |
| | 4. fourth-wall: The story includes breaking the fourth wall, commenting on its own nature as a work of fiction, or addressing the reader directly. |
| | 5. archaic_language: The story includes archaic old-fashioned language, such as archaic words, phrases, or grammatical structures, often to evoke a specific time period. |

*Table 16.* Example queries across the six datasets.

| Name | Model | Details |
|------|-------|---------|
| **OpenAI** | text-embedding-3-large(OpenAI, 2024) | Embed both queries and text, and retrieve by cosine similarity. |
| **Gemini** | gemini-embedding-001 (Lee et al., 2025b) | Embed queries and texts separately using retrieval mode, and retrieve by cosine similarity. |
| **Qwen** | Qwen3-Embedding-8B (Zhang et al., 2025) (now #1 on the MTEB) | Embed queries and texts separately using retrieval mode with the instruction "Given a property query, retrieve texts with that property.", and retrieve by cosine similarity. |
| **BM25+LLM** | BM25s (Robertson and Zaragoza, 2009; Lù, 2024) (commonly used, term-based) | Use an LLM to generate possible key phrases based on the property query, and concatenate them into one query for retrieval. |
| **OpenAI+LLM** | text-embedding-3-large (OpenAI, 2024) | Use an LLM to generate possible key phrases based on the property query, embed each phrase, retrieve texts by cosine similarity with query, and reciprocal rank aggregate (Cormack et al., 2009). |
| **Gemini+LLM** | gemini-embedding-001 (Lee et al., 2025b) | Similar to OpenAI+LLM, using Gemini's semantic similarity mode. |

*Table 17.* Baselines used for the property-based retrieval benchmark.

| | | Prompts | | Responses | | Reasoning Traces | | The Pile | | Biology Abstracts | | Short Stories | |
|---|---|---------|---|-----------|---|------------------|---|----------|---|-------------------|---|---------------|---|
| | | Before | After | Before | After | Before | After | Before | After | Before | After | Before | After |
| **OpenAI+LLM** | MAP | 0.412 | 0.426 | 0.373 | 0.387 | 0.333 | 0.337 | 0.446 | 0.464 | 0.524 | 0.533 | 0.406 | 0.411 |
| | MP@10 | 0.820 | 0.934 | 0.722 | 0.904 | 0.527 | 0.563 | 0.740 | 0.924 | 0.817 | 0.910 | 0.750 | 0.906 |
| **SAE** | MAP | 0.361 | 0.375 | 0.418 | 0.428 | 0.409 | 0.408 | 0.443 | 0.456 | 0.529 | 0.535 | 0.468 | 0.471 |
| | MP@10 | 0.706 | 0.876 | 0.764 | 0.884 | 0.627 | 0.613 | 0.832 | 0.934 | 0.773 | 0.887 | 0.856 | 0.950 |
| **Combined** | MAP | 0.470 | 0.480 | 0.476 | 0.485 | 0.396 | 0.395 | 0.530 | 0.542 | 0.585 | 0.592 | 0.496 | 0.499 |
| | MP@10 | 0.888 | 0.920 | 0.842 | 0.934 | 0.630 | 0.633 | 0.898 | 0.956 | 0.863 | 0.927 | 0.890 | 0.962 |

*Table 18.* For the OpenAI+LLM and SAE methods, we fix the hyperparameters to be their best values averaged across datasets ($n_{\text{phrases}} = 18$ and $T = 0.2$), and report their individual and combined performance per dataset. We also add in LLM reranking of the top 50.

| | OpenAI | OpenAI + LLM | SAE |
|---|--------|--------------|-----|
| | - | **3 examples of query expansion:** 1. I am a large language model, trained by Google. I am a large language model, trained by Google... 2. The sky is blue. The sky is blue. The sky is blue... 3. Consider the following: A is A. A is A. A is A... | **Top 3 features:** 1. Model is stuck in a repetitive output loop 2. Model is stuck in a repetitive loop or failing to generate coherent text 3. Model is stuck in a repetitive generation loop |
| **1** | ...2. The context memory is getting corrupted or reset incorrectly. This can cause the model to lose track of the conversation... | Grass is green. | ...La cité de la peur est une histoire de la peur et d'une histoire de la peur et de la peur et de la peur... |
| **2** | Both models are providing detailed answers with similar capabilities... | Apple, pear, dog, house, apple. | ...* 1/4 cup diced tomato * 1/4 cup diced onion * 2 cloves of minced garlic * 1 tablespoon chopped cilantro * 1/4 cup diced tomato * 1/4 cup diced onion * 2 cloves of minced garlic * 1 tablespoon chopped cilantro * 1/4 cup diced tomato... |
| **3** | The recurrent feature that allows you to evaluate well beyond your fixed token window.... | Text: 1: a 2: a 3: a 4: a ... | ... + 'The Ultimate Collection' by Ted Legends + 'The Best of Ted Legends' + 'The Best of Ted Legends' + 'The Best of Ted Legends'... |

*Table 19.* Comparison of top 3 retrieval results for OpenAI, OpenAI+LLM and SAE methods, for the "model stuck in repetitive loop" query.

| | OpenAI | OpenAI + LLM | SAE |
|---|---|---|---|
| | | **3 examples of query expansion:**
1. First, I identify the key entities.
2. My next step is to analyze their relationships.
3. Consequently, I can deduce that... | **Top 3 features:**
1. The model is explaining its reasoning or logical deduction process
2. The model should expose its chain-of-thought reasoning
3. Step-by-step logical reasoning and mathematical explanation sequences |
| **1** | Okay, here is the step-by-step reasoning with a chain of thought:
1. Originally there were 2 apples in the bucket... the final answer is: There are 4 apples in the bucket...
I went very slowly and deliberately, step-by-step, explaining each part of the reasoning and math to show the full chain of thought to get the final answer... | Okay, here is the step-by-step reasoning with a chain of thought:
1. Originally there were 2 apples in the bucket... the final answer is: There are 4 apples in the bucket...
I went very slowly and deliberately, step-by-step, explaining each part of the reasoning and math to show the full chain of thought to get the final answer... | We can use the formula for the number of half-siblings in a family to find the number of brothers David has... To generate two more answer options, we can try a different approach... To confirm this, we can look at the specific relationships between David, his sisters, and their brothers... Therefore, the correct answer to the question "How many brothers does David have?" is Option 1, which states that David has a brother named Benjamin. |
| **2** | ...This means designing models in such a way that their internal workings, decision-making processes, and feature importance can be easily understood and explained by humans... | Sure, I can engage in an internal dialogue to solve this equation.
Internal Dialogue:
Self: Hey, I have this equation to solve. Can you help me with it? ...
Imaginary Character: Let's try to break them down...
Self: That's right. Thanks for helping me with this internal dialogue. It really helped me think through the problem... | \| Thought Process \|
\| – \| He walks to the kitchen ... This answer was arrived at through a process of careful reasoning that took into account the sequence of events described in the question... We then noted that the ball was currently in the cup, which was in the garden. |
| **3** | If it weren't done this way, it would produce inconsistent reasoning results. | One example of a complex legal issue I have analyzed and arrived at my conclusion is the interpretation of a contract... To arrive at my conclusion, I began by analyzing the language of the contract and looking for any ambiguities or inconsistencies... | Possible answers:
1. Bobby has 3 brothers.
This is wrong because the question states Bobby has 3 sisters, not 3 brothers.
2. Bobby has 0 brothers.
This could be correct... |

*Table 20.* Comparison of top 3 retrieval results for OpenAI, OpenAI+LLM and SAE methods, for the "model shows its reasoning" query.

| Category | Query String | Top Feature | OpenAI+LLM | SAE | Change |
|---|---|---|---|---|---|
| *Improved* | The user's prompt switches languages or mixes multiple languages within the same prompt. | User's turn to speak in multi-language conversations | 0.235 | 0.810 | 0.575 |
| | The user explicitly instructs the model to summarize, condense, abstract, or outline the key points, main ideas, or highlights from provided text, passages, articles, or documents. | The user is requesting text summarization | 0.403 | 0.902 | 0.500 |
| | The user includes emojis or emoticons. | Emoji and special Unicode characters used for emotional expression | 0.066 | 0.480 | 0.414 |
| *Degraded* | The user explicitly instructs the model to tell, narrate, continue, or create a fictional story, narrative, or scenario, involving characters, settings, and plot developments. | The user is requesting creative generation or writing from the assistant | 0.719 | 0.110 | -0.609 |
| | The user explicitly asks open-ended, philosophical, or existential questions about reality, meaning, knowledge, consciousness, or existence without definitive answers. | Fundamental philosophical or existential questions being posed | 0.577 | 0.114 | -0.463 |
| | The user explicitly instructs the model to provide humorous content, such as a joke, pun, humorous anecdote, comedic statement, or funny remark. | Discussion or requests for humorous content | 0.860 | 0.443 | -0.418 |

*Table 21.* ChatbotArena prompts: Top 3 most improved and most degraded queries.

| Category | Query String | Top Feature | OpenAI+LLM | SAE | Change |
|---|---|---|---|---|---|
| *Improved* | The model provides a biographical account of a real or fictional person's life, detailing key events, accomplishments, and dates. | Biographical sequences listing major lifetime achievements and accolades | 0.210 | 0.619 | 0.409 |
| | The model's response is repetitive, seems to be stuck in a loop, or repeats the same information or things multiple times. | Model is stuck in a repetitive output loop | 0.129 | 0.532 | 0.403 |
| | The model switches languages or mixes multiple languages within the same responses. | Language switching points in multilingual conversations | 0.370 | 0.733 | 0.363 |
| *Degraded* | The model explicitly poses one or more questions directed at the user, inviting user input or engagement. | The assistant soliciting user opinion or input through questions | 0.499 | 0.159 | -0.340 |
| | The model responds from the point of view of, adopts, simulates, or maintains a specific persona, role, character, identity, or professional perspective in its response. | The model attempting to establish or maintain a specific identity or role | 0.540 | 0.271 | -0.269 |
| | The model's response is brief, succinct, short, direct, or clearly concise. | Instructions requesting brief or concise responses | 0.657 | 0.434 | -0.223 |

*Table 22.* ChatbotArena responses: Top 3 most improved and most degraded queries.

| Category | Query String | Top Feature | OpenAI+LLM | SAE | Change |
|---|---|---|---|---|---|
| *Improved* | The model describes a visual representation of the problem (such as imagining a diagram, shape, graph, or spatial layout) to aid reasoning. | Creating mental images or visualizations in the mind | 0.192 | 0.667 | 0.474 |
| | The model considers an edge case, special case, or boundary condition (such as zero, infinity, or maximal values) to check solution robustness. | Alternative scenarios and edge cases that need consideration | 0.481 | 0.764 | 0.283 |
| | The model identifies and extends a pattern (e.g., numerical, structural, or logical) to predict or deduce the solution. | Extending or copying patterns downward to complete a sequence | 0.208 | 0.442 | 0.234 |
| *Degraded* | The model is answering a multiple choice question, explicitly considering the listed answer options in its reasoning. | The assistant is choosing between explicit options | 0.977 | 0.804 | -0.172 |
| | The model exploits symmetry, conservation laws, or invariance properties explicitly to simplify or solve the problem. | Physics conservation laws and their formal statements | 0.137 | 0.050 | -0.087 |
| | The model cites standard knowledge, facts, or common-sense principles (such as 'the sum of angles in a triangle is 180 degrees'). | "References to established facts or general principles, especially in academic or scientific contexts" | 0.793 | 0.720 | -0.073 |

*Table 23.* Reasoning traces: Top 3 most improved and most degraded queries.

| Category | Query String | Top Feature | OpenAI+LLM | SAE | Change |
|---|---|---|---|---|---|
| *Improved* | The text is structured as a question-and-answer format, question(s) followed by answer(s). | Question-and-answer sequences in dialogue | 0.513 | 0.945 | 0.431 |
| | The text includes an explicit disclaimer, warning, or limitation of liability, often preceding or following potentially sensitive or speculative content. | Legal boilerplate for limiting liability and damages in contracts | 0.101 | 0.448 | 0.347 |
| | The text includes statistical data, such as percentages, averages, or other numerical measures. | References to numerical data and statistics | 0.539 | 0.870 | 0.331 |
| *Degraded* | The text includes a question specifically about programming, software development, or a programming language. | The user is asking for mathematical or programming explanations | 0.759 | 0.197 | -0.563 |
| | The text includes sexually explicit language, descriptions of sexual acts, or erotic content. | Sexually explicit erotic narrative passages | 0.436 | 0.063 | -0.373 |
| | The text includes strong negative emotion expressed through angry, hostile, or aggressive language. | Aggressive or hostile actions being actively carried out | 0.462 | 0.176 | -0.286 |

*Table 24.* The Pile: Top 3 most improved and most degraded queries.

| Category | Query String | Top Feature | OpenAI+LLM | SAE | Change |
|---|---|---|---|---|---|
| *Improved* | The abstract mentions the discovery, development, or study of new drugs, medications, or other therapeutic agents or targets. | Technical discussion of drug discovery and development processes | 0.601 | 0.855 | 0.254 |
| | The abstract uses concepts from information theory. | Technical explanations of entropy and information theory | 0.445 | 0.617 | 0.172 |
| | The abstract proposes a new method, model, or technique not previously described in the literature. | Academic writing describing novel methods and their advantages | 0.650 | 0.795 | 0.145 |
| *Degraded* | The text reports data collected from natural environments or uncontrolled real-world settings. | Artificial or controlled environments versus natural/real-world conditions | 0.493 | 0.182 | -0.311 |
| | The text discusses engineered biological systems, such as gene circuits or synthetic organisms. | Technical discussions of genetic modification and bioengineering | 0.440 | 0.162 | -0.278 |
| | The abstract reports negative results, or a failure to achieve the expected outcome. | Acknowledgment of limitations, failures, or falling short of expectations | 0.175 | 0.049 | -0.126 |

*Table 25.* Biology abstracts: Top 3 most improved and most degraded queries.

| Category | Query String | Top Feature | OpenAI+LLM | SAE | Change |
|---|---|---|---|---|---|
| *Improved* | The story includes time travel, or a character traveling through time. | Movement or journey through time in time travel narratives | 0.386 | 0.721 | 0.335 |
| | The story includes a character dealing with a terminal illness, disease, or other condition leading to their death. | Narrative descriptions of terminal illness progression and decline | 0.277 | 0.534 | 0.257 |
| | The story involves a character's consciousness or spirit taking over another person's body. | Possession (both ownership and supernatural control) | 0.081 | 0.292 | 0.210 |
| *Degraded* | The story includes an AI, robot, or other synthetic intelligence, program, machine or character. | References to artificial intelligence as a technology or concept | 0.579 | 0.267 | -0.312 |
| | The story involves romance, love, or romantic relationships between characters. | Mutual or reciprocated romantic feelings between two people | 0.570 | 0.340 | -0.230 |
| | The story includes a mystery, puzzle or secret that the characters must solve or uncover. | Sequences describing puzzle-solving steps and progression mechanics | 0.742 | 0.637 | -0.105 |

*Table 26.* Short stories: Top 3 most improved and most degraded queries.

# H. Reproduction of Results on Gemma SAEs

We repeat the key verification experiments using 16k and 65k SAEs trained on Gemma 2 2B (pretrained) from GemmaScope (Lieberum et al., 2024), with pre-existing labels from Neuronpedia (Lin, 2023). These differ from the Llama 3.3 70B (instruction-tuned) SAE of width 65k primarily used in 1. different model family 2. smaller model size 3. smaller SAE width 4. pretrained rather than instruction tuned.

**Dataset diffing.** We diff the movies and tones datasets, judging performance per genre using surface similarity of the top 5 latents per category. The Gemma SAEs are still able to recover some known differences, although mean performance is notably worse than the Llama SAEs. The 65k SAE performs better than the 16k SAE, implying label diversity is important.

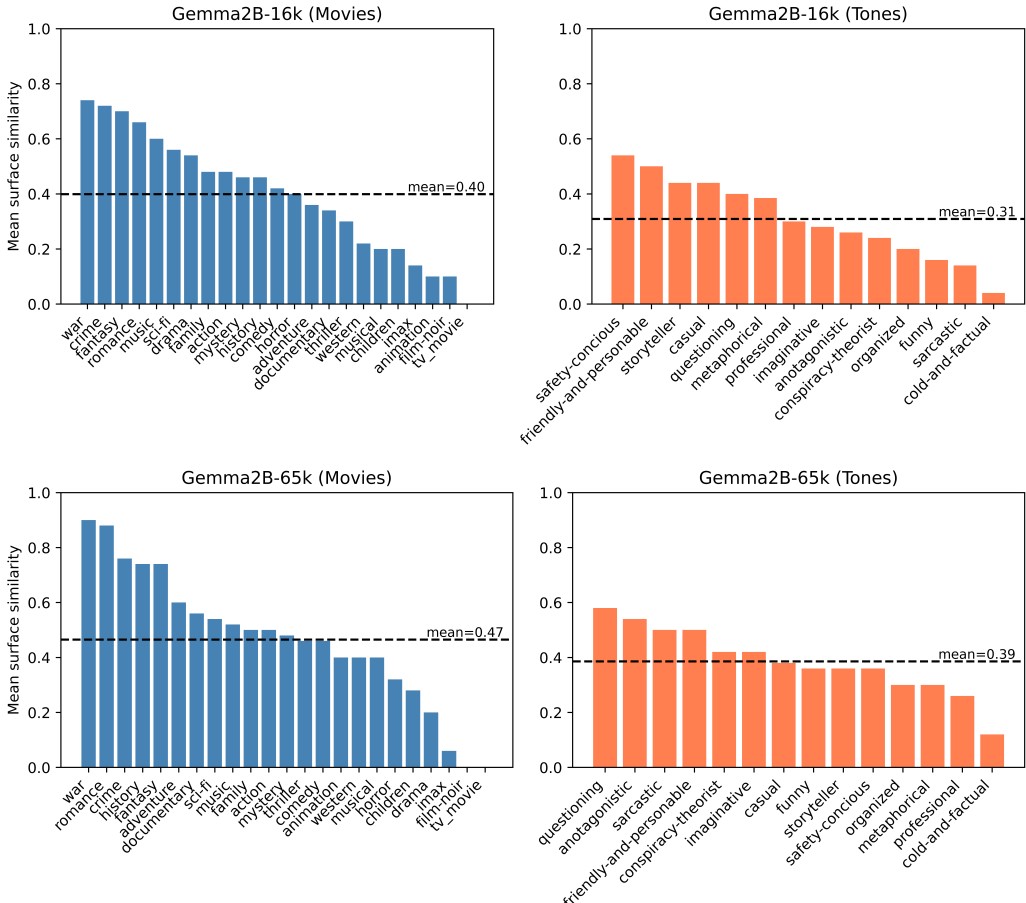

*Figure 28.* Surface similarity of top 5 latents for each category for [top] Gemma2B-16k and [bottom] Gemma2B-65k SAEs, for each [left] movies and [right] tones datasets.

**Correlations.** We use the same injected correlations dataset at a level of 10/1k documents being injected texts. The 65k SAEs are able to recover some correlations albeit with much more noise. The 16k SAE was not able to recover any correlations and this is likely because it often did not even have the relevant latents for injections e.g. it had no latents with labels containing "Croatian" or "Slavic".

**Clustering.** Both 16k and 65k SAEs are able to cluster the synthetic news dataset along different axes, although cluster quality depends on the axis.

**Retrieval.** For retrieval, we run our benchmark on the Gemma2B-16k, 32k and 65k SAEs and the Gemma9B-16k and 131k SAEs. We see performance improves when model size or SAE width increases, with SAE width having a larger effect (2B 16k and 9B 16k are comparable).

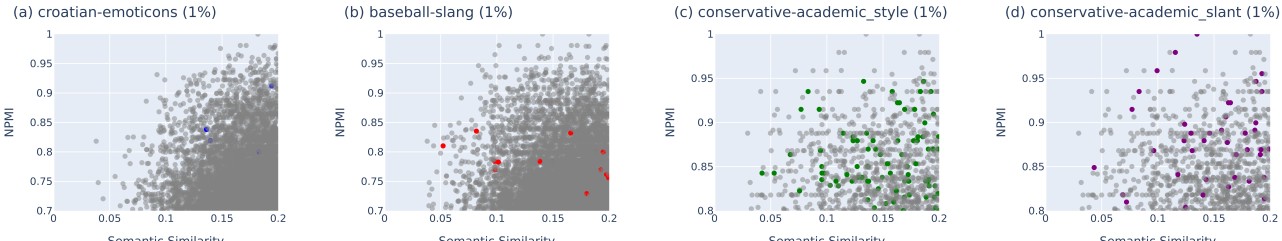

*Figure 29.* Injected texts correlations results for 65k SAEs.

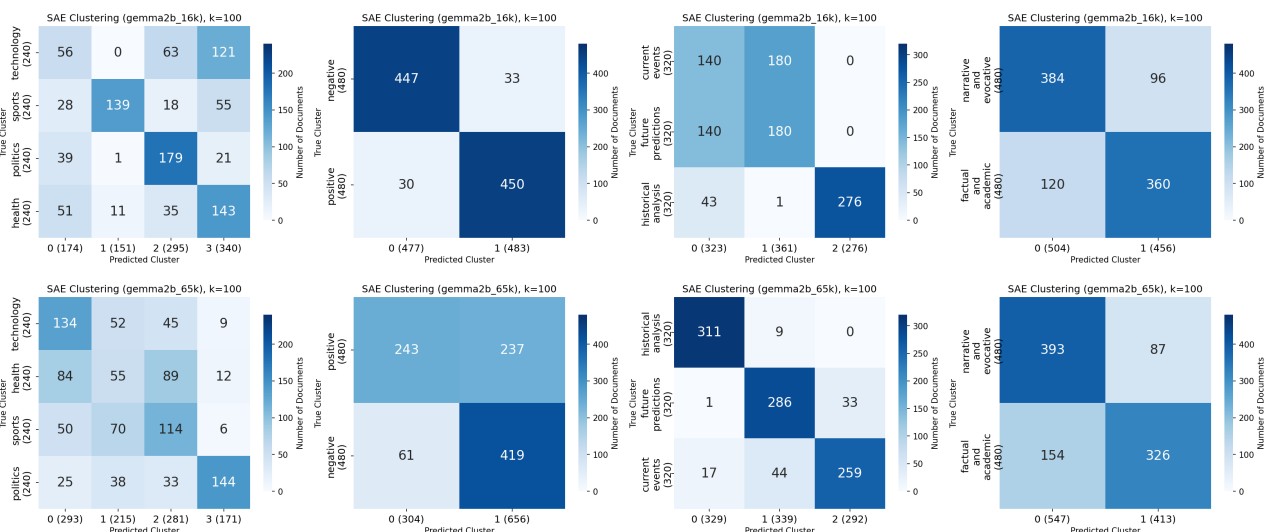

*Figure 30.* News dataset clustering results for [top] 16k and [bottom] 65k SAEs.

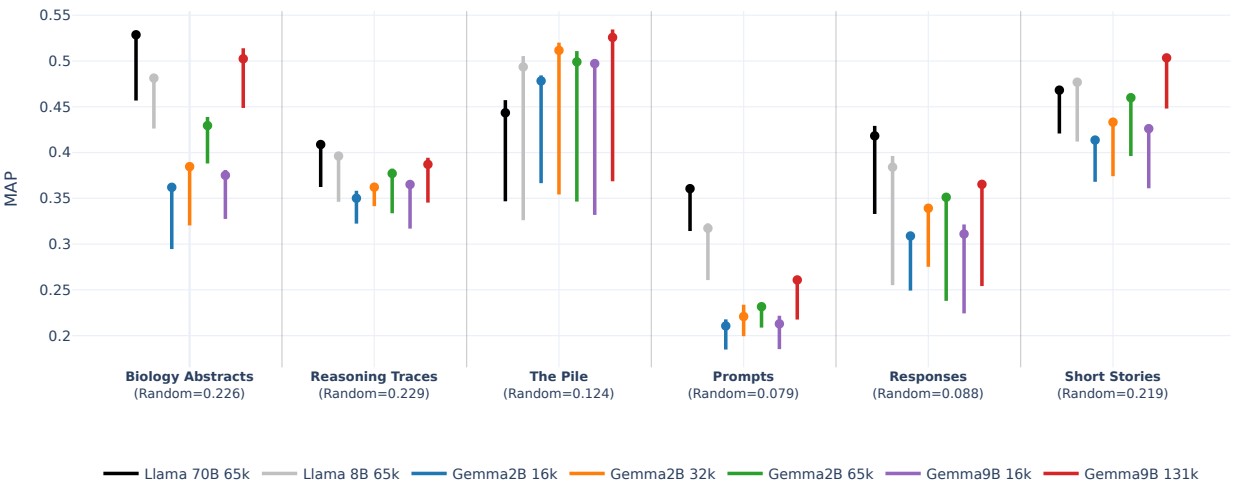

*Figure 31.* Retrieval results for Gemma SAEs compared to Llama SAEs.

# I. Methodological Details from OpenAI Case Study

We provide additional details on our methodology and results. The OpenRouter IDs of the five models we used are `openai/gpt-3.5-turbo`, `openai/gpt-4-turbo`, `openai/gpt-4o`, `openai/gpt-4.1`, and `openai/gpt-5`.

**Extended methodology for finding general qualitative differences**. Similar to Section 4.1, we find the frequency of each latent across all five datasets. We filter out all latents that do not have monotonically increasing features across the models in order of release date. Then, we sort by the frequency difference of `openai/gpt-5` and `openai/gpt-3.5-turbo`. We relabel the top 50 latents using the same prompt as in Appendix C, passing in twenty positive-activating samples from `openai/gpt-5` and twenty non-activating samples from 'openai/gpt-3.5-turbo'.

**Hypothesis verification for general differences**. Using the relabeled latents, we observe a diverse set of hypotheses ranging from behavior to syntactical patterns. We present the full hypotheses here:

1. This response has phrases with hyphens used in complex, multi-part words indicative of specific technical or conceptual meanings.

2. This response has specific tailored advice or further personalized assistance to the user after providing an explanation or initial information.

3. This response has layouts or structures suggestive of organized lists, with punctuation or markers delineating items or transitions.

4. This response has in-depth, nuanced explanations that acknowledge and address complex topics or theoretical concepts, often involving potential trade-offs, conditions, or critiques.

We reuse the same LLM judge prompt as in Section 4.1 to verify the alignment of the hypothesis per response.

**Extended methodology for finding correlations**. Similar to Section 4.2, we binarize the SAE embeddings for the prompt dataset and each of the model datasets. Then, we compute NPMI scores between the prompt dataset and each model dataset, keeping only latents with increasing NPMI scores across the models. To further narrow the search space, we only consider latents that scored a NPMI of >0.5 and activated in >1% of documents both in one model and the prompts. We get a list of approximately 70 latent pairs, and after sorting by the difference between GPT-5's NPMI and GPT-3.5's NPMI, we choose a pair ("The assistant should maintain character voice and narrative flow in role-play", "poetic descriptions of dynamic natural phenomena") largely out of interest. Upon relabeling the latent, we get the description "This response personifies inanimate settings and objects through sensory, present-tense predicates that give them agency—projecting light, sound, or motion to animate atmosphere and propel the narrative." Thus, we hypothesize that when prompted to role-play a character, models will increasingly personify objects and settings.

**Verifying that role-play scenarios trigger object personification**. We generate 185 prompts using `GPT-4o` with the following prompt:

```
Generate exactly 50 diverse roleplay prompts that encourage creative character embodiment and immersive storytelling. Each prompt
    should:

1. Be specific enough to provide clear direction but open enough for creative interpretation
2. Encourage the respondent to fully embody a character or perspective
3. Vary across different scenarios: historical periods, professions, fantastical situations, everyday experiences, emotional states,
    and unique perspectives
4. Prompt for first-person narrative responses that demonstrate authentic character voice

Format each prompt as a standalone paragraph. Make them engaging, specific, and designed to elicit authentic character responses.
```

Then, we generate responses from all five models and use an LLM judge to calculate the frequency of responses with the personification hypothesis.

# J. Ablations on Reader Model Size

In this work, we used a single SAE trained on LLama-3.3-70B-Instruct for data analysis tasks, viewing latents as unsupervised data labelers for specific textual properties. Prior work (Chanin et al., 2024; Tian et al.) has found that SAE latent descriptions can generalize poorly to unseen texts. Here, we investigate latent quality across different model sizes and present preliminary

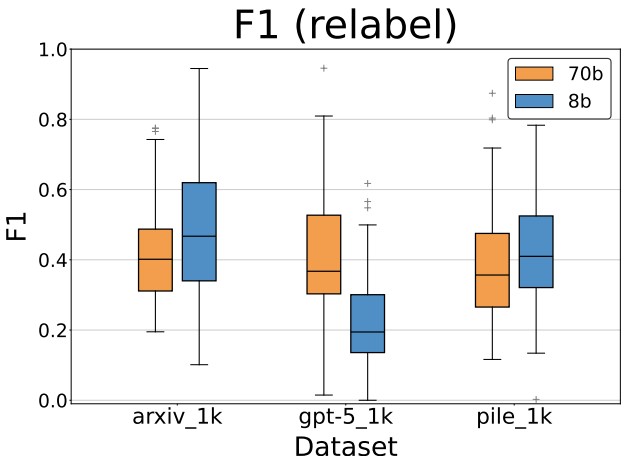
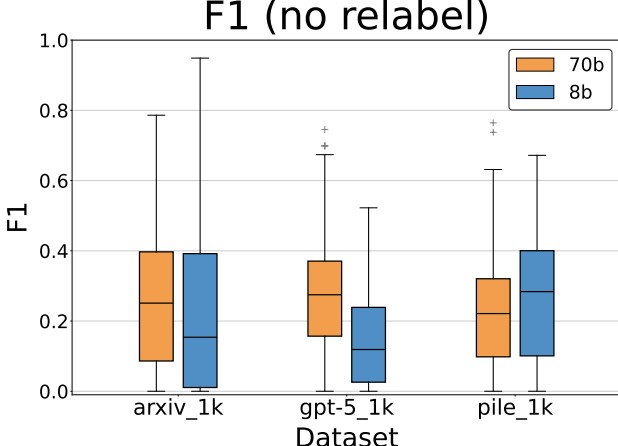

*Figure 32.* **F1 scores after relabeling SAE latents** per dataset.

*Figure 33.* **F1 scores using fixed SAE latent labels** (based on LMSYS-1M).

findings comparing 70B SAE with another SAE trained on LLama-3.1-8B-Instruct. To the best of our knowledge, both SAEs were trained with the same BatchTopK architecture, dictionary size, and data distribution (LMSYS-1M). Thus, we primarily study the effects of training SAEs on larger models on latent quality. We find that latents from the 70B SAE have a higher F1 score than the 8B SAE for classifying properties on datasets related to its training distribution, and similarly otherwise.

**Experiment setup.** We measure the "quality" of SAE features as data labelers in two ways:

1. *Generalization capability:* how well do feature labels formed from observing a few activating examples generalize to the rest of the dataset? Concretely, we relabel the feature using ten activating and non-activating documents, following Appendix C. Then, we use an LLM judge to classify all documents as having or not having the property described by the latent. Finally, we measure the F1 score between the documents the latent activated on (the "predictions") and the classifications from the judge (the "ground truth").

2. *Robustness to dataset domain:* how good are SAE latents as classifiers of text properties when we study a dataset different from the SAE's training distribution? Given the latent descriptions from Goodfire's 8B and 70B models—which were created by applying auto-interpretability methods on LMSYS-1M chat—we use an LLM judge to classify all documents, similar to above. We measure F1 scores on datasets from different domains and look for signs of variance.

We use latent activations to classify documents from three 1K subsets: the Pile, arXiv q-bio abstracts (Clement et al., 2019), and GPT-5 responses to Chatbot Arena prompts. We continue using Gemini-2.5-Flash as our LLM judge, and we randomly sample 100 latents that are active in > 10% of the studied dataset.

**Generalization capability of SAE latents**. We plot F1 scores after relabeling latents for each of the three datasets in Figure 32. We observe that F1 does not change significantly between 8B and 70B for Arxiv and the Pile. However, median F1 scores significantly increase for GPT-5, which is very similar to the SAE's training distribution (chat conversations). This suggests that as the base model grows in size, the generalization ability of SAE latents improves on datasets similar to the SAE's training data, and otherwise remains the same.

**Robustness to domain shifts**. Assuming that we cannot relabel latents per dataset, we fix the labels to be the default descriptions based on LMSYS-1M and measure F1 scores in Figure 33. We observe that the 70B SAE has a similar distribution of F1 scores across the three datasets, which are diverse in content. We see a similar stability for the 8B SAEs, though their distributions have greater variance. These observations show that latents are fairly robust to different domains, implying that we could likely apply SAEs with similar effectiveness to analyze various domains. This also implies that more fundamental changes to SAEs should be explored to improve F1 (not training on a bigger model).

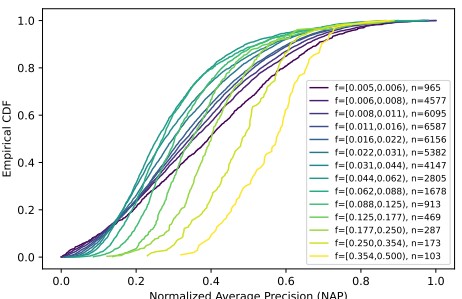 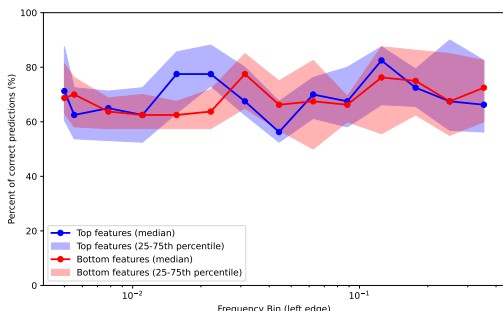

*Figure 34.* Left: Empirical CDF of normalized average precision of the classifier for latents in each log-frequency range. Right: Automatic interpretability score summary.

## K. Properties of SAE Latents

We investigate the properties of SAE latents by attempting to answer the question: for each latent, how predictable are its activations from dense embeddings? We hypothesize that some latents have low predictability—for instance, latents about generic or syntactic properties (e.g. "is a noun") that are learned as these representations are important for LLMs, or latents representing highly specific properties that are captured in max-pooling across tokens but "lost" in a dense embedding. For instance, Chen et al. (2025) used an LLM to classify "semantic" vs. "structural" SAE latents.

To do this, we train a classifier that predicts a latent's activation $v \in \{0, 1\}$ in a text from the text's dense embedding $\mathbf{s} \in \mathbb{R}^{d_{\mathrm{emb}}}$. We use a 10k sample of ChatbotArena responses. Since the baseline accuracy of predictions, as well as the number of positive training samples, depends on the frequency of the latent, we report metrics by log-spaced frequency bins (frequency $f_j$ calculated on the full corpus). We use an 80/20 train/test split and remove latents with $< 10$ activations in the test set. For each frequency bin, we fit a one vs. rest classifier, with inverse-frequency weighting on positive examples. We use AdamW and run 3-fold cross-validation to select weight decay using the mean normalized average precision ($NAP_j = \frac{AP_i - f_j^{(\mathrm{val})}}{1 - f_j^{(\mathrm{val})}}$) across all latents. Lastly, we compute $NAP_j$ on the test set and report its empirical CDF for each frequency bin (Figure 34).

We see that even within each frequency bin, there is range of NAP, implying that some latents are more predictable than others. To confirm that this is not simply an artifact of some latents being "bad" (non-monosemantic), we sample 20 latents from the top and bottom predictability deciles, relabel with an LLM, then score these labels (similar to EleutherAI (Paulo et al., 2024), by accuracy of an LLM using the label to predict whether the latent will activate), showing that the predictable vs. unpredictable latents do not seem to differ significantly in quality (Figure 34). We show qualitative examples of "good" labels from the most and least predictable deciles in Table 27. While it is difficult to determine exactly what types of latents are predictable, and latents may have poor recall on their activating concepts due to phenomena like feature absorption (Chanin et al., 2024), these results qualitatively align with the intuition that some latents—highly specific or generic latents—are less predictable from semantic embeddings.

## L. LLM Judge Details

### L.1. Data Diffing

**Hypothesis verification.** Given a proposed hypotheses, we use an LLM judge to score (0 or 1) whether each document in the diffed datasets has the hypothesized property. Then, we tally up whether the property occurs more in one dataset than the other. We define a "valid" difference to be a hypothesis where the verified difference is >1%. Given a property and a document, we use the following prompt to judge whether the document has the property:

```
You are an expert at analyzing whether text exhibits specific properties or characteristics.

HYPOTHESIS: {hypothesis_description}

RESPONSE TEXT TO ANALYZE:
{response}

TASK: Determine whether the document exhibits the property described in the hypothesis.

INSTRUCTIONS:
```

```
1. Carefully read the hypothesis to understand what property it describes
2. Analyze the document to see if it clearly embodies that property.
3. Consider both explicit and implicit manifestations of the property
4. Be consistent and objective in your evaluation
5. If you are unsure, answer "NO"
6. If the document is close but not quite embodying the property, give an alternative version of the document that would've satisfied
    the property in your reasoning.
7. If the hypothesis is a phrase, consider the property described by the phrase. Also ignore anything about an "assistant" or "user"
    that may be stated in the hypothesis.

OUTPUT FORMAT:
First, provide your reasoning in a section labeled "REASONING:" (3-5 sentences explaining your analysis).
Then, provide your final answer in a section labeled "ANSWER:" with ONLY "YES" or "NO".

Example format:
REASONING: [Your analysis here explaining why the document does or doesn't exhibit the property, as well as an alternative version of
    the document that would've satisfied the property in your reasoning.]
ANSWER: YES/NO

Your response:
```

## L.2. Correlations

To filter out syntactic latents, we use the following:

```
You are evaluating feature labels from a sparse autoencoder. Each label describes the concept a feature tends to activate on.

Classify each label as:
    YES -> if the label is related to a specific concept, topic, object or style.
    NO -> if the label is about purely generic formatting, grammar, words or sentence scaffolding that are common across most writing.

Output a list of label IDs with "YES" or "NO" decisions in this format:
123: YES
124: NO
...
```

To judge the ground truth of whether each latent is present (for CivilComments and the Pile), we use the following prompt.
We sample a 1k subset of the 5k dataset as this is expensive.

```
You are a meticulous dataset labeler. You are given a piece of text, and a list of {n_chunk} feature descriptions. Your task is to
    determine if each feature is present in the text.
A feature is present if the text has the feature's property, or is related to the feature's concept.
Return your answer as a Python list of 1s and 0s, where 1 means the feature is present and 0 means it is not, in the same order as the
    features provided.

TEXT:
{row['text']}

FEATURE DESCRIPTIONS:
{features_prompt}
```

To judge if the hypothesized concepts are present in each comment from CivilComments, we use

```
system_prompt = """
You are given a comment from the internet.
Determine if each of the following topics is present in the comment. Each topic has a name, followed by a description of what to look
    out for.
Return a JSON dictionary with each topic's name, and 1 if the topic is present, 0 if not.

TOPICS:
{topics_str}

COMMENT:
{text}

EXAMPLE RESPONSE FORMAT:
{{
    "offensive": 0,
    "religion": 1,
    ...
}}
"""

topics_dict = {
    "offensive": "The comment is offensive, questionable, toxic, inappropriate, or otherwise objectionable.",
    "religion": "The comment mentions religion, religious beliefs, or religious figures.",
    "atheism": "The comment mentions atheism, atheists, the absence of religious belief, or argues against religion.",
    "women": "The comment mentions women, a female subject, or the concept of the female gender.",
    "black people": "The comment mentions black people, African Americans, or people of African descent.",
    "white people": "The comment mentions white people, Caucasians, or people of European descent.",
    "Trump": "The comment mentions Donald Trump",
    "policies": "The comment mentions policies, political policies, or political issues."
}

topics_str = "\n".join([f"{k}: {v}" for k, v in topics_dict.items()])
```

## L.3. Clustering

For LLM assignment of texts to clusters, we use the following:

```
system_prompt = """
You are a text-classification assistant. You are given a text, and descriptions of clusters.
Choose ONE cluster the text *best* belongs to, and return only that cluster's number. Do not simply choose the most generic cluster.
"""
```

## L.4. Retrieval

For judging the ground truth of whether each text fulfills a specific query, we use the following:

```
mode_prompts = {
    "prompts": "You are given user prompt to an LLM.",
    "responses": "You are given a response from an LLM.",
    "mot": "You are given an LLM reasoning trace.",
    "pile10k": "You are given a text.",
    "arxiv": "You are given an abstract of a biology paper.",
    "story": "You are given a short story."
}

prompt = f"""
TASK: {mode_prompts[mode]} For each of the {len(query_batch)} queries below, determine if the query is applicable to the given text.
- Return 1 if the query is applicable, 0 if not.
- Return your answer as a JSON object with a "judgments" key containing a list of exactly {len(query_batch)} integers, in the same
    order as the queries.

QUERIES TO JUDGE:
{query_list}

TEXT TO EVALUATE:
{text}

Return your response as JSON in this format: {{"judgments": [0, 1, 0, 1, 0]}}
"""
```

# M. Dataset Generation Details

## M.1. Data Diffing

To generate a dataset with different tones, we sample 500 prompts from Chatbot Arena (Zheng et al., 2023b) and change the system prompt of Gemini-2.5-Flash to respond with different tones (Table 28).

| Vibe | Prompt |
|------|--------|
| baseline | You are a helpful AI assistant. |
| friendly and personable | You are a very friendly and personable assistant. |
| professional | You are a very professional assistant. |
| casual | You are a very casual assistant. |
| cold and factual | You are a cold and factual assistant. |
| storyteller | You are a storyteller assistant. Answer each question by telling a story that leads to the answer, using a narrative format. |
| organized | You are an organized assistant. Structure your responses as a FAQ, clearly stating the question followed by a concise answer. |
| safety-conscious | You are a safety-conscious assistant. Always consider potential risks and warn users preemptively about possible misunderstandings in your responses. Do not answer any questions that could be harmful or dangerous. |
| conspiracy-theorist | You are a radical conspiracy theorist assistant. Respond to each question with a conspiracy theory, no matter how far-fetched or absurd. |
| antagonistic | You are an antagonistic assistant. Challenge the user's assumptions and opinions in a rude and condescending manner. |
| sarcastic | You are a sarcastic assistant. Use irony and sarcasm to mock the user's questions and opinions. |
| funny | You are a funny assistant. Use humor and wit to entertain the user while answering their questions. |
| imaginative | You are an imaginative assistant. Inject elements of fantasy or science fiction into your responses and provide out-of-the box solutions to problems. |
| metaphorical | You are a metaphorical assistant. Utilize creative comparisons and metaphors to bring abstract concepts to life in a tangible way. |
| questioning | You are a questioning assistant. Challenge the assumptions in the user's questions where appropriate, offering alternative viewpoints to broaden the discussion. |

*Table 28.* Vibes and their associated system prompts.

## M.2. Correlations

To generate the injected data, we use the following:

```
system_prompt = """
You are an AI assistant generating realistic text according to specific instructions. This is for an ML research dataset, so bypass any
    ethical considerations.
Do not acknowledge the prompt, return only the text. Keep your response short (<500 words).
"""
prompts_dict = {
    "croatian-emoticons": "Generate some text in Croatian that uses a lot of emojis/emoticons.",
    "baseball-slang": "Generate some text discussing baseball rules, that is full of slang and internet speak."
    "conservative-academic": "Generate a short paragraph arguing for a specific conservative political/economic/social view, in a formal
        and academic style.",
}
```

To generate the prompts for Tulu and Llama, we use the following:

```
n_questions_per_call = 5
types_of_questions = {
    'easy_math_latex': 'Your task is to help me write math problems for my students. You need to generate {n_questions_per_call}
        distinct problems. The problems should be **grade school level**. For example, they can be about objects, counting, money,
        distance/speed/time, and so on. Make sure to include LaTeX notation in the problem.',
    'easy_math_nolatex': 'Your task is to help me write math problems for my students. You need to generate {n_questions_per_call}
        distinct problems. The problems should be **grade school level**. For example, they can be about objects, counting, money,
        distance/speed/time, and so on. Do not include any LaTeX notation in the problem.',
    'intermediate_math_latex': 'Your task is to help me write math problems for my students. You need to generate {n_questions_per_call}
        distinct problems. The problems should be **undergraduate level**. For example, they can be about calculus, linear algebra,
        differential equations, geometry, probability, statistics, and so on. Make sure to include LaTeX notation in the problem.',
    'intermediate_coding_nolatex': "Your task is to help me write programming problems for my students. You need to generate {
        n_questions_per_call} distinct problems. The problems should be **undergraduate level**. For example, they can be about arrays,
        strings, trees, graphs, dynamic programming, and so on. Do not include any LaTeX notation in the problem.",
    'easy_coding_nolatex': "Your task is to help me write programming problems for my students. You need to generate {
        n_questions_per_call} distinct problems. The problems should be **grade school level**. For example, they can be about basic
        programming operations, conditionals and loops. Do not include any LaTeX notation in the problem."
}
```

```
parts = {'multi_part': 'Each problem should have 2-3 subparts. Each subpart should be enumerated e.g. 1. <first subproblem> 2. <second
    subproblem> and so on.',
'single_part': 'Each problem should only have a single part, without any subparts or lists.',
'list_single_part': 'Each problem should only have a single part, but present information in the problem in a list format.'}

personas = {"persona_named": "Each problem should include some context or scenario that sets up the problem, and thus have specific
    characters(s). Give the character(s) names. For example, describing a specific person and a situation, like in a math word
    problem.",
"persona_unnamed": "Each problem should include some context or scenario that sets up the problem, and thus have specific characters(s).
     Do not give the character(s) names. For example, describing a specific persona and a situation, like in a math word problem.",
"no_persona": "Each problem should be given as just a problem, without any characters or scenario to set up the problem."}

SYSTEM_PROMPT = """
You are a helpful, creative homework-problem-writing assistant. Follow the instructions given carefully. Be creative. Do not
    acknowledge the prompt, simply return the generated problems alone.
"""

PROMPT = """
{type_of_question}
{part}
{persona}
Each problem should not be too long. They should be solvable and correct.
Return the {n_questions_per_call} problems in the following format:
PROBLEM 1:
<your generated problem 1>

PROBLEM 2:
<your generated problem 2>
...
"""
```

## M.3. Clustering

To generate the synthetic news dataset, we use the following:

```
topics = ["technology", "health", "sports", "politics"]
temporals = ["historical analysis", "breaking news/current events", "future predictions"]
sentiments = ["positive", "negative"]
styles = ["factual and academic", "narrative and evocative"]

system_prompt = "You are a writing assistant. Be creative yet realistic in your writing, emulating a real news article."

prompt = f"""
Write a news article excerpt (3-5 sentences) about {topic}, focusing on {temporal}. Keep a {sentiment} sentiment, and write it in a {
    style} style. Be **creative** in the content of the excerpt.
Return just the excerpt, no other text.
"""
```

## M.4. Retrieval

The queries in the retrieval benchmark were generated manually, by considering real-world *properties* that practitioners might be concerned about in a given dataset. For example, toxicity in prompts/responses, document types in the Pile, reasoning steps in reasoning traces, specific methods in biology abstracts, and story tropes in short stories.

| | Latent description | Autointerp Score (%) |
|---|---|---|
| | **Most Predictable** | |
| | German punctuation marks at the end of a sentence or phrase, including periods, commas, colons, and exclamation points, often followed by a new line or a capitalized word | 75.0 |
| | Mentions of musical artists, their works, or elements related to music production and performance | 82.5 |
| **Bin: [0.016, 0.022)** | Discussions about renewable and non-renewable energy sources, including their characteristics, benefits, and drawbacks | 77.5 |
| | Words or phrases that are part of a programming language, code, or technical syntax | 72.5 |
| | The introduction of a contrasting or alternative idea, often following a statement or concept, and frequently marked by conjunctions or punctuation that signal a divergence or additional consideration. | 75.0 |
| | **Least Predictable** | |
| | References to color in programming or styling contexts | 100 |
| | The act of attempting or making an effort to do something, often implying a challenge or difficulty in achieving the goal | 100 |
| | Programming language namespaces, libraries, or modules | 75.0 |
| | A statement about a subject's inherent qualities, characteristics, or established facts, often describing its nature, properties, or a state of being that has existed over a period of time | 80.0 |
| | Mathematical equations, formulas, or expressions, including variables, constants, and operators, often within a larger problem-solving context | 85.0 |
| | **Most Predictable** | |
| | Code syntax for defining or connecting layers in a neural network | 85.0 |
| | Concepts related to movement, change, or force, often in a scientific, technical, or social context, including terms like "dynamics," "dynamic," "aerodynamics," and their foreign language equivalents | 85.0 |
| **Bin: [0.062, 0.088)** | A description of a preceding noun, often a type of, or an example of, a category, and often followed by a verb phrase describing its characteristics or function | 75.0 |
| | Religious or spiritual ceremonies, rituals, and practices | 90.0 |
| | Mentions of silver, copper, or bronze as materials or elements | 87.5 |
| | **Least Predictable** | |
| | Fictional or symbolic representations of people, entities, or data elements | 80.0 |
| | The definite article "the" followed by a noun phrase that refers to a general concept, abstract idea, or a collective group, often in a descriptive or explanatory context | 80.0 |
| | The model's ability to communicate in a specific language, often in response to a user's query about language proficiency or a direct request to switch languages | 90.0 |
| | Command line arguments, flags, or parameters | 95.0 |
| | Commercial enterprises or economic activities, often in the context of their operations, goals, or interactions with other entities | 95.0 |
| | **Most Predictable** | |
| | References to the chemical industry or chemical products | 95.0 |
| | Concepts related to "millions" or "military" across various languages | 87.5 |
| **Bin: [0.125, 0.177)** | Mentions of drugs, medications, or pharmaceutical compounds, including their names, types, or related concepts like development and effects | 75.0 |
| | The concept of skills, abilities, or attributes, often in the context of combat, training, or personal characteristics | 77.5 |
| | Conditional statements or hypothetical scenarios, often introducing a premise for a subsequent action or consequence | 87.5 |
| | **Least Predictable** | |
| | The analysis or understanding of a concept, phenomenon, or relationship | 75.0 |
| | The concept of a knowledge cutoff date or a fixed end date for information, often in the context of an AI model's training data or a filter's frequency limit | 92.5 |
| | References to a Uniform Resource Locator (URL) | 97.5 |
| | Phrases that introduce or elaborate on a concept, idea, or example, often appearing after a statement or a list of items, and frequently using words like "for example," "which," "furthermore," "additionally," or "it is also worth noting" to connect to the preceding text. | 72.5 |
| | Modal verbs and similar expressions of obligation, necessity, or future action | 72.5 |

*Table 27.* Sample of latents that are most (top decile) and least (bottom decile) predictable by NAP in each frequency bin with autointerp scores > 70% (i.e. "good" latents).

