# OpenReview forum: "Interpretable Embeddings with Sparse Autoencoders: A Data Analysis Toolkit"
_ICML.cc/2026/Conference — ICML 2026 regular_

### Official Review · Reviewer_vJc7 · 2026-03-08

**Soundness:** 3
**Presentation:** 3
**Significance:** 3
**Originality:** 3
**Overall Recommendation:** 5
**Confidence:** 4

**Summary:**

This paper proposes using sparse autoencoders (SAEs) to construct interpretable embeddings for text analysis. By applying a pretrained SAE to hidden states from a reader language model and aggregating token-level activations, the method produces document-level representations where each dimension corresponds to a human-interpretable concept. The authors show that these embeddings can support several data analysis tasks, including dataset diffing, concept correlation discovery, controllable clustering, and property-based retrieval. Experiments compare the approach with LLM-based analysis and dense embeddings, suggesting that SAE-based representations can identify dataset differences and correlations more cost-effectively while providing interpretable and controllable features.

**Compliance With Llm Reviewing Policy:**

Affirmed.

**Final Justification:**

After considering both the paper and the authors' rebuttal, I have decided to increase my score from 4 (Weak Accept) to 5 (Accept). The authors have addressed my previous concerns by providing additional experimental data, resolving the issues related to the limited methodological novelty and the strong dependence on a specific pretrained SAE.

Key Points of Adjustment:

Effective Response: The authors demonstrated the robustness of the SAE approach through experiments, showing consistent performance across different SAE configurations (e.g., Gemma and Llama), which addressed my concerns about method dependence.
Contribution Acknowledgement: While the SAE methodology itself is not novel, the authors reframed SAEs as practical data labeling tools, showing their effectiveness in various data analysis tasks, which adds significant practical value.

Due to unforeseen delays, I was unable to submit the final justification on time. However, after thoroughly considering the paper and rebuttal, I have made my final assessment. Overall, the paper presents a valuable and practical method, addressing my concerns, and I recommend acceptance.

**Key Questions For Authors:**

1. The proposed approach relies on a specific pretrained SAE. How sensitive are the discovered concepts and downstream results to the choice of SAE (e.g., different SAE checkpoints, dictionary sizes, or training setups)?

**Limitations:**

yes

**Strengths And Weaknesses:**

Strengths

1. Interesting perspective on interpretable embeddings. The paper proposes using sparse autoencoders trained on LLM activations to construct document-level interpretable embeddings, where each dimension corresponds to a human-interpretable concept. This provides a potentially useful bridge between mechanistic interpretability and practical data analysis.


Weaknesses

1. Limited methodological novelty. The paper mainly applies existing sparse autoencoders trained on LLM activations as a feature extractor for downstream analysis tasks. The core SAE methodology itself is not developed or improved in this work, which makes the methodological contribution somewhat limited.

2. Strong dependence on a specific pretrained SAE. The approach relies on a particular SAE trained on hidden states of a specific model. However, the paper does not analyze how sensitive the results are to the choice of SAE, training data, layer selection, or dictionary size, leaving questions about robustness and generalizability. SAE methodology itself is still evolving, and the concepts extracted may vary depending on the SAE’s size and configuration.

---

> ### Author Rebuttal · Authors · 2026-03-31
>
> Thank you for your comments! We address your concerns below:
>
> > The core SAE methodology itself is not...improved in this work, which makes the methodological contribution somewhat limited.
>
> While we rely on pre-existing SAEs, this simplicity shows versatile data analysis is possible without additional training. Our key contribution is reframing SAEs not as interpretability tools that faithfully reconstruct model representations, but as practical data labelers. We introduce four new ways to leverage SAE-derived labels, substantially expanding on prior work focused on semantic retrieval (O’Neill 2024 [1]) and hypothesis generation (Movva 2025 [2]). Additionally, while prior work focuses on SAEs trained on dense document embeddings, we work with SAEs trained on token-level LLM representations, enabling the capture of fine-grained, token-specific features.
>
> > Strong dependence on a specific pretrained SAE.
>
> For detailed plots, please visit our supplementary material at https://docs.google.com/document/d/16fo_FI8UvZVuya8bEwiAHEMxaHEt55KSYGDQWoYBwPg/edit?usp=sharing
>
> **We have reproduced our ground-truth experiments using Gemma-2-2B/9B-PT SAEs, finding that Gemma SAEs work similarly well**
> - Retrieval: Our original paper includes Llama-3.1-8B-IT SAE’s performance on the retrieval benchmark with the 70B SAEs (Figure 6), showing comparable but slightly worse performance. We have now run our benchmark on Gemma-2-2B-PT and Gemma-2-9B-PT SAEs across various widths. The MAPs are comparable with the existing Llama-3.3-70B-IT SAE.
>     - **Performance increases with width and modestly with model size**: On Gemma2-2B, performance consistently increases as width goes from 16k to 65k, and similarly for 9B from 16k to 131k. Gemma-2-9B 16k achieved better performance than Gemma-2-2B 16k, but this improvement is less pronounced. They both perform similarly to the Llama 70B model. This suggests that SAE width matters more than reader model size.
>     - **Pretrained vs instruction-tuned**: The Llama 70B SAE was trained on an instruction-tuned model and performed well on the “chat responses” retrieval dataset. The Gemma-2 SAEs were on the pre-trained model and performed slightly better on the Pile but slightly worse on prompts/responses.
> - Diffing: On the tones and movies datasets, **Gemma2-2B 16k and 65k SAEs identify ground-truth differences**, although the labels are less directly related than Llama 70B SAE. The 65k SAE performs better than the 16k SAE (e.g. on movies dataset, mean surface similarity of 0.47 vs 0.40).
> - Correlations: **Gemma2-2B 65k SAEs can recover 3 out of 4 of the injected correlations** (except conservative-academic_style), albeit with more noise (i.e. there are many more correlated latents). However, the 16k SAE could not recover the injected correlations because it lacks relevant features e.g. “Croatian/Slavic” or “slang” features.
> - Clustering: On the synthetic news dataset, Gemma2-2B 16k and 65k SAEs can separate along each of the 4 axes with F1 scores 0.47-0.93, comparable to the original SAEs.
>
> In real-world settings where we analyze unlabeled datasets like model responses, we note that SAEs have different features and will naturally surface different insights. Nonetheless, we produce our findings on model diffing (Section 4.1) using Goodfire’s Llama 8B SAE. We recover many of the original differences in the main text (e.g. Grok-4’s propensity to clarify ambiguities) as well as achieve similar, even better frequency differences than the Llama 70B SAE.
>
> | Model              | 8B SAE | 70B SAE | LLM-C (baseline) |
> |--------------------|--------|---------|------------------|
> | Grok-4             | 28.1   | **32.9**    | 14.7             |
> | GPT-OSS-120B       | **31.7**   | 22.3    | 16.6             |
> | Gemini 2.5 Pro     | **41.3**   | 27.1    | 13.1             |
> | Llava v. Vicuna    | **9.3**    | 5.9     | 3.3              |
> | Deployment         | 15.6   | **15.9**    | 9.9              |
> | Evaluation         | 18.7   | **19.6**    | 13.8             |
>
> We also rigorously evaluate the quality of feature labels across Llama 8B and Llama 70B SAEs in Appendix I, where we compute the F1 score for SAE latents using ground truth determined by LLM judge. We have since expanded the test datasets to LMSYS-1M, European legislation (coastalcph/lex_glue), and medical research (TimSchopf/medical_abstracts). The SAEs have similar F1 scores across the datasets, suggesting that the 8B SAE would be an effective alternative to the 70B SAE. While both SAEs perform better on in-distribution datasets, relabeling features remains a consistently effective strategy for raising F1 scores on OOD data.
>
> Overall, results suggest SAE width matters more than model size, yielding more specific (but redundant) latents.
>
> [1] O'Neill et al. Disentangling Dense Embeddings with Sparse Autoencoders. arXiv preprint arXiv:2408.00657.
> [2] Movva, R. et al. Sparse Autoencoders for Hypothesis Generation. arXiv preprint arXiv:2502.04382.

---

> > ### Author Rebuttal · Reviewer_vJc7 · 2026-04-01
> >
> > My concerns have been largely addressed, and I am now happy to increase my score to a 5 Accept.

---

> > > ### Author Response · Authors · 2026-04-08
> > >
> > > Thank you for the helpful comments! We are glad our additional experiments have addressed your concerns, and we would gently remind you to edit the score if you were planning to do so.

---

### Official Review · Reviewer_xoqR · 2026-03-12

**Soundness:** 2
**Presentation:** 2
**Significance:** 3
**Originality:** 3
**Overall Recommendation:** 3
**Confidence:** 3

**Summary:**

This paper proposes using sparse autoencoder (SAE) embeddings as a toolkit for large-scale text corpus analysis, demonstrating their utility across four tasks: data diffing, correlation discovery, clustering, and property-based retrieval. The central claim is that SAE embeddings are more cost-effective than LLM-based annotation pipelines and more controllable than dense embeddings, because their dimensions map to interpretable concepts. The two case studies on OpenAI model behavior over time and Tulu-3 trigger phrase discovery are the most compelling demonstrations of the approach.

**Compliance With Llm Reviewing Policy:**

Affirmed.

**Key Questions For Authors:**

- How do you expect results to change when using SAEs trained on different layers, different base models, or SAEs with substantially different dictionary sizes? Without some additional experiments, it is difficult to assess whether the reported gains are a property of one SAE embedding or are more general.

- How do you distinguish a genuine semantic co-occurrence from noise or an artifact of the SAE's training distribution, and what would a negative result look like under your current framework? Would it be possible to run a small-scale experiment with human labels and compare to the llm judge?

- Since all texts are truncated to 2048 tokens, the method does not operate on full documents.. Did you observe any performance degradation on longer documents relative to shorter ones, and do you have plans to address this limitation as SAE context windows grow? Or potentially explore representing a document with averaging over multiple context windows?

**Limitations:**

yes

**Strengths And Weaknesses:**

__Strengths__
- The core idea of leveraging SAE embeddings for data analysis is genuinely novel and well-motivated. SAEs have primarily been studied as a mechanistic interpretability tool, and repositioning them as a data analysis primitive is an interesting and under-explored direction. The claim that SAE embeddings uncover semantic differences at 2 to 8x lower cost than LLMs is a strong practical result. The OpenAI model behavior case study is particularly compelling and represents, in my opinion, the strongest contribution of the paper. The finding that Grok-4 clarifies ambiguities more frequently than nine other frontier models is a concrete, interpretable result that demonstrates real-world value.


__Weaknesses__

- __Generalizability of the SAE choice__. All experiments use Goodfire SAEs trained on layer 50 hidden states of Llama 3.3 70B. The most pressing unanswered question is whether results generalize beyond this particular SAE on this particular layer of this particular model, or whether the paper is implicitly overfitting to an especially well-trained SAE. A brief ablation over layers or a comparison with a second SAE architecture would substantially strengthen the empirical claims.

- __Context window truncation.__ All texts are limited to 2048 tokens due to the model's context window, meaning that the proposed method does not produce true document-level embeddings. The paper does not discuss whether this truncation introduces systematic biases, particularly for long documents where the semantic content may appear later in the text.

- __Presentation quality.__ Many figures and tables are difficult to read. They are too small and insufficiently explained in the main text. For Figure 3 the text claims correlations are recovered across all splits, but this is not visually apparent from the figure. The results are insufficiently discussed in relation to one another, and the paper reads as a laundry list of experiments without a coherent narrative connecting them beyond the broad framing of SAE embeddings for data analysis. In my opinion, significant reorganization is needed.

- __Correlation analysis.__ The co-occurrence findings are presented as meaningful but the validation is subjective. The paper relies on an LLM judge to assess these relationships, and no calibration or correlation analysis is run against the judge's outputs to establish reliability. A more rigorous evaluation of whether these co-occurrences reflect genuine semantic structure rather than noise/llm bias is required.

---

> ### Author Rebuttal · Authors · 2026-03-31
>
> Thank you for your constructive comments! We appreciate that you think our case studies are “compelling” and that the core idea is “genuinely novel and well-motivated”. We address each of your concerns below.
>
> > Generalizability of the SAE choice...the most pressing unanswered question is whether results generalize beyond this particular SAE...a brief ablation over layers or a comparison with a second SAE architecture would substantially strengthen the empirical claims.
>
> We have run additional experiments using Gemma 2 SAEs and Llama 8B SAE. Please see our response to reviewer vjc7 on this subject under “Strong dependence on a specific pretrained SAE.”
>
> > Context window truncation. All texts are limited to 2048 tokens...the paper does not discuss whether this truncation introduces systematic biases, particularly for long documents where the semantic content may appear later in the text...Did you observe any performance degradation on longer documents relative to shorter ones, and do you have plans to address this limitation as SAE context windows grow? Or potentially explore representing a document with averaging over multiple context windows?
>
> Supplementary material: https://docs.google.com/document/d/16fo_FI8UvZVuya8bEwiAHEMxaHEt55KSYGDQWoYBwPg/edit?usp=sharing
>
> Thank you for pointing this out. We agree that this truncation may cause concepts that are only mentioned later on to be entirely missed. To overcome this, a user could divide the document into chunks that fit in the context window and simply pool all activations, which may miss out some contextual information but should still capture mostly correct features.
>
> We have since run a sanity check experiment: we plot the cumulative number of unique features fired so far against token position for a sample of our Pile dataset. This curve follows a logarithmic pattern (see supplementary material), meaning the later in the document, the fewer new features fire—this is expected as features related to e.g. style or general topic of the document would likely fire early on as soon as mentioned. We think this log-like relationship gives us some evidence that any bias in our results is likely small. We will add some discussion of this to the Appendix.
>
> > Presentation quality. Many figures and tables are difficult to read. They are too small and insufficiently explained in the main text.
>
> Please see our response to reviewer bYTG under “Overall presentation of paper”.
>
> > Correlation analysis. The co-occurrence findings are presented as meaningful but the validation is subjective. The paper relies on an LLM judge to assess these relationships, and no calibration or correlation analysis is run against the judge's outputs to establish reliability. A more rigorous evaluation of whether these co-occurrences reflect genuine semantic structure rather than noise/llm bias is required. & How do you distinguish a genuine semantic co-occurrence from noise or an artifact of the SAE's training distribution, and what would a negative result look like under your current framework? Would it be possible to run a small-scale experiment with human labels and compare to the llm judge?
>
> We agree that there is some inherent subjectiveness in the notion of “correlated concepts”. However, we emphasize that when verifying correlations, our LLM judge is never told to look for a co-occurrence of two concepts, so it is not biased to search for that—instead, each concept is always labelled independently, and pairwise correlations computed afterwards. In other words, if the LLM judge is correct in labelling the presence/absence of each concept X (e.g. “offensive”) and Y (“religion”) independently, then the calculated correlation of X and Y should be correct. We acknowledge there is some subjectivity in the concepts e.g. what is “offensive”, which is why we have tried to be as specific in our prompts as possible (see Appendix K.2).
>
> It is true that some co-occurrences of SAE features are not genuine semantic co-occurrences: e.g. pairs of latents that always co-occur on the same token due to the SAE’s training distribution. These are the “false positives” in the correlations i.e. correlations raised by the SAE that are not verified by the judge, and these were quantified in Figure 4 (those with low verified NPMI would be false positives). We agree that there are indeed many false positives in the SAE-discovered co-occurrences, but these can be checked for by reading co-occurring documents.
>
> Overall, we believe that the additional experiments detailed above on different SAEs (Gemma, Llama 8B) have strengthened the paper by showing that our findings are indeed not overfitted to one SAE or one model, although performance depends on SAE quality and specific qualitative insights would differ depending on an SAE’s available features. We also address concerns about context window truncation and correlation verification. We hope this will encourage the reviewer to reconsider their score.

---

> > ### Author Rebuttal · Reviewer_xoqR · 2026-04-01
> >
> > After reading reviews and rebuttals, I believe my score is fair and that paper requires changes that cannot be fully realized before the camera ready version.

---

### Official Review · Reviewer_ERpJ · 2026-03-13

**Soundness:** 3
**Presentation:** 3
**Significance:** 3
**Originality:** 3
**Overall Recommendation:** 5
**Confidence:** 4

**Summary:**

This paper proposes a method based on **Sparse Autoencoders (SAE)** to construct interpretable text embedding representations. Traditional text embeddings are typically dense representations, where each dimension does not have a clear semantic meaning, making it difficult to explain why a model makes certain predictions. In contrast, this work uses SAE to learn representations from large language models, producing a feature dictionary with explicit semantic meanings, where each dimension corresponds to an interpretable concept. This approach addresses the lack of interpretability in conventional embeddings.

The authors further evaluate the effectiveness of the SAE embeddings through four tasks: (1) dataset difference analysis; (2) concept correlation discovery; (3) controllable clustering; and (4) attribute-based retrieval. Experimental results show that the proposed method can uncover hidden semantic structures in datasets and provide a new tool for analyzing the behavior of language models.

**Compliance With Llm Reviewing Policy:**

Affirmed.

**Final Justification:**

The reviewer thanks the authors for their excellent work, which benefits both the reviewers and future readers. I will support the acceptance of this paper and increase my score accordingly.

**Key Questions For Authors:**

1. In the experiments, LLMs are used to evaluate the results. If different LLMs are used for evaluation, would the experimental results change?

2. Have the authors considered comparing their approach with other interpretability methods?

3. The main novelty of this work lies in a shift in perspective—transforming **SAE from a model interpretability tool into a data analysis tool**. However, the related work section mentions that **Movva et al. (2025)** has already explored using SAE to generate feature hypotheses. Could the authors further clarify the key differences and advantages of this work compared with these contemporaneous studies?

**Limitations:**

yes

**Strengths And Weaknesses:**

**Strengths:**

1. This paper proposes the use of **Sparse Autoencoders (SAE)** to construct highly controllable and interpretable embeddings, addressing the issue that traditional embedding vectors lack interpretability at the dimension level.
2. The paper provides a systematic data analysis framework covering four tasks (difference analysis, correlation analysis, clustering, and retrieval), offering researchers a new tool for understanding both datasets and model behavior.
3. The paper conducts dedicated experimental evaluations for the four tasks—difference analysis, correlation analysis, clustering, and retrieval—and compares the results with current state-of-the-art baselines in each setting.
4. In the dataset difference analysis task, the paper also performs cost analysis, showing that SAE is **2–8× cheaper than directly using LLMs**. Moreover, SAE embeddings can be reused once computed, while LLM-based methods require repeated processing.
5. Traditional embedding-based clustering often lacks control over clustering dimensions. The proposed SAE embedding allows clustering to be performed on specific semantic features, which is highly valuable for data analysis.
6. The paper further demonstrates practical value through two real-world industrial case studies: **“tracking model evolution”** and **“debugging training data.”**

**Weaknesses:**

1. In Experiment 3 (the text clustering task), the evaluation of single-cluster quality relies on a large language model (LLM) instead of traditional metrics such as the silhouette coefficient. This choice may reduce the objectivity and rigor of the evaluation.
2. When generating document embeddings, the paper uses **max pooling** to aggregate token activations. Although this approach can capture sparse concepts, it may be overly sensitive to single-point noise, as a misprediction on a single token can directly determine the score of the entire document on a particular feature dimension.

---

> ### Author Rebuttal · Authors · 2026-03-31
>
> Thank you for your positive feedback! We address your questions below.
>
> > In Experiment 3 (the text clustering task), the evaluation of single-cluster quality relies on a large language model (LLM) instead of traditional metrics such as the silhouette coefficient. This choice may reduce the objectivity and rigor of the evaluation.
>
> We used single-cluster quality as a measure for how accurate the clustering is, which is a proxy for how “interpretable and useful” a cluster is. This is important as a high silhouette score alone may not mean SAE clusters are meaningful—it could have been that SAE embeddings create well-separated clusters due to some geometry of the SAE but these clusters are not meaningful.
>
> In the case of GSM-8K, we find that silhouette scores for both dense embeddings and SAE embeddings are low, implying that the text could not be cleanly partitioned in geometric space. For dense embeddings n_clusters=3, spectral clustering gives a silhouette score of 0.0713 and K-means clustering gives 0.0730, so the text is not easily separated by dense embeddings (likely since they are all similar math solutions). Spectral clustering on SAE embeddings gives a similarly low 0.0200, so texts are also not easily separated, but the SAE embeddings still find meaningful reasoning-style-related cluster centers that are validated with their accuracy.
>
> > When generating document embeddings, the paper uses max pooling to aggregate token activations. Although this approach can capture sparse concepts, it may be overly sensitive to single-point noise, as a misprediction on a single token can directly determine the score of the entire document on a particular feature dimension.
>
> We agree that the max-pooling on its own may be sensitive to single-point noise— for instance, on an average ChatbotArena response, about 65% of latents active in a document are latents that only fire once on that document. This allows us to capture sparse concepts that only appear once but may be sensitive to noise. We sometimes apply thresholds to filter out latents that activate below a certain strength (e.g. correlations). This threshold is a hyperparameter that can be adjusted for an analysis task based on the tolerance for noise. In other experiment settings like diffing, subtracting feature frequencies naturally cancels out some of the noise. We will update Section 3 to clarify this tradeoff.
>
> > In the experiments, LLMs are used to evaluate the results. If different LLMs are used for evaluation, would the experimental results change?
>
> Our core use for LLM judges is to evaluate whether a hypothesis/feature is present in a document or not. To evaluate inter-LLM agreement, we took 500 random latents from both LLama 70B SAE and LLama 8B SAE and used three judges (Gemini 2.5 Flash, GPT-5.2, Haiku 4.5) to compute F1 scores of the latents on a 1K sample from LMSYS-1M. We measure Cohen’s kappa between every pair of judges, finding that they have high agreement with one another (0.6-0.7).
>
> |           | Gemini 2.5 Flash| GPT 5.2 | Haiku 4.5 |
> |-----------|--------|---------|-----------|
> | Gemini 2.5 Flash   | 1.000  | 0.700   | 0.633     |
> | GPT 5.2   | 0.700  | 1.000   | 0.719     |
> | Haiku 4.5 | 0.633  | 0.719   | 1.000     |
>
> We also measure the Spearman rank correlation between judges. All pairs have high correlation, indicating that even if a judge tends to say “yes” or “no” more often, its overall ranking of features by frequency is similar to that of other judges.
>
> |           | Gemini 2.5 Flash | GPT 5.2 | Haiku 4.5 |
> |-----------|--------|---------|-----------|
> | Gemini 2.5 Flash   | 1.000  | 0.899   | 0.905     |
> | GPT 5.2   | 0.899  | 1.000   | 0.914     |
> | Haiku 4.5 | 0.905  | 0.914   | 1.000     |
>
> > Have the authors considered comparing their approach with other interpretability methods?
>
> We do compare SAEs with other interpretability methods in the paper. On correlations, we compare against correlated topic models (Figure 4), and on retrieval, we use BM25 as a baseline. We find that SAEs are less noisy than CTMs and have stronger retrieval performance than BM25.
>
> > ...the related work section mentions that Movva et al. (2025) has already explored using SAE to generate feature hypotheses. Could the authors further clarify the key differences and advantages of this work compared with these contemporaneous studies?
>
> Movva et al. focus on SAEs trained on dense document embeddings, whereas we work with pre-existing SAEs trained on token-level LLM representations, which removes training overhead and enables the capture of fine-grained, token-specific features within longer model responses. Our work aims to advance understanding of model data and also explores different tasks like feature correlations and targeted clustering.
>
> Overall, we thank the reviewer for bringing the discussion on silhouette coefficient, which we will clarify in the paper, as well as the max-pooling point which we will update.

---

> > ### Author Rebuttal · Reviewer_ERpJ · 2026-04-03
> >
> > The reviewer thanks the authors for their excellent work, which benefits both the reviewers and future readers. I will support the acceptance of this paper and increase my score accordingly.

---

### Official Review · Reviewer_bYTG · 2026-03-16

**Soundness:** 3
**Presentation:** 1
**Significance:** 3
**Originality:** 2
**Overall Recommendation:** 4
**Confidence:** 4

**Summary:**

This paper proposes using sparse autoencoders (SAEs) for general-purpose text data analysis tasks. The text documents are passed through a reader LLM, and SAE activations for layer to produce a high-dimensional embedding vector in which each latent corresponds to a specific concept. These SAE embeddings are then applied to four tasks: (1) dataset diffing (finding property-frequency differences between corpora), (2) correlation discovery (finding unexpected co-occurring concept pairs), (3) clustering (grouping documents by axes of interest), and (4) property-based retrieval (ranking documents by implicit attributes). The authors demonstrate that SAE embeddings show a competitive performance compared to LLM-based baselines on dataset diffing tasks, at finding spurious correlations, and property-based retrieval. They show two usecases: tracking qualitative behavioral changes across OpenAI model generations, and identifying a specific learned trigger phrase ("I hope it is correct") in the Tulu-3 post-training dataset.

**Compliance With Llm Reviewing Policy:**

Affirmed.

**Key Questions For Authors:**

1. Can the authors fix figure 6? It currently overlaps with the surrounding text.
2. The document-level SAE embedding is constructed by max-pooling activations across tokens. This could introduce noise for retrieval and clustering as a latent fire at the document level if it activates on any token. Did the authors experiment with alternative pooling strategies?

**Limitations:**

yes

**Strengths And Weaknesses:**

Strengths
1. The paper is comprehensive with results validated on both controlled and real-world settings. The case studies show interesting novel insights that support the usage of SAE embeddings.
2. The authors make a good effort towards transparency by including all prompts templates, detailed ablation results and additional information related to the experiments.

Weaknesses:
1. The overall presentation of the paper is quite dense, with the authors attempting to present wide coverage of their experimental results in lieu of overall readability of the paper. This is particularly visible in their use of extremely small plots and tables that do not present the information clearly. Significant restructuring of the paper may be need to present the most important results in the main body.
2. Since all the steps in the evaluation pipeline involve an LLM, this can create a risk that the SAE pipeline appears to outperform baselines partly because of LLM-in-the-loop circularity throughout the evaluation.
3. The performance degrades on out-of-distribution datasets, so it might be worth addressing how this approach can be applied to domains such as clinical text, legal documents, code, or non-English text that might not be extensively covered by SAE training corpus.

---

> ### Author Rebuttal · Authors · 2026-03-31
>
> Thank you for your feedback! We are glad you found the case studies to be “interesting novel insights”, and we address each of your points below:
>
> > The overall presentation of the paper is quite dense, with the authors attempting to present wide coverage of their experimental results in lieu of overall readability of the paper. This is particularly visible in their use of extremely small plots and tables that do not present the information clearly. Significant restructuring of the paper may be need to present the most important results in the main body.
>
> Thank you for the feedback. We intend to use the extra page to give more space to existing figures (Figure 1-3, Table 2-3, Figure 5). In each experiment task, we aim to show that SAEs are stronger than baselines at discovering insights in both synthetic ground-truth tasks and real-world settings. We will condense each part to focus on the analysis of one main result. We will move Table 1, Table 3, and Figure 4 to the appendix and limit the discussion for the real-world setting in Correlations (Section 4.2) to the CivilComments dataset, as Section 5.2 (Debugging Tulu) already shows a case study on a training dataset. Our goal is to show simple, high-impact use cases of SAEs within practical data analysis that future work can build on. If you have any comments on which parts are the most crucial, we would love to hear!
>
> > Since all the steps in the evaluation pipeline involve an LLM, this can create a risk that the SAE pipeline appears to outperform baselines partly because of LLM-in-the-loop circularity throughout the evaluation.
>
> We acknowledge biases can arise from involving LLMs in our methodology and evaluation.
>
> Although we often lean on LLM judges for evaluation, each of our experiments has a ground truth task to verify that SAEs can perform the task on their own. For instance, in diffing (Section 4.1), we test whether the top feature differences correspond with known dataset differences (e.g. movie genre).
>
> On the methodology side—while we use LLMs to relabel SAE features for model diffing, we note that SAEs outperform the baselines despite relying on the same LLM (Gemini 2.5 Flash), which implies that SAEs play some crucial role for surfacing patterns that wouldn’t be easily surfaced from passing the dataset to the LLM. For correlations and clustering, we do not rely on LLMs for the SAE method. For retrieval, we use LLMs for feature selection, but also use LLMs in the dense embeddings baselines for query augmentation. Overall, we use LLMs in the pipeline to achieve the best possible performance for real-world hypothesis generation, but still validate the SAE method without LLMs.
>
> > The performance degrades on out-of-distribution datasets, so it might be worth addressing how this approach can be applied to domains such as clinical text, legal documents, code, or non-English text that might not be extensively covered by SAE training corpus
>
> While performance can degrade for OOD datasets, we can either relabel features or train the SAE from scratch to improve feature quality. In Appendix I, we show that relabeling features consistently improves their F1 scores across all datasets and SAEs (LLama 70B SAE and LLama 8B SAE). We furthermore relabel features on European legislation and medical research artifacts and find similar conclusions (see supplementary material: https://docs.google.com/document/d/16fo_FI8UvZVuya8bEwiAHEMxaHEt55KSYGDQWoYBwPg/edit?usp=sharing). Although we focus on pre-existing SAEs in this work to simplify our methodology, prior work (Movva et al., 2025) shows that training SAEs from scratch can be effective for hypothesis generation and would likely boost performance on OOD datasets.
>
> > Can the authors fix figure 6? It currently overlaps with the surrounding text.
>
> Sorry about that, it will be fixed!
>
> > The document-level SAE embedding is constructed by max-pooling activations across tokens. This could introduce noise for retrieval and clustering as a latent fire at the document level if it activates on any token. Did the authors experiment with alternative pooling strategies?
>
> See response for reviewer ERpJ on max-pooling.
>
> Overall, we thank the reviewer for their comments on restructuring the paper and will make those edits, and we invite any further suggestions. We also address the reviewer’s concerns on LLM reliance and out-of-distribution datasets.

---

> > ### Author Rebuttal · Reviewer_bYTG · 2026-04-04
> >
> > Thanks for the response. I have two comments:
> > 1. On restructuring, I would additionally suggest keeping a single consolidated ablation table in the main body.
> > 2. Since the LLM baseline outperforms the SAE on the movies dataset, some analysis and a discussion of when LLM-based diffing remains preferable would be important to add.

---

> > > ### Author Response · Authors · 2026-04-07
> > >
> > > > **Consolidated tables**
> > >
> > > Thank you for the suggestion! We will include a table in the main text summarizing ground-truth results of the Gemma SAEs across all four tasks and provide a link to the appendix for the complete set of results.
> > >
> > > > **LLM-based diffing analysis**
> > >
> > > The movie experiment did not include feature relabeling, so SAE features often lacked direct alignment with genre categories. Instead, they surfaced related but indirect concepts (e.g., “will they/won’t they” tropes rather than “romance”, Table 1). In contrast, LLM-based diffing could more immediately notice these higher-level semantic categories, even when descriptions vary at the sub-genre level. This distinction reflects a broader pattern we also observe in model diffing (Section 4.1): LLM-based approaches tend to highlight abstract, high-level differences (e.g., “flawed reasoning”), whereas SAE features provide more specific, granular signals (e.g., “asking clarifying questions”). The method of choice depends on whether the user values abstraction or interpretability at a finer level.
> > >
> > > Dataset size is another key factor. The movies dataset is small enough to fit within an LLM’s context window, enabling direct comparison. For larger datasets, LLM-based diffing becomes less reliable, as it typically requires sampling rather than full coverage.
> > >
> > > Finally, LLM-based diffing is more expensive, particularly when diffing the same datasets repeatedly (e.g. multi-model comparisons, subsets of the same training corpus), whereas SAEs amortize this cost more effectively.
> > >
> > > Overall, LLM-based diffing remains preferable in settings with small datasets, sufficient budget, and a preference for high-level semantic comparisons, while SAEs are better suited for scalable, repeated, and more granular analysis.

---

### Decision · Program_Chairs · 2026-04-30

**Decision:**

Accept (regular)

**Comment:**

This paper suggests to use embeddings obtained with a pre-trained sparse autoencoder trained on LLM activations, as interpretable "text embeddings" to analyze text datasets: e.g. to summarize differences between two text datasets, or between outputs of two LLM models.

The reviewers liked this approach and also appreciated the data-science-like case studies. Some concerns were about using only one particular SAE, but in the rebuttal the authors provided additional results using other SAEs. My recommendation is "accept".

I fully agree with the reviewers that some of the tables and figures are way too crowded and the font is way too small. Please make sure to reformat all figures and tables to be comfortably readable in a printout.

Some further minor points:

* For some reason the paper is using square brackets for citations. I think it should be round brackets, according to the ICML's style guide.

* Sometimes \citet should be used instead of \citep, please check.

* The abstract says "SAE embeddings are more cost-effective and reliable than LLMs". The same statement about cost-effectiveness appears many times in the text. This is confusing to me because SAE embeddings are (obviously) some transformation of LLM embeddings! One pushes the text through an LLM, gets activations, then transforms them into the SAE representation. So saying that "SAE is more cost-effective than LLM" is confusing. I would suggest to rephrase to explain what you mean by "LLMs" in this sentence.